EMBO
Molecular Medicine

# High-throughput screening identifies suppressors of mitochondrial fragmentation in *OPA1* fibroblasts

Emma Cretin[1,2] (ID), Priscilla Lopes[1], Elodie Vimont[1], Takashi Tatsuta[3] (ID), Thomas Langer[3,4] (ID), Anastasia Gazi[5] (ID), Martin Sachse[5] (ID), Patrick Yu-Wai-Man[6,7,8,9] (ID), Pascal Reynier[10,11] (ID) & Timothy Wai[1,2,*] (ID)

## Abstract

Mutations in OPA1 cause autosomal dominant optic atrophy (DOA) as well as DOA+, a phenotype characterized by more severe neurological deficits. OPA1 deficiency causes mitochondrial fragmentation and also disrupts cristae, respiration, mitochondrial DNA (mtDNA) maintenance, and cell viability. It has not yet been established whether phenotypic severity can be modulated by genetic modifiers of OPA1. We screened the entire known mitochondrial proteome (1,531 genes) to identify genes that control mitochondrial morphology using a first-in-kind imaging pipeline. We identified 145 known and novel candidate genes whose depletion promoted elongation or fragmentation of the mitochondrial network in control fibroblasts and 91 in DOA+ patient fibroblasts that prevented mitochondrial fragmentation, including phosphatidyl glycerophosphate synthase (*PGS1*). PGS1 depletion reduces CL content in mitochondria and rebalances mitochondrial dynamics in OPA1-deficient fibroblasts by inhibiting mitochondrial fission, which improves defective respiration, but does not rescue mtDNA depletion, cristae dysmorphology, or apoptotic sensitivity. Our data reveal that the multifaceted roles of OPA1 in mitochondria can be functionally uncoupled by modulating mitochondrial lipid metabolism, providing novel insights into the cellular relevance of mitochondrial fragmentation.

**Keywords** genetic modifiers; high-throughput screening; mitochondrial dynamics; OPA1; phospholipid metabolism
**Subject Categories** Genetics, Gene Therapy & Genetic Disease; Neuroscience; Organelles
See also: **JJ Collier & RW Taylor** (June 2021)

## Introduction

The morphology that mitochondria adapt within a cell is shaped by opposing events of membrane fusion and fission executed by dynamin-like GTPases (Giacomello *et al*, 2020). Fission is performed upon recruitment of dynamin-related protein 1 (DRP1, encoded by *DNM1L*) to the outer membrane (OMM) via its receptors mitochondrial fission factor (MFF) and mitochondrial division (MiD) 49 and 51, which coalesce at sites of contact with the endoplasmic reticulum (ER)(Friedman *et al*, 2011) in a manner that depends on the lipid composition of the OMM (Choi *et al*, 2006; Khacho *et al*, 2014). Mitochondrial fusion is controlled by Mitofusins (MFN) 1 and 2 at the outer membrane and optic atrophy protein 1 (OPA1) in the inner membrane (IMM) (Chen *et al*, 2003; Olichon *et al*, 2003; Cipolat *et al*, 2004). Post-translational modifications (PTM) of these proteins can regulate mitochondrial dynamics: DRP1 phosphorylation can alter the recruitment to future sites of mitochondrial division on OMM while at the IMM, proteolytic cleavage of OPA1 from L-OPA1 to S-OPA1 by the mitochondrial proteases OMA1 and the *i*-AAA protease YME1L balances the rates of fusion and fission in response to stress conditions and metabolic stimulation (MacVicar & Langer, 2016).

Mitochondrial shape can shift in response to cellular and extracellular cues both *in vitro* and *in vivo* (Twig *et al*, 2008; Gomes *et al*, 2011; Arruda *et al*, 2014; Khacho *et al*, 2014; Jacobi *et al*, 2015). Mitochondrial fusion has been proposed to preserve cellular integrity, increase ATP production, and maintain mitochondrial DNA levels (mtDNA) (Chen *et al*, 2010; Elachouri *et al*, 2011). Stress-induced mitochondrial hyperfusion (SiMH) is a cytoprotective response that occurs in response to exogenous cellular insults including protein synthesis inhibition and nutrient and oxygen deprivation (Tondera *et al*, 2009; Gomes *et al*, 2011; Rambold *et al*,

1   Mitochondrial Biology Group, Institut Pasteur, CNRS UMR 3691, Paris, France
2   Université de Paris, Paris, France
3   Max-Planck-Institute for Biology of Ageing, Cologne, Germany
4   Cologne Excellence Cluster on Cellular Stress Responses in Aging-Associated Diseases (CECAD), University of Cologne, Cologne, Germany
5   UTechS Ultrastructural Bio Imaging, Institut Pasteur, Paris, France
6   Cambridge Centre for Brain Repair and MRC Mitochondrial Biology Unit, Department of Clinical Neurosciences, University of Cambridge, Cambridge, UK
7   Cambridge Eye Unit, Addenbrooke's Hospital, Cambridge University Hospitals, Cambridge, UK
8   Moorfields Eye Hospital, London, UK
9   UCL Institute of Ophthalmology, University College London, London, UK
10  Laboratoire de Biochimie et biologie moléculaire, Centre Hospitalier Universitaire, Angers, France
11  Unité Mixte de Recherche MITOVASC, CNRS 6015, INSERM U1083, Université d'Angers, Angers, France
    *Corresponding author. Tel: +33 1 44 38 91 41; E-mail: timothy.wai@pasteur.fr

2011; Khacho *et al,* 2014) characterized by an elongation of the mitochondrial network resulting from unopposed fusion that requires OPA1 and MFN1 (but not MFN2) and the IMM proteolytic scaffold protein stomatin-like protein 2 (SLP2) (Tondera *et al,* 2009; Wai *et al,* 2016). SLP2 is a cardiolipin (CL)-binding protein that defines CL-rich membrane domains of the IMM. CL is a mitochondrial-specific non-bilayer-forming phospholipid that is implicated in a wide array of mitochondrial processes including apoptosis, respiratory chain assembly, protein import, inflammation, and mitochondrial dynamics (Claypool, 2009). The association between mitochondrial dynamics and lipids in mitochondrial and cellular homeostasis is well established, although the nature of this interdependence is less clear.

Unopposed fission causes mitochondrial fragmentation, which is associated with cellular dysfunction and has been observed in a variety of acquired and inborn disorders, in particular mitochondrial genetic diseases (MD) (Giacomello *et al,* 2020). Mutations in *OPA1,* which encodes for a dynamin-like GTPase protein, cause autosomal dominant optic atrophy (DOA). The majority of patients manifest isolated optic atrophy (DOA, MIM#165500), but a subgroup develop a more severe disseminated neurological phenotype as part of a DOA "plus" phenotype (DOA+, MIM#125250), including an early-onset Behr-like syndrome (MIM#210000) or encephalomyopathy (MIM# 616896) in a few reported patients with recessive *OPA1* mutations (Carelli *et al,* 2015; Spiegel *et al,* 2016). OPA1-deficient cells exhibit a fragmented mitochondrial network due to unopposed fission (Olichon *et al,* 2003; Cipolat *et al,* 2004). Beyond mitochondrial fusion, OPA1 plays essential roles in the maintenance of cristae shape, mtDNA levels, OXPHOS complex assembly, cellular proliferation, and apoptotic sensitivity (Giacomello *et al,* 2020). Overexpression of OPA1 can confer protection against apoptotic cell death (Varanita *et al,* 2015) without necessarily altering mitochondrial morphology (Frezza *et al,* 2006), leading to the notion that non-fusion roles of *OPA1* (e.g., cristae maintenance) are functionally separable from IMM fusion but this hypothesis has never been put to the test in OPA1 deficiency (Patten *et al,* 2014). Indeed, how OPA1 is capable of regulating different processes within mitochondria is unclear as is the cellular relevance of mitochondrial fragmentation in OPA1-deficient cells.

Mitochondrial morphology exists on a dynamic spectrum, with *fragmented* and *hypertubulated* (or *hyperfused*) referring to the characteristic network morphologies adopted by mitochondria in cells when fusion and fission are inhibited, respectively (Giacomello *et al,* 2020). Quantification of mitochondrial morphology performed by subjective, user-defined manual classification cells with aberrant mitochondrial networks caused by inhibited fusion (Ishihara *et al,* 2006) or fission (Osellame *et al,* 2016) as well as enhanced fusion (Tondera *et al,* 2009; Wai *et al,* 2016) or fission (Anand *et al,* 2014) has been successfully applied for the over two decades. More recently, the use of computer-assisted segmentation measurement of mitochondrial features (Kane *et al,* 2017), such as the length, width, or aspect ratio of mitochondria has gained traction (Iannetti *et al,* 2016). However, major drawbacks to these approaches remain the manual collection of images, the possibility of user bias, and the laborious segmentation of mitochondria needed to ascribe morphological traits. The latter also requires spatial resolution at the physical limits of light microscopy in order to accurately and unequivocally separate one mitochondrion from the next. While

recent advances in super-resolution nanoscopy of mitochondria may soon render this concern moot (Jakobs *et al,* 2020), only a handful of laboratories have successfully applied this technology for high-resolution mitochondrial imaging and its application to high-throughput imaging has yet to be established.

In this study, we developed a first-in-kind, high-throughput imaging screening pipeline and identified known and novel mitochondrial genes that can modulate mitochondrial morphology in healthy human fibroblasts and prevent mitochondrial fragmentation in *OPA1* patient fibroblasts, most of which have never previously been linked to mitochondrial dynamics. Among the 91 candidate genes found to suppress mitochondrial fragmentation, we discovered that depletion of PGS1, the mitochondrial phosphatidyl glycerophosphate (PGP) synthase, lowers cardiolipin levels, inhibits mitochondrial fission and rescues mitochondrial fragmentation and respiration in OPA1-deficient mouse embryonic fibroblasts. Our data unravel an unexpected role of PGS1 in the regulation of mitochondrial form and function.

# Results

### Inhibiting fission rescues mitochondrial fragmentation in *OPA1* patient fibroblasts

To overcome limitations of conventional approaches for imaging and quantification of mitochondria in cells, we developed a high-content imaging pipeline using confocal spinning disk fluorescence microscopy compatible with multi-well, high-throughput automated imaging of live or fixed cells (Fig EV1). We adopted an image analysis pipeline (Dataset EV1) that automatically executes cell segmentation enabling the single-cell classification of mitochondrial morphology using supervised machine learning (ML) algorithms trained on defined classes of mitochondrial morphologies, which do not rely on measuring the absolute length or width of a mitochondrion. Instead, training sets (ground truths) were empirically generated by knocking down genes whose depletion is known to provoke either increased or decreased mitochondrial network lengths. To promote mitochondrial fragmentation, we depleted control fibroblasts of *OPA1,* and to define hypertubulated mitochondria, we inhibited mitochondrial fission by downregulation of *DNM1L.* To define normal, tubular mitochondrial morphology, we treated control cells with non-targeting (NT) siRNAs. Confocal images of hundreds of cells (315–586 cells/training condition) acquired from these training sets were used as ground truths to train the supervised ML algorithm to classify cells as either fragmented, normal, or hypertubulated (Fig 1A) during each imaging experiment. This approach proved tremendously robust: siRNA-mediated induction of fragmentation of either *YME1L* or *MFN1/2* was accurately recognized as such by supervised ML training of mitochondrial fragmentation using *OPA1* siRNAs (Appendix Fig S1A) and chemical induction of fission with the protonophore carbonyl cyanide m-chlorophenyl hydrazone (CCCP) or hyperfusion with the cytosolic protein synthesis inhibitor cycloheximide (CHX) could be used to accurately quantify mitochondrial fragmentation in OPA1-depleted fibroblasts (Appendix Fig S1B). Together, these data validate the supervised ML approach to mitochondrial morphology quantification as a rapid, robust, and unbiased approach for the quantitative

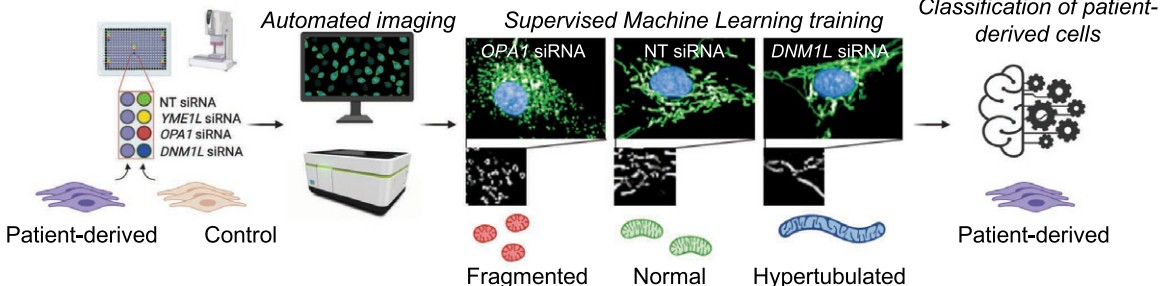

Figure 1.

**Figure 1. Inhibition of mitochondrial division prevents mitochondrial fragmentation caused by OPA1 deficiency in DOA+ patient-derived fibroblasts.**

A   Schematic of supervised machine learning (ML) mitochondrial morphology imaging and quantification pipeline. Fibroblasts plated in 384-well plates are stained for mitochondria (anti-TOMM40, green), nuclei (DAPI, blue), and cell body (CellMask, blue). Supervised ML training performed on cells with fragmented (*OPA1* or *YME1L* siRNA), normal (non-targeting NT siRNA), and hypertubular (*DNM1L* siRNA) mitochondria. Automatic single-cell trinary classification of control (CTL-1, 2, 3) and *OPA1$^{S545R}$* patient fibroblasts by supervised ML.

B   Representative confocal images of control (CTL-1, 2, 3) and DOA+ patient fibroblasts carrying indicated mono-allelic mutations imaged as described in (A). Scale bar = 20 μm. Passage number between P12–15.

C   Mitochondrial morphology quantification of (B). Data represent mean ± SD of two independent experiments, (195–2,496 cells per cell line), One-way ANOVA; **$P < 0.01$, ****$P < 0.0001$, ns; not significant.

D   Representative confocal images of control (CTL-1) and *OPA1$^{S545R}$* patient fibroblasts treated with *OPA1*, *DNM1L*, or non-targeting (NT) siRNAs for 72 h and imaged as described in (A). Scale bar = 20 μm. Passage number between P12–14.

E   Mitochondrial morphology quantification of (D). Data represent mean ± SD of three independent experiments (3,219–5,857 cells per cell line), One-way ANOVA; ****$P < 0.0001$, ns; not significant.

F   Representative confocal images of control (CTL-1) and *OPA1$^{S545R}$* patient fibroblasts treated with 50 μM cycloheximide (CHX) where indicated for 6 h. Imaging as described in (A). Scale bar = 20 μm. Passage number between P14–P15.

G   Mitochondrial morphology quantification of (F). Data represent mean ± SD of two independent experiments (879–4,154 cells per cell line), One-way ANOVA; ****$P < 0.0001$, ns; not significant.

Source data are available online for this figure.

assessment of mitochondrial shape in fibroblasts using a variety of genetic or chemical training sets as ground truths.

Genetic knockouts, siRNA depletion, and chemical modulation experiments induce drastic alterations in mitochondrial shape that are easily recognizable but do not necessarily reflect the phenotypic severity observed in patient cells or disease models, which are often hypomorphic, yielding more subtle biochemical and cell biological alterations. To determine whether our supervised ML approach to mitochondrial morphology quantification was compatible with the high-throughput interrogation of patient cells, we imaged and analyzed control and DOA+ patient-derived skin fibroblasts carrying pathogenic, mono-allelic variants in *OPA1* known to cause mitochondrial fragmentation including p.Arg445His (*OPA1$^{R445H}$*) and p.Ser545Arg (*OPA1$^{S545R}$*) (Amati-Bonneau *et al*, 2005; Yu-Wai-Man *et al*, 2010) and pathogenic variants whose effects on mitochondrial morphology have not yet been reported such as p.Ile432Val (*OPA1$^{I432V}$*), c.2356G>T (*OPA1$^{c.2356G>T}$*), and p.Gln297*(*OPA1$^{Q297X}$*) (Yu-Wai-Man *et al*, 2010) (Fig 1B and Table 1). Our analyses revealed both *OPA1$^{S545R}$* and, to a lesser extent, *OPA1$^{R445H}$* patient fibroblasts exhibited significant increases in the proportion of cells with a fragmented mitochondrial phenotype: 45.2 ± 5.3% of *OPA1$^{S545R}$* fibroblasts (2,282 cells analyzed) and 16.8 ± 9.2% of *OPA1$^{R445H}$* fibroblasts (2,683 cells analyzed) were fragmented compared with 4.5–11.1% of control fibroblasts from three healthy, unrelated individuals (CTL-1; 11.1 ± 7.1%, CTL-2; 6.1 ± 3.2%, CTL-3; 4.5 ± 5.2%, 879–3,823 cells analyzed; Fig 1B and C). These data are in accordance with previous measurements made in these cells using manual, lower-throughput imaging and quantification methods (Amati-Bonneau *et al*, 2005; Kane *et al*, 2017). Curiously, we did not detect significant mitochondrial morphology defects in *OPA1$^{I432V}$*, *OPA1$^{c.2356-1G>T}$* nor *OPA1$^{Q297X}$* patient fibroblasts even though they were derived from patients also suffering from the same pathology: DOA+. Western blot analyses revealed a reduction in OPA1 protein of 58.2% ± 9.2 in *OPA1$^{Q297X}$* lysates (Appendix Fig S1C) relative to control fibroblasts and no significant differences in other patient-derived fibroblasts. Aberrant mitochondrial morphology measured in patient-derived fibroblasts did not correlate with the steady-state levels of OPA1 nor with the reported clinical symptoms (Table 1), suggesting that additional factors beyond pathogenic mutations in *OPA1* may be capable of modulating mitochondrial morphology.

In animal models of MD, mitochondrial fragmentation can be rebalanced by additional inhibition of mitochondrial fission (Wai *et al*, 2015; Yamada *et al*, 2018), but this approach has not been tested in humans. To test whether decreasing mitochondrial fission is capable of rebalancing mitochondrial morphology in *OPA1* mutant patient fibroblasts, we knocked down *DNM1L* by siRNA (Fig 1D). DRP1 depletion in *OPA1$^{S545R}$* fibroblasts led to an increased proportion of cells with normal and hypertubular mitochondria while reducing those with fragmented mitochondria (Fig 1E), reaching proportions similar to those observed in control fibroblasts (13.4% ± 11.0 in CTL-1 vs. 18.5% ± 13.9 in *OPA1$^{S545R}$*). These data indicate that inhibiting fission can restore mitochondrial morphology in OPA1 mutant fibroblasts exhibiting mitochondrial fragmentation. In addition, depletion of *OPA1* by siRNA treatment in *OPA1$^{S545R}$* patient fibroblasts further increased mitochondrial fragmentation by 34.5% (1.34-fold change), implying partial functionality of OPA1 protein present in *OPA1$^{S545R}$* patient fibroblasts. Indeed, treatment of *OPA1$^{S545R}$* patient fibroblasts with CHX led to an elongation of the mitochondrial network (Fig 1F) characterized by reduced mitochondrial fragmentation (Fig 1G), indicating that *OPA1$^{S545R}$* cells are capable of performing SiMH and therefore retain some functional OPA1 (Tondera *et al*, 2009). These data lend experimental support to a previously proposed genetic haploinsufficiency in DOA (Pesch *et al*, 2001) caused by mono-allelic pathogenic variants. Taken together, these data outline a straightforward and unbiased manner to identify and correct mitochondrial fragmentation in patient-derived fibroblasts.

## High-throughput screening identifies known and novel modifiers of mitochondrial morphology in control fibroblasts

In an effort to identify mitochondrial proteins that regulate OPA1 dynamics, we established an imaging-based screening pipeline to quantitatively assess the impact of all mitochondrial genes on mitochondrial morphology. To do this, we coupled automated imaging and supervised ML mitochondrial morphology quantification workflow (Fig 1A) with a bespoke siRNA library targeting 1,531 known and putative nuclear-encoded mitochondrial genes (henceforth termed the *Mitome* siRNA library) generated based on publicly accessible databases of mitochondrial genes (Smith & Robinson, 2019; Rath *et al*, 2021) (see Dataset EV2 for gene list and plate distribution). This list is more extensive than MitoCarta 3.0 and also

**Table 1. Clinical features of patients from which fibroblasts were derived.**

| Patients (gender, age) | Age of onset | Optic atrophy | CPEO | Ataxia | Spasticity | Peripheral neuropathy | Deafness | OPA1 variant and effect on protein: variant 1 RefSeq NM_015560.2 | OPA1 variant and effect on protein: variant 8 RefSeq NM_130837.2 | OPA1 domain | Ref. |
|---|---|---|---|---|---|---|---|---|---|---|---|
| OPA1^S545R (M, 30 years) | Childhood | + | − | + | − | + | + | c.1635C>G p.(Ser545Arg) | c.1800C>G p.(Ser600Arg) | Dynamin | Patient FR-1 (Yu-Wai-Man et al, 2016) |
| OPA1^R445H (F, 37 years) | 6 years | + | − | + | − | − | + | c.1334G>A p.(Arg445His) | c.1499G>A p.(Arg500His) | GTPase | Patient 1 (Amati-Bonneau et al, 2005) |
| OPA1^c2356-1G>T (F, 60 years) | 50 years | + | − | + | + | − | − | c.2356-1G>T r.spl? | c.2521-1G>T r.spl | Dynamin | Patient A (Yu-Wai-Man et al, 2016) |
| OPA1^I432V (M, 43 years) | Childhood | + | + | + | + | − | − | c.1294A>G p.(Ile432Val) | c.1459A>G p.(Ile487Val) | GTPase | Patient UK-12 (Yu-Wai-Man et al, 2010) |
| OPA1^G297X (F, 48 years) | < 5 years | + | − | − | + | + | − | c.899C>T p.(Gln297*) | c.1054C>T p.(Gln352*) | GTPase | Patient UK-5 (Yu-Wai-Man et al, 2010) |

Mutational data are described using the nomenclature of the Human Genome Variation Society (http://www.hgvs.org/mutnomen). Nucleotide numbering reflects cDNA numbering with "+1" corresponding to the A of the ATG translation initiation codon. The initiation codon is codon 1.
CPEO, chronic progressive external ophthalmoplegia; F, female; M, male.

includes targets gene products whose function and localization have not yet been experimentally defined. *SmartPool* siRNAs (4 siRNAs per gene per pool) were spotted individually across six 384-well plates, which also contained siRNAs for *DNM1L*, *OPA1*, and *YME1L* that could serve as read-outs for downregulation efficiency within and between plates as well as ground truths for supervised ML (Fig EV2A–C, (Z-score = 0.72875 ± 0.1106). We began by *Mitome* screening in healthy control fibroblasts (CTL-1 and CTL-2) and identified 22 genes whose downregulation led to the fragmentation of the mitochondrial network and 145 genes that lead to hypertubulation above thresholds that were defined *post hoc* using a univariate 3-component statistical model we developed in R (Dataset EV3). Among the genes whose ablation induced mitochondrial fragmentation, we identified established components required for the maintenance of tubular mitochondria including *YME1L*, *OPA1*, and *MFN1* (Fig 2B, Dataset EV3). We also identified factors already described to modify mitochondrial morphology including *AMBRA1*, *GOLPH3*, and *PPTC7*. AMBRA1, which stands for activating molecule in Beclin-1-regulated autophagy, is an autophagy adapter protein regulated by mTORC1 that has been linked to mitophagy and programmed cell death, all of which are associated with fragmentation of the mitochondrial network. Golgi phosphoprotein 3 (GOLPH3) regulates Golgi morphology and mitochondrial mass and cardiolipin content through undefined mechanisms (Sechi *et al*, 2015). PPTC7 encodes a mitochondrial phosphatase shown to be essential for post-natal viability in mice. EM analyses in heart and liver sections of *Pptc7*^−/− mice revealed smaller, fragmented mitochondria (Niemi *et al*, 2019), consistent with our findings in human fibroblasts (Appendix Fig S2A).

Among the genes whose ablation induced mitochondrial hypertubulation (Fig 2C), we identified *DNM1L*, its receptors *MIEF1* and *MFF*, as well as *USP30* and *SLC25A46*. USP30 encodes a deubiquitinase that is anchored to the OMM where it contributes to mitochondrial fission in a DRP1-dependent fashion (Bingol *et al*, 2014). Depletion of USP30 has been shown to promote mitochondrial elongation and mitophagy (Nakamura & Hirose, 2008). SLC25A46, which encodes for an outer membrane protein with sequence homology to the yeast mitochondrial dynamics regulator Ugo1, is required for mitochondrial fission. In human fibroblasts, depletion by siRNA or pathogenic loss-of-function variants leads to hypertubulation of the mitochondrial network (Abrams *et al*, 2015; Janer *et al*, 2016). Similarly, depletion of MFF and/or MiD51 in fibroblasts inhibits DRP1-dependent mitochondrial fission and results in mitochondrial hypertubulation (Osellame *et al*, 2016). Pathogenic variants in *MFF* cause optic and peripheral neuropathy and fibroblasts from these patients exhibit mitochondrial elongation (Koch *et al*, 2016). In addition to known regulators of mitochondria morphology, we also discovered a number of known mitochondrial genes whose functions have not previously associated with mitochondrial dynamics, including *LIPT1*, *LIPT2*, and *BCKDHA*. LIPT1 and LIPT2 encode mitochondrial lipoyltransferases, which are involved in the activation of TCA cycle enzyme complexes and branched-chain ketoacid dehydrogenase (BCKD) complex. *BCKDHA* the E1-alpha subunit of the BCKD that is involved in the catabolism of amino acids isoleucine, leucine, and valine. Mutations in either *LIPT1* (Soreze *et al*, 2013, 1), *LIPT2* (Habarou *et al*, 2017, 2), or *BCKDHA* (Flaschker *et al*, 2007) causes inborn errors of metabolism, although the effects on mitochondrial morphology have never been investigated. Finally, we also discovered

a cluster of genes (Appendix Fig S2B) encoding proteins required for ribosome assembly and cytosolic translation (*RPL10, RPL10A, RPL8, RPL36AL, RPS18*). To our knowledge, depletion of cytosolic ribosomal genes has never been associated with mitochondrial hyperfusion, although chemical inhibition of proteins synthesis is the most commonly used trigger for SiMH (Tondera, 2005). These data are consistent with the mitochondrial elongation induced by treatment of control fibroblasts (Fig 1G and H) with CHX, which inhibits cytosolic translation. Altogether, our data demonstrate the robustness of our imaging-based phenotypic screening and mitochondrial morphology quantification approach for the identification of both known and novel genes controlling mitochondrial morphology and provide a valuable resource for the investigation of mitochondrial dynamics.

## High-throughput screening in patient-derived *OPA1* mutant fibroblasts identifies suppressors of mitochondrial fragmentation

We sought to apply the *Mitome* screening approach to identify novel regulators of OPA1 acting as genetic suppressors of mitochondrial fragmentation in *OPA1*$^{S545R}$ fibroblasts. After 72 h of siRNA treatment, we acquired images of hundreds of cells per well (257–1,606) and then classified mitochondrial morphology by applying a training sets comprised of *OPA1*$^{S545R}$ fibroblasts transfected with NT siRNAs (fragmented), *OPA1* siRNAs (hyperfragmented), or *DNM1L* siRNAs (rescued). Application of our imaging and quantification pipeline identified 91 candidate genes whose downregulation rescued mitochondrial fragmentation (Figs 2D and EV2C, Dataset EV4) as well as 27 genes that further fragmented the mitochondrial network (Fig EV2D and E, Dataset EV4) such as *OPA1, YME1L,* and *SURF1*. As expected, among the 91 candidates, 39 of these genes were also discovered to hypertubulate mitochondria in control fibroblasts upon downregulation (Fig 2C and F), including regulators of mitochondrial fission such as *SLC25A46* (Janer *et al,* 2016), *MFF* (Gandre-Babbe & van der Bliek, 2008), *MIEF1* (Osellame *et al,* 2016, 49), and *DNM1L* (Smirnova *et al,* 2001). We also discovered factors interacting with the MICOS complex (DNAJC4, DNAJC11), which was unexpected given that

disruption of the MICOS and respiratory chain complexes is usually associated with fragmentation rather than elongation of the mitochondrial network (Stephan *et al,* 2020). Nevertheless, validation studies revealed that depletion of DNAJC4 or 11 could rescue mitochondrial fragmentation caused by OPA1 deficiency (Fig EV2F and G). Like in control fibroblasts, our data revealed a cluster of ribosomal genes bioinformatically predicted to be targeted to mitochondria according to the Integrated Mitochondrial Protein Index (IMPI) score of the Mitominer 4.0 database including *RPL15, RPS15A, RPLP2, RPL36AL, RPL5,* and *RPS18*, essential for cytosolic translation, implying that inhibition of protein synthesis can suppress mitochondrial fragmentation in *OPA1*$^{S545R}$ patient fibroblasts. These data are concordant with the discovery that *OPA1*$^{S545R}$ patient fibroblasts can perform SiMH in the presence of the cytosolic protein inhibitor CHX (Fig 1F and G). The *Mitome* siRNA screen of *OPA1*$^{S545R}$ fibroblasts identified a wide array of well-characterized genes not previously linked to mitochondrial dynamics including some required for mitochondrial gene expression and maintenance (*TFB1M, MTERF4, MRPL53, GFM2, MRPS18A*), oxidative phosphorylation (*NDUFAF1, COX6A2, ETHE1, COX20, ETFDH*), amino acid metabolism (*BCKDHA, GLUD2, DAOA, MCCC1, GLYAT*), one-carbon and serine metabolism (*MMAA, SHMT2, MTHFD1L, MTHFD2L*), and lipid biosynthesis (*PGS1, PISD, BZRAP1*) as well as orphan genes (*C15orf62, C15orf61, C3orf33*; Fig 2E, Dataset EV4). In conclusion, we successfully applied an unbiased, high-throughput imaging approach and identified a large number of candidate suppressors of mitochondrial dysfunction in MD patient-derived fibroblasts, none of which are known to be implicated in the modulation of clinical or biochemical severity caused by OPA1 mutations.

## PGS1 depletion rescues mitochondrial fragmentation in OPA1-deficient fibroblasts

One of the top hits from the *Mitome* siRNA screen able to rescue aberrant mitochondrial morphology in *OPA1*$^{S545R}$ patient fibroblasts or promote mitochondrial hypertubulation in control fibroblasts,

**Figure 2. High-throughput screening identifies known and novel genetic modifiers of mitochondrial morphology in control and DOA+ patient-derived fibroblasts.**

A  Schematic of *Mitome* siRNA imaging screen for mitochondrial morphology in control human fibroblasts. Fibroblasts were reverse-transfected with siRNAs directed against 1,531 nuclear-encoded mitochondrial genes in 384-well plates and stained for mitochondria (anti-TOMM40, green), nuclei (DAPI, blue), and cytoplasm (CellMask, blue). Supervised ML training performed on control fibroblasts treated with siRNAs for *OPA1* or *YME1L* (fragmented) NT control (normal), and *DNM1L* (hypertubular) were applied to single-cell trinary classification of *Mitome* siRNA-treated fibroblasts. Passage number P14.

B  Candidate siRNAs (purple) causing mitochondrial fragmentation relative to grounds truths for fragmentation (*OPA1* siRNA). Violin plot representing % fragmented morphology of *Mitome* siRNAs (purple). Hits were selected with a univariate three-components statistical model programmed in R using ground truths (n = 30) for morphology shown in (A). The defined threshold for positive hits (thick dotted line inset) was 68.9% (solid dash on the y-axis and thin dotted line in the inset) and identified 22 candidate genes, including *OPA1, YME1L,* and *AMBRA1* from two independent experiments.

C  Candidate siRNAs (purple) causing mitochondrial hypertubulation relative to grounds truths for hypertubulation (*DNM1L* siRNA). Violin plot representing % hypertubulated morphology of *Mitome* siRNAs (purple). Hits were selected with a univariate 3-components statistical model programmed in R using ground truths (n = 30) for morphology shown in (A). The defined threshold for positive hits (thick dotted line inset) was 69.2% (solid dash on the y-axis and thin dotted line in the inset) and identified 145 candidate genes, including *DNM1L, MIEF1,* and *PGS1* from two independent experiments.

D  Schematic of *Mitome* siRNA imaging screen in *OPA1*$^{S545R}$ patient fibroblasts. Fibroblasts transfection and imaging as described in A. Supervised ML training performed on *OPA1*$^{S545R}$ fibroblasts treated with siRNA for *OPA1* (hyperfragmented) NT control (normal), and *DNM1L* (rescued) were applied to single-cell trinary classification of *OPA1*$^{S545R}$ patient fibroblasts. Passages number P12.

E  Violin plot representing % rescued morphology of *Mitome* siRNAs. The siRNA able to rescue mitochondrial fragmentation were selected with a univariate 3-components statistical model programmed in R using the following ground truths for morphology: fragmented (NT siRNA, n = 30), rescued (*DNM1L* siRNA, n = 30), and hyperfragmented (*OPA1* siRNA, n = 30). The defined threshold for positive rescued hits (thick dotted line inset) was 49.81% (solid dash on the y-axis and thin dotted line in the inset) and identified 91 candidate genes from one experiment.

F  Overlap between 91 candidates identified in (E) and (C) identify 38 overlapping genes leading to mitochondrial elongation (*hypertubulation* in CTL-1, CTL-2, and *rescued* in *OPA1*$^{S545R}$ fibroblasts) and 53 genes that specifically rescue mitochondrial fragmentation in *OPA1*$^{S545R}$ fibroblasts.

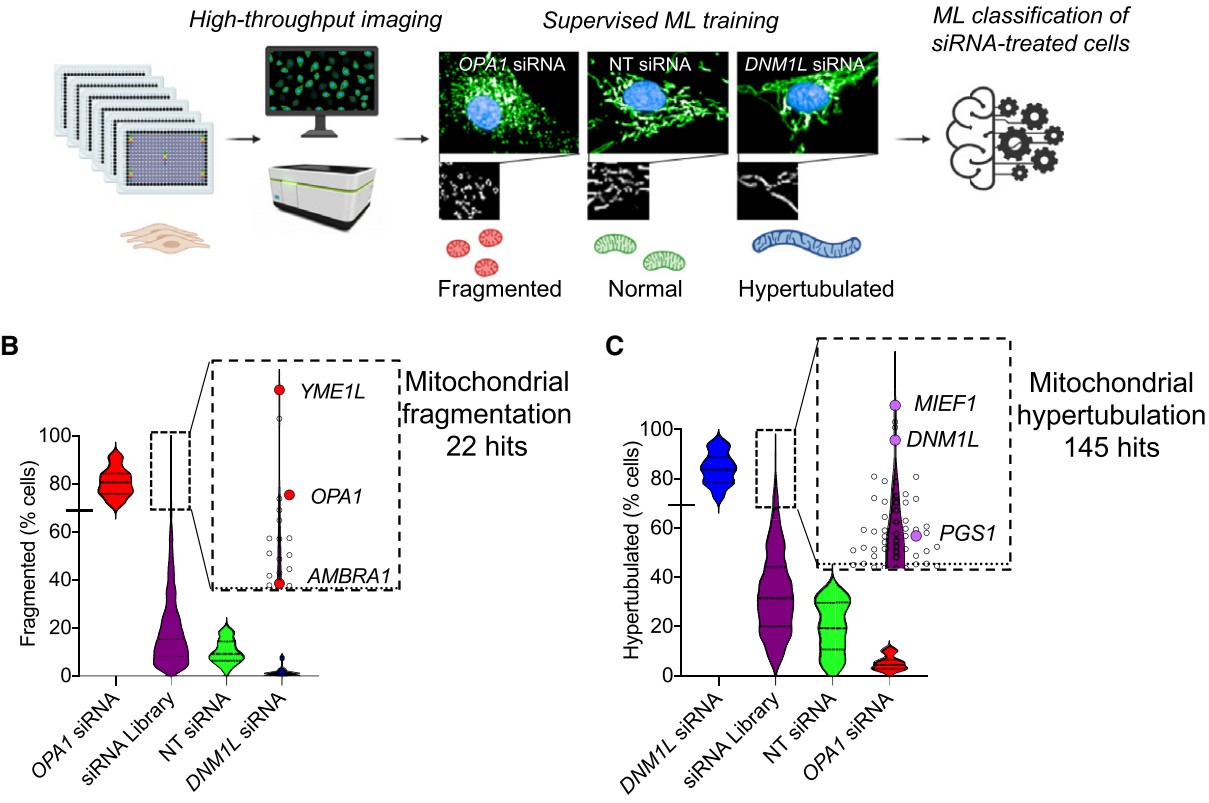

**A** *Mitome* siRNA screen – Control Fibroblasts

High-throughput imaging

Supervised ML training

ML classification of siRNA-treated cells

OPA1 siRNA    NT siRNA    DNM1L siRNA

Fragmented    Normal    Hypertubulated

**B** Mitochondrial fragmentation 22 hits

YME1L
OPA1
AMBRA1

**C** Mitochondrial hypertubulation 145 hits

MIEF1
DNM1L
PGS1

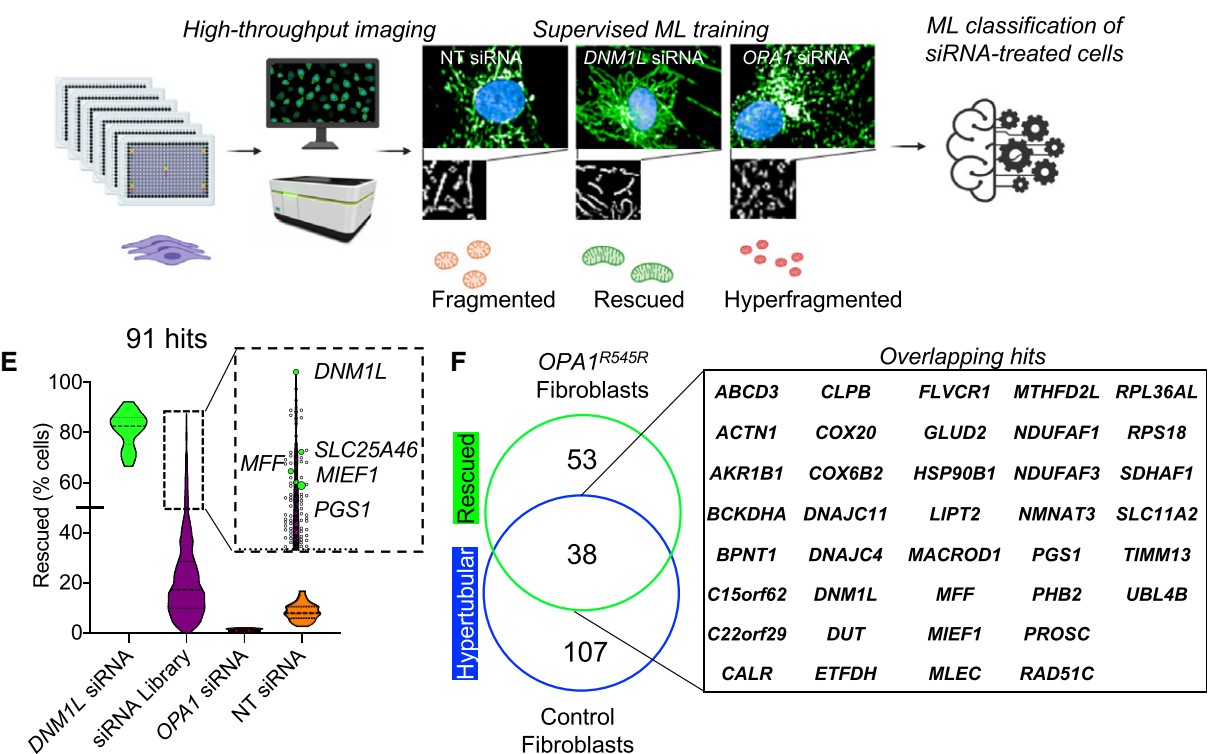

**D** *Mitome* siRNA screen – *OPA1^{S545R}* Fibroblasts

High-throughput imaging

Supervised ML training

ML classification of siRNA-treated cells

NT siRNA    DNM1L siRNA    OPA1 siRNA

Fragmented    Rescued    Hyperfragmented

**E** 91 hits

DNM1L
SLC25A46
MIEF1
MFF
PGS1

**F** *OPA1^{R545R}* Fibroblasts    Overlapping hits

Rescued

53

38

107

Hypertubular

Control Fibroblasts

| ABCD3 | CLPB | FLVCR1 | MTHFD2L | RPL36AL |
|---|---|---|---|---|
| ACTN1 | COX20 | GLUD2 | NDUFAF1 | RPS18 |
| AKR1B1 | COX6B2 | HSP90B1 | NDUFAF3 | SDHAF1 |
| BCKDHA | DNAJC11 | LIPT2 | NMNAT3 | SLC11A2 |
| BPNT1 | DNAJC4 | MACROD1 | PGS1 | TIMM13 |
| C15orf62 | DNM1L | MFF | PHB2 | UBL4B |
| C22orf29 | DUT | MIEF1 | PROSC | |
| CALR | ETFDH | MLEC | RAD51C | |

**Figure 2.**

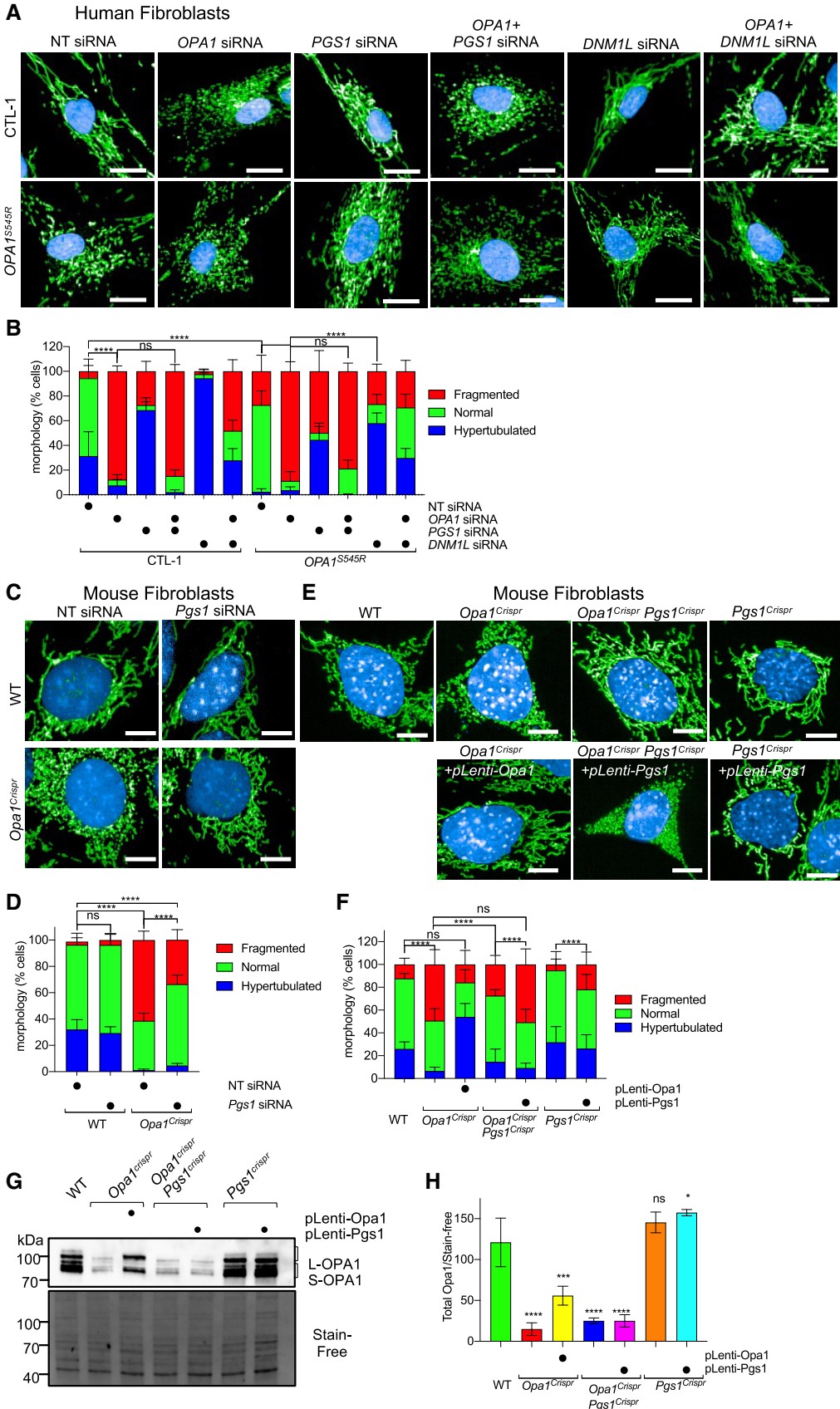

**Figure 3.**

**Figure 3. PGS1 depletion rescues mitochondrial fragmentation in OPA1-deficient human and mouse fibroblasts.**

A  Representative confocal images of control (CTL-1) and *OPA1^S545R^* patient fibroblasts treated with *OPA1*, *DNM1L*, *PGS1*, and non-targeting (NT) siRNAs or indicated combinations for 72 h. Mitochondria (anti-TOMM40, green) and nuclei (DAPI, blue). Scale bar = 20 μm. Passages number between P10–15.

B  Mitochondrial morphology quantification of (A) using control fibroblasts with fragmented (*OPA1* siRNA), normal (non-targeting NT siRNA), and hypertubulated (*DNM1L* siRNA) mitochondria. Data represent mean ± SD of three independent experiments, One-way ANOVA (905–3,695 cells per cell line), (% fragmented); ****$P < 0.0001$, ns; not significant.

C  Representative confocal images of wild-type (WT) and *Opa1^Crispr^* MEFs treated with NT or *Pgs1* siRNA for 72 h. Live imaging of mitochondria (mitoYFP, green) and nuclei (NucBlue, blue). Scale bar = 10 μm.

D  Mitochondrial morphology quantification of (C) using WT MEFs treated with *Opa1* siRNA (fragmented), NT siRNA (normal), or *Dnm1l* siRNA (hypertubulated) ground truth training sets. Data represent mean ± SD of three independent experiments, One-way ANOVA (6,613–8,758 cells per cell line), (% fragmented); ****$P < 0.0001$, ns; not significant.

E  Representative confocal images of WT, *Opa1^Crispr^* MEFs complemented with pLenti-Opa1, *Opa1^Crispr^Pgs1^Crispr^* MEFs, and *Pgs1^Crispr^* MEFs complemented with pLenti-Pgs1 by lentiviral delivery. Live imaging of mitochondria (mitoYFP, green) and nuclei (NucBlue, blue). Scale bar = 10 μm.

F  Supervised ML mitochondrial morphology quantification of (E) using WT MEFs treated with *Opa1* siRNA (fragmented), NT siRNA (normal), or *Dnm1l* siRNA (hypertubulated) training sets. Data represent mean ± SD of three independent experiments, One-way ANOVA (691–3,990 cells per cell line), (% fragmented); ****$P < 0.0001$, ns; not significant.

G, H  (G) Equal amounts of protein extracted from MEFs were separated by SDS–PAGE, immunoblotted with anti-OPA1 antibody, and quantified (H) by densitometry relative to Stain-Free. Data represent mean ± SD of three independent experiments, One-way ANOVA; *$P < 0.05$, ***$P < 0.001$, ****$P < 0.0001$, ns; not significant.

Source data are available online for this figure.

*PGS1*, encodes a CDP-diacylglycerol-glycerol-3-phosphate 3-phosphatidyltransferase (Chang *et al*, 1998) that catalyzes the synthesis of phosphatidylglycerol phosphate (PGP), the rate-limiting step in the synthesis of cardiolipin (CL; Fig 5A) (Tamura *et al*, 2020). CL is a mitochondria-specific phospholipid synthesized and primarily located in the IMM and is important for various mitochondrial functions including protein and metabolite import, cristae maintenance, programmed cell death regulation, and oxidative phosphorylation (Dudek, 2017). Recent work from the Ishihara laboratory reported CL to be important for membrane fusion by OPA1, implying that CL deficiency would impair mitochondrial fusion and drive fragmentation (Ban *et al*, 2017).

We sought to confirm that *PGS1* depletion indeed inhibits mitochondrial fragmentation by treating *OPA1^S545R^* fibroblasts with siRNAs directed against it. PGS1 depletion significantly reduced the proportion of cells with fragmented mitochondria, and we discovered it could only do so if OPA1 was not totally depleted (Fig 3A and B). *OPA1^S545R^* patient fibroblasts and *OPA1* siRNA-treated CTL-1 fibroblasts were resistant to mitochondrial elongation by *PGS1* depletion, although *DNM1L* ablation could still rescue mitochondrial fragmentation in these cells. These data argue that *PGS1* depletion is effective in rebalancing mitochondrial dynamics in the context of a hypomorphic OPA1 mutations (Del Dotto *et al*, 2018) and not when OPA1 is completely absent.

Functional exploration of mitochondrial biology in primary human fibroblasts is challenging due to the slow proliferation rates, low metabolic activity, poor transfection efficiency, genetic heterogeneity, and cellular senescence. To circumvent these limitations, we pursued further studies in mouse embryonic fibroblasts (MEFs) in which we partially (*Opa1^Crispr^*) or completely (*Opa1^KO^*) ablated OPA1 (Fig EV3A and B). To generate hypomorphic *Opa1* mutant MEFs (*Opa1^Crispr^*), we employed Crispr/Cas9 to initiate a targeted disruption of Exon 4, which is in the most highly expressed functional splice isoforms of the eight isoforms of *Opa1* in mice (Song *et al*, 2007; Akepati *et al*, 2008) (Fig EV3A). We sorted individual *Opa1^Crispr^* MEF clones by flow cytometry and screened for positive clones using mitochondrial fragmentation as an initial readout. DNA sequencing of *Opa1* in positive clones was performed by Illumina HighSeq Deep Sequencing of PCR amplicons covering the targeted

region. *Opa1^Crispr^* MEFs harbored a c.5013delA mutation, predicted to prematurely truncate OPA1 at position 178, and a 107 bp deletion at c.503 extending through the end of Exon 4 and into Intron 4, predicted to prematurely truncate OPA1 at position 182 in Exon 5. These deletions yielded frameshift and missense mutations causing a ~ 80% reduction in steady-state protein levels in *Opa1^Crispr^* MEFs (Fig 3G and H) and a ~ 50% reduction in *Opa1* mRNA levels (Fig EV3C), which could also be effectively achieved by siRNA-mediated downregulation (Fig EV3D). *Opa1^Crispr^* MEFs exhibited mitochondrial fragmentation (Fig 3C and D) that could be rescued by stable re-expression of OPA1 isoform 1 with (Fig EV3D–F) or without a C-terminal 9xMyc tag construct (Mishra *et al*, 2014) (Fig 3E and F), validating the targeted disruption of *Opa1*. Similarly to hypomorphic *OPA1^S545R^* patient-derived fibroblasts, *Opa1^Crispr^* MEFs exhibited hypomorphy, as evidenced by the ability of *Opa1* siRNA treatment to further increase mitochondrial fragmentation (Appendix Fig S4A and B) to levels observed in *Opa1^KO^* MEFs (Appendix Fig S4E and F) and the ability of *Opa1^Crispr^* MEFs to undergo SiMH (Appendix Fig S4C and D), which was not possible in *Opa1^KO^* MEFs (Appendix Fig S4E and F).

Next, we tested whether PGS1 depletion could rescue mitochondrial fragmentation in *Opa1^Crispr^* MEFs. PGS1 ablation, either by siRNA (Fig 3C and D) or Crispr/Cas9-mediated NHEJ (Fig 3E and F) prevented mitochondrial fragmentation, leading to the re-establishment of wild-type mitochondrial network morphology. qRT–PCR measurement of *Pgs1* mRNA levels showed a 25 ± 8.3% reduction in Pgs1 mRNA in *Opa1^Crispr^Pgs1^Crispr^* MEFs (Fig EV3C) and a 71.9 ± 8.4% percent reduction in *Pgs1* siRNA-treated *Opa1^Crispr^* MEFs (Fig 5D). To confirm that mitochondrial morphology rescue in *Opa1^Crispr^Pgs1^Crispr^* MEFs did not arise from unlikely and unintended reversions of mutant *Opa1*, we performed DNA sequence analyses by Illumina HighSeq Deep Sequencing of *Opa1^Crispr^Pgs1^Crispr^* MEF PCR amplicons from the targeted locus. *Opa1^Crispr^Pgs1^Crispr^* MEFs carried the same *Opa1* loss-of-function mutations as the parental *Opa1^Crispr^* MEFs as well as an additional mutation in *Pgs1* (c.218delGTGTA), predicted to result in a frameshift at Gly73. Stable re-expression of PGS1 restored *Pgs1* mRNA levels in *Pgs1^Crispr^* MEFs (Fig EV3C) and resulted in fragmentation of the (rescued) mitochondrial network in *Opa1^Crispr^Pgs1^Crispr^* MEFs

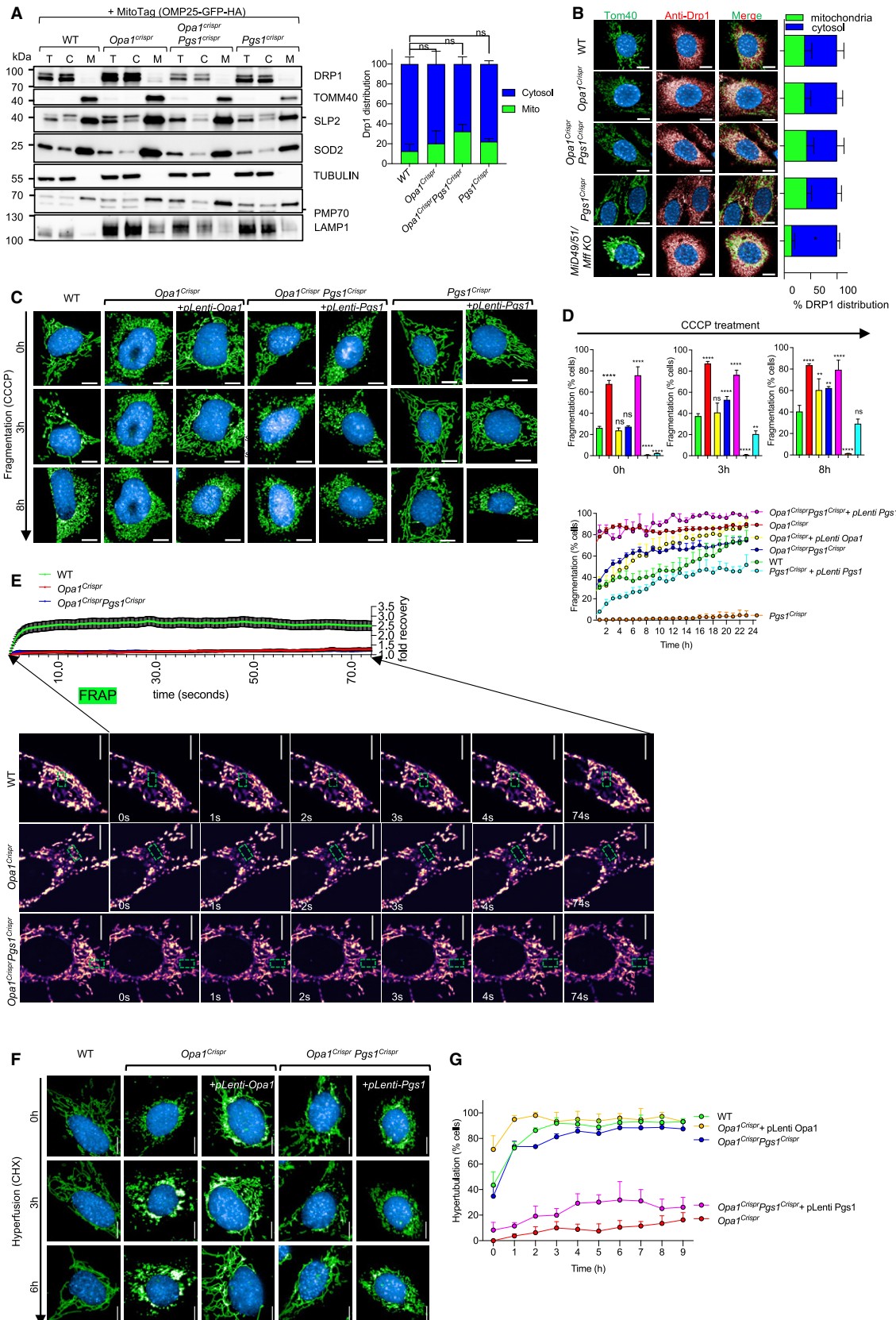

**Figure 4.**

**Figure 4. PGS1 depletion rescues mitochondrial fragmentation by inhibiting mitochondrial fission.**

A   Equal amounts of protein extracted from total (T), cytosolic flow-through (C), and mitochondrial eluate (M) from MEFs of the indicated genotypes stably expressing MitoTag (pMXs-3XHA-EGFP-OMP25) obtained following mitochondrial immunocapture were separated by SDS–PAGE, immunoblotted with indicated antibody, and quantified by densitometry. Data represent mean ± SD of three independent experiments, One-way ANOVA.

B   Representative confocal images of MEFs of the indicated genotypes showing subcellular DRP1 distribution. Mitochondria (TOMM40, green), DRP1 labeled with anti-DRP1 antibody (red) and nuclei (NucBlue, blue). Scale bar = 10 μm. *MiD49/51/Mff* KO MEFs lack all 3 DRP1 receptors (MiD49, MiD51, and MFF). Bar graph representation of DRP1 localized to mitochondria (green) vs cytosol (blue). Data represent mean ± SD of three independent experiments (884–3,116 cells per cell line), unpaired *t*-test; *P < 0.05.

C   Representative confocal images of live cell imaging of MEFs of the indicated genotypes subjected fragmentation with 5 μM carbonyl cyanide m-chlorophenyl hydrazine (CCCP) for the indicated time points. Images were captured every hour for 18 h. Scale bar = 10 μm.

D   Supervised ML mitochondrial morphology quantification using WT MEFs treated with 5 μM CCCP for 18 h (fragmented), untreated (normal), or treated with 10 μM CHX for 9 h (hypertubular) training sets. Data represent mean ± SD of three independent experiments (131–426 cells per cell line), One-way ANOVA; *P < 0.05, **P < 0.01, ****P < 0.0001, ns; not significant.

E   FRAP fusion assay in MEFs of the indicated genotype (see Movies EV1–EV3). Scale bar = 10 μm. Quantification of mitoYFP signal intensity measured at 200 ms intervals in the photobleached area (green box) for the indicated time (seconds), represented as relative fold recovery post-bleach. Data represent mean ± SEM of two independent experiments (n = 18–52 cells per genotype), One-way ANOVA.

F   Representative confocal images of live cell imaging of MEFs of the indicated genotypes subjected hyperfusion (SiMH) with 10 μM cycloheximide (CHX) for the indicated time points. Images were captured every hour for 9 h.

G   Mitochondrial morphology quantification of using WT MEFs treated with 5 μM CCCP for 18 h (fragmented), untreated (normal), or treated with 10 μM CHX for 9 h (hypertubular) training sets. Data represent mean ± SD of four independent experiments, (155–745 cells per cell line), One-way ANOVA.

Source data are available online for this figure.

(Fig 3E and F) back to WT levels. To exclude the possibility that PGS1 depletion rescues mitochondrial morphology of *Opa1^Crisprr^* MEFs by indirectly elevating OPA1 expression, we assessed OPA1 protein levels by Western blot. *Opa1^Crispr^Pgs1^Crispr^* MEFs exhibited levels of total OPA1 levels and L-OPA1/S-OPA1 ratios (Fig 3G and H) similar to the parental *Opa1^Crispr^* cells, indicating that restored mitochondrial morphology in *Opa1^Crispr^Pgs1^Crispr^* MEFs is not the result of rescued OPA1 expression. Taken together, our results demonstrate that PGS1 depletion can rescue mitochondrial fragmentation caused by OPA1 deficiency in both mouse and human fibroblasts.

## PGS1 depletion rescues mitochondrial fragmentation by inhibiting mitochondrial fission

We sought to understand whether PGS1 depletion restores normal mitochondrial morphology by increasing mitochondrial fusion or reducing mitochondrial fission. We examined the levels of proteins involved in mitochondrial dynamics by Western blot (Fig EV4A and B), and we observed no significant alterations in the steady-state levels of known fusion (MFN1, MFN2) and fission (MFF, MiD49, MiD51, FIS1) regulators in *Opa1^Crispr^Pgs1^Crispr^* MEFs, yet we did observe elevated DRP1 levels in *Opa1^Crispr^* MEFs, which returned to WT levels in *Opa1^Crispr^Pgs1^Crispr^* MEFs. To test whether increased levels of DRP1 promoted its recruitment to mitochondria, we stably expressed mitoTag constructs (3XHA-EGFP-OMP25 or 3XMyc-EGFP-OMP25) in MEFs in order to perform affinity purification and partitioning of mitochondria from cytosolic contents (Chen *et al*, 2016). Immunoblot analyses demonstrated an increase in total DRP1 levels in *Opa1^Crispr mitoTag^* MEFs compared with other genotypes but did not show an increase in the partitioning of mitochondrial and non-mitochondrial (cytosolic) DRP1 at steady state (Fig 4A). We further corroborated these findings by examining the subcellular distribution of DRP1 by indirect immunochemistry studies, which also revealed no differences in DRP1 colocalization *Opa1^Crispr^*, *Opa1^Crispr^Pgs1^Crispr^*, and *Pgs1^Crispr^* MEFs relative to WT (Fig 4B). MEFs deleted of all three essential DRP1 receptors, *Mid51/Mid49/Mff*, exhibited markedly less DRP1 recruitment as previously

demonstrated and were used as a negative control (Osellame *et al*, 2016). Live confocal imaging of endogenously, fluorescently tagged DRP1 in WT and *Opa1^Crispr^* MEF (WT^Drp1KI^ and *Opa1^Crispr-Drp1KI^* MEFs, respectively) showed no differences in the degree of subcellular distribution of DRP1 in the presence or absence of PGS1 (Fig EV4D and E). We also assessed the phosphorylation status of DRP1 by immunoblot analysis using site-specific antibodies for mouse serine 579 (S579, which is equivalent to S600 for human DRP1) and mouse serine 600 (S600, which is equivalent to S637 for human DRP1; Appendix Fig S5A and B). *Opa1^Crispr^* MEFs showed increased S579 phosphorylation of DRP1, consistent with the pro-fission role of this post-translational modification, which was lowered to wild-type levels in *Opa1^Crispr^Pgs1^Crispr^* MEFs. Consistent with these observations, DRP1 oligomerization assessed by chemical cross-linking (Karbowski *et al*, 2007; Prudent *et al*, 2015) did not reveal genotype-specific differences (Appendix Fig S5C). Altogether, these data indicate the altered DRP1 recruitment to mitochondria does not explain the restoration of mitochondrial morphology caused by the depletion of PGS1 in *Opa1^Crispr^* MEFs.

To assess mitochondrial division in living *Opa1^Crispr^Pgs1^Crispr^* MEFs, we performed quantitative kinetic measurements of mitochondrial morphology in the presence of established pharmacological inducers of mitochondrial fragmentation: CCCP and the $Ca^{2+}$ ionophore 4Br-A23187. Both chemicals cause DRP1-dependent mitochondrial fragmentation but CCCP triggers OMA1-dependent OPA1 processing (MacVicar & Langer, 2016) that both accelerates fission and inhibits fusion while 4Br-A23187 treatment induces $Ca^{2+}$-dependent fragmentation without stress-induced OPA1 processing (Adachi *et al*, 2016) (Fig EV4C). Treatment of *Opa1^Crispr^Pgs1^Crispr^* MEFs with CCCP (Fig 4C) or 4Br-A23187 (Appendix Fig S6A and B) induced a progressive fragmentation of the mitochondrial network over several hours with kinetics similar to that of WT MEFs, implying that rescued mitochondrial morphology conferred to *Opa1^Crispr^* MEFs depleted (Fig 3C) or deleted (Fig 3E) was not caused by an inhibition of DRP1. We discovered PGS1-depleted MEFs to be largely resistant to CCCP-induced fragmentation for the duration of the experiment: incubation with 5 μM CCCP for 10 h led to a 1.70 rate of fragmentation in WT MEFs and only 0.06 rate of

fragmentation in $Pgs1^{Crispr}$ MEFs (Fig 4C and D). Similarly, induction of mitochondrial fission with 4Br-A23187 did not promote mitochondrial fragmentation rates observed in WT MEFs (Appendix Fig S6A and B). Given the resistance to uncoupler-induced mitochondrial fragmentation, we determined the mitochondrial membrane potential of $Pgs1^{Crispr}$ MEFs by labeling MEFs with the potentiometric membrane marker TMRE, which we normalized to genetically encoded mitoYFP. We observed a significant increase in membrane potential in $Pgs1^{Crispr}$ MEFs (Appendix Fig S6C), which was reduced upon stable re-expression of Pgs1, which also re-sensitized cells to CCCP (Fig 4C and D) and 4Br-A23187-induced fragmentation (Appendix Fig S6A and B). Despite the increase in basal membrane potential, we observed no difference in the proclivity of $Pgs1^{Crispr}$ MEFs to undergo proteolytic cleavage of OPA1 in response to CCCP-induced OMA1 activation (Appendix Fig S6D), indicating that the proteolytic activity of OMA1 is functional in PGS1-depleted cells. Taken together, we conclude that PGS1 depletion can inhibit mitochondrial fragmentation by slowing mitochondrial fission in a manner that is independent of OPA1 processing by OMA1.

### PGS1 depletion improves SiMH without restoring basal fusion to OPA1-deficient cells

To test whether PGS1 depletion also affected mitochondrial fusion in $Opa1^{Crispr}$ MEFs, we assessed IMM fusion kinetics using a fluorescence recovery after photobleaching (FRAP) assay (Mitra & Lippincott-Schwartz, 2010). Genetically encoded matrix-localized YFP (mitoYFP) was photobleached in a subsection of mitochondria and imaged 200 ms intervals (Fig 4F). In WT MEFs, mitoYFP single increased ~ 2.5-fold in the photobleached region of the network within a few seconds, demonstrating active mitochondrial fusion in these cells. As expected, FRAP experiments performed under the same conditions in $Opa1^{Crispr}$ MEFs revealed no significant recovery of mitoYFP signal, indicating a block in mitochondrial fusion, which was not improved upon additional deletion of $Pgs1$ (in $Opa1^{Crispr}Pgs1^{Crispr}$ MEFs) despite the appearance of a normal, tubular network in these cells (Movies EV1–EV3). These results indicate PGS1

depletion does not restore basal mitochondrial fusion function to $Opa1^{Crispr}$ MEFs.

Next, we sought to determine $Opa1^{Crispr}Pgs1^{Crispr}$ cells could undergo mitochondrial elongation induced by SiMH, despite an inhibition of IMM fusion. Live imaging of cells stimulated with CHX (Fig 4F and G) or the transcriptional inhibitor Actinomycin D (ActD) (Appendix Fig S6E and F) induced progressive mitochondrial hypertubulation in both WT and $Opa1^{Crispr}Pgs1^{Crispr}$ MEFs, implying normal hyperfusion capacity. These responses could be blunted in $Opa1^{Crispr}Pgs1^{Crispr}$ MEFs by re-expression of PGS1 (Fig 4F and G, Appendix Fig S6E and F), indicating that PGS1 activity inhibits SiMH in OPA1-deficient cells. In $Pgs1^{Crispr}$ cells, we observed a more rapid hypertubulation in response to SiMH than in WT MEFs (Appendix Fig S6E and F). In hypomorphic $Opa1^{Crispr}$ MEFs, we also observed a very modest but significant SiMH response, characterized by mitochondrial aggregation in $Opa1^{Crispr}$ MEFs in the presence of CHX (Fig 4F and G) or ActD (Appendix Fig S6E and F) and stable re-expression of OPA1 fully rescued mitochondrial morphology and SiMH response. MEFs devoid of any detectable OPA1 protein were unable to perform SiMH (Appendix Fig S4E and F) consistent with previous reports (Tondera et al, 2009). Notably, PGS1 depletion also failed to restore normal mitochondrial morphology in $Opa1^{KO}$ MEFs (Fig EV5A and B) or $Yme1l^{KO}$ MEFs (Fig EV5C–E), implying that the functional suppression of mitochondrial fragmentation by PGS1 depletion depends on the functional severity. Thus, we conclude that PGS1 depletion can re-establish SiMH response to $Opa1^{Crispr}$ MEFs without improving mitochondrial fusion under basal condition. Altogether, our data demonstrate that PGS1 depletion inhibits mitochondrial fragmentation in hypomorphic OPA1 mutant fibroblasts by inhibiting mitochondrial fission and not be increasing mitochondrial fusion.

### Downregulation of cardiolipin synthesis pathway enzymes can prevent mitochondrial fragmentation in OPA1-deficient cells

PGS1 synthetizes PGP from CDP-diacylglycerol (CDP-DAG) and glycerol 3-phosphate (G3P) (Chang et al, 1998) (Fig 5A). PGP is

---

**Figure 5.  Interfering with the cardiolipin synthesis pathway can prevent mitochondrial fragmentation in OPA1-deficient fibroblasts.**

A  Schematic of cardiolipin (CL) biosynthesis pathway in mitochondria. Phosphatidic acid (PA) is transported to the inner membrane by PRELID1 where it is converted to CDP-diacylglycerol (CDP-DAG) and glycerol 3-phosphate (G3P) by TAMM41. Phosphatidylglycerol phosphate (PGP) is dephosphorylated to phosphatidylglycerol (PG) by PTPMT1. PG is either degraded to DAG or reacts with CDP-DAG to form CL in a reaction catalyzed by cardiolipin synthase (CLS1). Tafazzin (TAZ) catalyzes the remodeling of monolysocardiolipin (MLCL) to mature CL. CL is transported to the outer membrane and converted to PA by mitoPLD. PA is converted to DAG by LIPIN1. PA can be supplied to the inner membrane from DAG conversion by Acylglycerol Kinase (AGK).

B  Representative confocal micrographs of MEFs WT and $Opa1^{Crispr}$ MEFs treated with indicated siRNAs for 72 h. Mitochondria (anti-TOMM40, green) and nuclei (DAPI, blue). Scale bar = 10 µm.

C  Supervised ML mitochondrial morphology quantification of (B) using WT MEFs with fragmented ($Opa1$ siRNA), normal (non-targeting NT siRNA), and hypertubular ($Dnm1l$ siRNA) mitochondria. Data represent mean ± SD of three independent experiments, One-way ANOVA (726–4,236 cells per cell line), (% fragmented); ***$P < 0.001$, ****$P < 0.0001$, ns; not significant.

D  Quantitative RT–PCR (qRT-PCR) measurement of $Prelid1$, $Tamm41$, $Pgs1$, $Ptpmt1$, and $Cls1$ expression in $Opa1^{Crispr}$ and WT MEFs. Fold change is indicated relative to WT control. Data represent mean ± SD of three independent experiments, One-way ANOVA.

E  Whole cell phospholipidome of WT and $Opa1^{Crispr}$ MEFs treated with NT (non-targeting), $Tamm41$ or $Pgs1$ siRNAs. Data represent mean ± SD of five independent experiments; *$P < 0.05$, ***$P < 0.001$, ****$P < 0.0001$, ns; not significant.

F  Representative confocal micrographs of MEFs WT, $Pgs1^{Crispr}$, and $Dnm1l^{Crispr}$ MEFs treated with indicated siRNAs for 72 h. Mitochondria (anti-TOMM40, green) and nuclei (DAPI, blue). Scale bar = 10 µm.

G  Supervised ML mitochondrial morphology quantification of (G) using WT MEFs with fragmented ($Opa1$ siRNA), normal (non-targeting NT siRNA), and hypertubulated ($Dnm1l$ siRNA) mitochondria. Data represent mean ± SD of >3 independent experiments (3,096–7,238 cells per cell line), One-way ANOVA (% fragmented); *$P < 0.05$, **$P < 0.01$, ****$P < 0.0001$, ns; not significant.

Source data are available online for this figure.

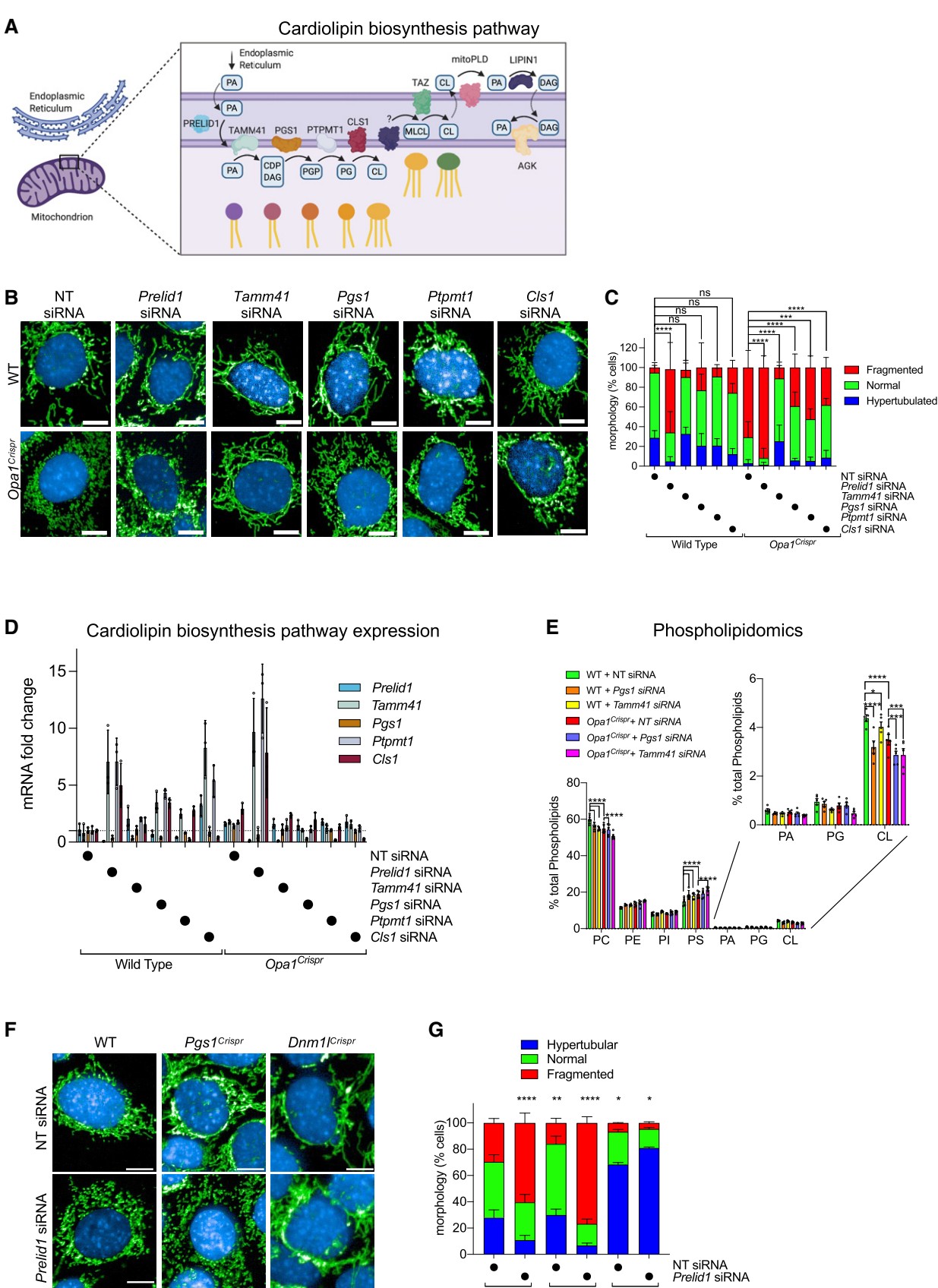

**Figure 5.**

dephosphorylated to phosphatidylglycerol (PG) by PTPMT1 (Zhang *et al*, 2011), which is either degraded to DAG or reacts with CDP-DAG to form CL in a reaction catalyzed by cardiolipin synthase, encoded by *Cls1* (Chen *et al*, 2006). Export of mature CL to the OMM is subsequently converted by mitoPLD to phosphatidic acid (PA), which inhibits fission by reducing DRP1 recruitment. PA can also be converted to DAG by LIPIN1B to promote DRP1 recruitment and mitochondrial fragmentation (Choi *et al*, 2006; Huang *et al*, 2011; Watanabe *et al*, 2011; Adachi *et al*, 2016). Since we observed no alterations in DRP1 recruitment in PGS1-depleted cells (Fig 4A and B) and PGS1 itself is an IMM enzyme, we decided to test whether interfering with CL biosynthesis enzymes localized in the IMM (Fig 5A) could reverse mitochondrial fragmentation of OPA1-deficient fibroblasts. We performed a series of knockdown experiments in WT and *Opa1$^{Crispr}$* MEFs using siRNAs targeting genes encoding enzymes both upstream (*Prelid1*, *Tamm41*) and downstream (*Ptpmt1*, *Cls1*) of *Pgs1* and analyzed mitochondrial morphology after 72 h (Fig 5B). Like the downregulation of *Pgs1*, we discovered that acute, single depletion of *Tamm41*, *Ptpmt1*, or *Cls1* could prevent mitochondrial fragmentation in *Opa1$^{Crispr}$* MEFs (Fig 5B and C). *Opa1$^{KO}$* MEFs did not respond to *Pgs1* or *Tamm41* depletion: Mitochondrial morphology still remains fragmented upon siRNA treatment (Fig EV5A and B). *Prelid1* depletion lead to increased mitochondrial fragmentation in both *Opa1$^{Crispr}$* and WT MEFs, confirming previous observations in HeLa cells (Potting *et al*, 2013). qRT–PCR analyses revealed significant transcriptional remodeling of CL enzymes in *Opa1$^{Crispr}$* and WT MEFs (Fig 5D). *Opa1$^{Crispr}$* MEFs showed an upregulation of *Prelid1*, *Tamm41*, *Pgs1*, *Ptpmt1*, and, to a greater extent, *Cls1*. *Prelid1* depletion led to an upregulation of *Tamm41*, *Ptpmt1*, and *Cls1* concomitant with a reduction in *Pgs1* mRNA levels in both *Opa1$^{Crispr}$* and WT MEFs. *Tamm41* depletion had more modest effects on the upregulation of *Prelid1* and *Ptpmt1*. Of note, *Pgs1* depletion led to threefold to fivefold increases in the levels of *Tamm41*, *Ptpmt1*, and *Cls1* in WT MEFs but not in *Opa1$^{Crispr}$* MEFs. Similarly, *Cls1* depletion led to similarly large increases the levels of *Prelid1*, *Tamm41*, and *Ptpmt1* mRNA in WT MEFs but not in *Opa1$^{Crispr}$* MEFs (Fig 5D), suggesting that there may be underlying defects in CL responses in *Opa1$^{Crispr}$* MEFs.

## Depletion of either OPA1 or PGS1 reduces cardiolipin levels

We sought to determine the impact of OPA1 and PGS1 depletion on the levels of CL. Quantitative phospholipidomic analyses of *Opa1$^{Crispr}$* MEFS revealed a reduction in CL content to $70.1 \pm 11.0\%$ of WT levels (Fig 5E). In addition, CL acyl chain composition analyses showed an increase in double bonds (Appendix Fig S7A) and altered acyl chain lengths (Appendix Fig S7B). Depletion via siRNA treatment of WT MEFs for *Pgs1* or, to a lesser degree, *Tamm41* (Fig 5E) reduced the steady-state levels of CL to levels similar to those of *Opa1$^{Crispr}$* MEFs. Depletion of either *Pgs1* or *Tamm41* in *Opa1$^{Crispr}$* MEFs lead to a further depletion of CL levels but not further alteration in acyl chain composition of CL. Overall, we found no correlation between the levels or saturation state of CL and mitochondrial morphology, prompting us to consider the possibility that suppression of PGS1 or TAMM41 in *Opa1$^{Crispr}$* MEFs restores mitochondrial morphology not via a reduction in CL production but rather through the accumulation of its precursor(s). The CL precursor common to cells depleted of *Tamm41*, *Pgs1*, and *Cls1* is PA,

which is first synthesized in the ER and shuttled from the OMM to the IMM by the lipid transfer protein PRELID1 (Potting *et al*, 2013). Suppression of PA delivery to the IMM via PRELID1 ablation causes mitochondrial fragmentation. PA accumulation in the IMM affects mitochondrial structure in yeast (Connerth *et al*, 2012), but its role in mammalian mitochondria has not been defined. To test whether local accumulation of PA in the IMM is responsible for the anti-fragmentation effect of PGS1 depletion on mitochondrial morphology, we pursued a genetic approach since lipid analyses of whole mitochondria cannot be used to define the submitochondrial localization of PA. We depleted *Prelid1* in WT and *Pgs1$^{Crispr}$* MEFs and assessed mitochondrial morphology after 72 h (Fig 5F). *Prelid1* depletion was able to fragment mitochondria in PGS1-deficient cells, arguing that the IMM accumulation of PA resulting from a block in the biosynthesis of CL (via PGS1 depletion) impedes mitochondrial fission (Fig 5G). PRELID1 depletion did not fragment mitochondria in DRP1-deficient (*Dnm1l$^{Crispr}$*) MEFs, demonstrating that PA depletion at the IMM promotes mitochondrial fragmentation in a DRP1-dependent fashion, perhaps by increasing the accumulation of PA at the OMM (Adachi *et al*, 2016). Taken together, these data argue that IMM accumulation of the CL precursor PA but not CL itself is responsible for the inhibition of mitochondrial fragmentation in a DRP1-dependent manner.

## PGS1 depletion does not alter apoptotic sensitivity nor cristae dysmorphology caused by OPA1 depletion

OPA1 regulates cristae morphology and apoptosis in cultured cells (Giacomello *et al*, 2020). To test whether restoration of mitochondrial morphology in *Opa1$^{Crispr}$* *Pgs1$^{Crispr}$* MEFs affects programmed cell death, we stimulated MEFs with apoptosis-inducing compounds and followed the evolution of cell death by live cell imaging (Figs 6A and B, and EV6A, Appendix Fig S8A and B). We kinetically imaged thousands of cells (2,000–12,000) every hour over 24 h and tracked NucBlue and propidium iodide (PI) as markers of total and dead cells, respectively. In the presence of ABT-737 and Actinomycin D (ActD) cell death was triggered more rapidly in *Opa1$^{Crispr}$* cells compared with WT, which could be inhibited by the pan-caspase inhibitor qVD. *Opa1$^{Crispr}$* *Pgs1$^{Crispr}$* MEFs exhibited cell death profiles indistinguishable from *Opa1$^{Crispr}$* MEFs, indicating that rescued mitochondrial morphology does not protect against apoptotic sensitivity caused by OPA1 depletion. In the presence of staurosporine (Fig EV6A and B) or etoposide (Appendix Fig S8A and B), *Opa1$^{Crispr}$* *Pgs1$^{Crispr}$* cell death sensitivity also did not return to WT levels. *Opa1$^{Crispr}$* cells exhibited reduced caspase-dependent cell death in the presence of staurosporine or etoposide, confirming previous observations of the stimuli-dependent apoptotic outcomes of haploinsufficient OPA1-deficient MEFs (Kushnareva *et al*, 2016). Notably, *Pgs1$^{Crispr}$* cells exhibited increased apoptotic resistance relative to WT cells when challenged with staurosporine, etoposide, or ABT-737 and ActD.

To assess the impacts on mitochondrial ultrastructure, we performed transmission electron microscopy on WT, *Opa1$^{Crispr}$* *Pgs1$^{Crispr}$*, *Opa1$^{Crispr}$* and *Pgs1$^{Crispr}$* MEFs. WT cells exhibited IMMs organized as lamellar cristae, which were disrupted as expected in *Opa1$^{Crispr}$* cells, which also had more rounded mitochondria consistent with the fragmented network morphology previously described (Fig 6C and D). However, IMM structure in *Opa1$^{Crispr}$* *Pgs1$^{Crispr}$* was

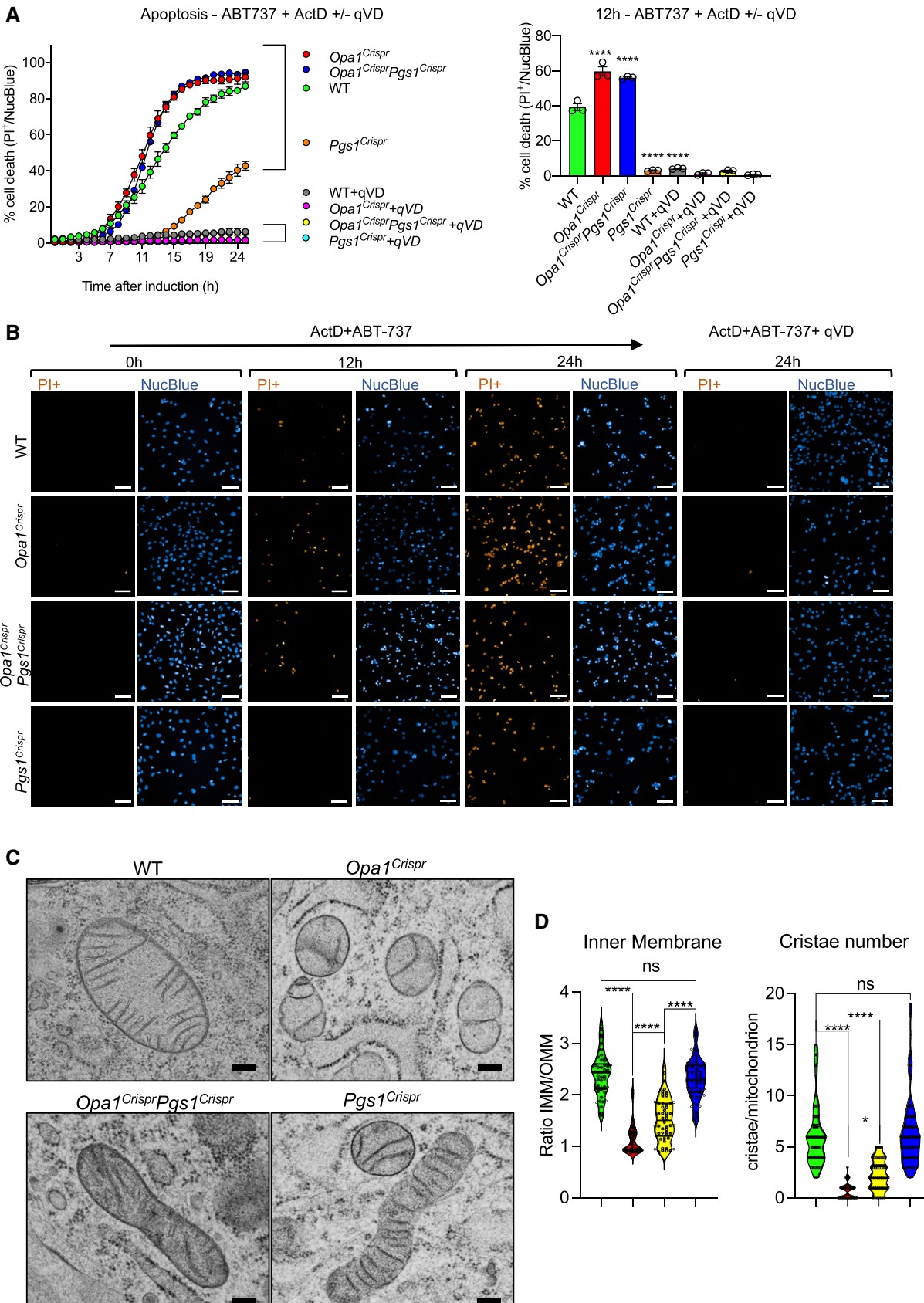

Figure 6.

**Figure 6. PGS1 depletion does not rescue apoptotic sensitivity nor cristae structure in OPA1-deficient MEFs.**

A, B  (A) MEFs of the indicated genotypes were subjected to 4 µM Actinomycin D and 10 µM ABT-737 in the presence or absence of the pan-caspase inhibitor qVD. Dead cells (PI$^+$ nuclei, orange) and total cells (NucBlue, blue) were imaged every hour for 25 h. PI$^+$ nuclei number divided by the total nuclei number was then quantified over time. (B) Representative confocal images of (A). Scale bar = 100 µm. Data represent mean ± SD of three independent experiments (1,380–2,157 cells per cell line), One-way ANOVA; ****$P < 0.0001$, ns; not significant.

C  Representative transmission electron micrographs of MEFs of the indicated genotypes showing loss of lamellar cristae in $Opa1^{Crispr}$ and $Opa1^{Crispr}Pgs1^{Crispr}$ MEFs. Scale bar = 200 nm.

D  Quantification of (C) of mitochondrial ultrastructure; outer membrane/inner membrane ration (IMM/OMM) and cristae number per mitochondrion. Violin plot of > 50 mitochondria per cell line, One-way ANOVA; *$P < 0.05$, ****$P < 0.0001$, ns; not significant.

Source data are available online for this figure.

not restored to WT morphology, despite a modest increase in cristae number and OMM/IMM ratios, indicating that mitochondrial morphology and cristae organization are largely uncoupled in these cells. We did not detect cristae defects in $Pgs1^{Crispr}$ cells, implying that CL reduction per se (Fig 5E) does cause defective mitochondrial ultrastructure in mammalian cells. Taken together, our data demonstrate that the role of OPA1 in balancing mitochondrial dynamics can be uncoupled from its role as an organizer of IMM structure and programmed cell death.

**Rebalancing mitochondrial dynamics OPA1-deficient fibroblasts through PGS1 improves bioenergetics but not mtDNA depletion**

In order to analyze the functional impact of re-establishing a tubular network on respiration and the oxidative phosphorylation (OXPHOS) system, we measured oxygen consumption rates using Seahorse FluxAnalyzer oxygraphy in intact MEFs depleted of OPA1 and/or PGS1 (Fig 7A). $Opa1^{Crispr}$ MEFs exhibited a modest reduction in basal (Fig 7B) and maximal oxygen consumption rates (Fig 7C) which could be improved upon deletion of PGS1 in $Opa1^{Crispr}Pgs1^{Crispr}$ MEFs, implying that rebalancing mitochondrial dynamics positively impacts mitochondrial respiration. Oxygen consumption rate (OCR) measurements performed using Seahorse FluxAnalyzer requires that plated adherent cells be submitted to a brief period of nutrient (glucose, $CO_2$) deprivation, which has previously been shown to induce mitochondrial hyperfusion (Gomes et al, 2011; Khacho et al, 2014). To exclude the possibility that nutrient starvation might confound bioenergetic measurements, we performed high-resolution respirometry (O2k, Oroboros) on intact, nutrient-replete MEFs in suspension (Fig EV7A). $Opa1^{Crispr}$ MEFs exhibited reduced oxygen consumption, which was rescued either by functional complementation with untagged OPA1 or depletion of PGS1. Interestingly, functional complementation of oxygen consumption and membrane potential defects present in $Opa1^{Crispr}$ MEFs was possible only with untagged OPA1 (Figs 7A–C, and EV7A) and not OPA1-Myc (Figs EV3G and H, and EV7B and C) even though both tagged and untagged OPA1 constructs were able to restore mitochondrial morphology (Figs 3E and F, and EV3E and F). These data further demonstrate that OPA1-dependent bioenergetic functions can be uncoupled from mitochondrial dynamics, in this case using a disruptive C-terminal epitope by the GED domain of the protein (Mishra et al, 2014).

Importantly, $Opa1^{Crispr}Pgs1^{Crispr}$ MEFs exhibited increased basal and maximal oxygen consumption rates relative to the parental $Opa1^{Crispr}$ MEFs, which could be lowered back to levels similar to $Opa1^{Crispr}$ MEFs by stable re-expression of PGS1 in $Opa1^{Crispr}Pgs1^{Crispr}$ MEFs. Pgs1-deficient cells exhibited increased respiration using both Seahorse and Oroboros oxygen consumption assays and were reduced upon re-expression of PGS1 (Figs 6A–C, and EV7A–C).

Next, we sought to determine the effects of restored mitochondrial morphology in $Opa1^{Crispr}Pgs1^{Crispr}$ MEFs on mitochondrial membrane potential. Cells were incubated with the potentiometric dye tetramethylrhodamine ethyl ester (TMRE) to label actively respiring mitochondria. TMRE signal intensity normalized to mitochondrial content (mitoYFP) and was recorded at the single-cell

**Figure 7. PGS1 depletion enhances respiration in wild-type and OPA1-deficient MEFs.**

A–C  (A) Mitochondrial respiration measured in adherent MEFs of the indicated genotypes using Seahorse FluxAnalyzer. Oxygen consumption rate (OCR) normalized to protein concentration. Following basal respiration, cells were treated sequentially with 1 µM Oligomycin (Omy), 2 µM CCCP, Antimycin A 1 µM + 1 µM Rotenone. Bar graphs of (A) representing basal (B) and maximum (C) respiration. Data represent mean ± SEM of 7–12 independent OCR measurements, One-way ANOVA; *$P < 0.05$, **$P < 0.01$, ***$P < 0.001$, ns; not significant.

D  Mitochondrial membrane potential measured by fluorescence microscopy in WT, $Opa1^{Crispr}$, $Opa1^{Crispr}$ + pLenti-Opa1, $Opa1^{Crispr}Pgs1^{Crispr}$, $Opa1^{Crispr}Pgs1^{Crispr}$ + pLenti-Pgs1, $Pgs1^{Crispr}$, and $Pgs1^{Crispr}$ MEFs + pLenti-Pgs1. Membrane potential is represented as the ratio between TMRE/mitoYFP. WT MEFs treated with 20 µM CCCP serve as a negative control for TMRE. Data represent mean ± SD of three independent experiments, number of analyzed cells indicated in inset, One-way ANOVA; **$P < 0.01$, ***$P < 0.001$, ****$P < 0.0001$, ns; not significant.

E  mtDNA content in MEFs from (F) was quantified by amplification of Mtll1, 16s, and Mt-nd1 genes relative to the Gapdh nuclear gene in MEFs. Data represent mean ± SD of three independent experiments, One-way ANOVA; ****$P < 0.0001$, ns; not significant.

F  mtDNA content in WT and mutant MEFs treated with indicated siRNAs for 72 h was quantified by amplification of Mttl1, 16s, and Mt-nd1 genes relative to the GapdhH nuclear gene in MEFs. Data represent mean ± SD of three independent experiments, One-way ANOVA; **$P < 0.01$, ****$P < 0.0001$, ns; not significant.

G, H  (G) Equal amounts of protein extracted from WT and mutant MEFs were separated by SDS–PAGE (horizontal line denotes separate membranes), immunoblotted with indicated antibodies, and quantified by densitometry (H). Data represent mean ± SD of three independent experiments, One-way ANOVA; **$P < 0.01$, ***$P < 0.001$, ****$P < 0.0001$, ns; not significant.

Source data are available online for this figure.

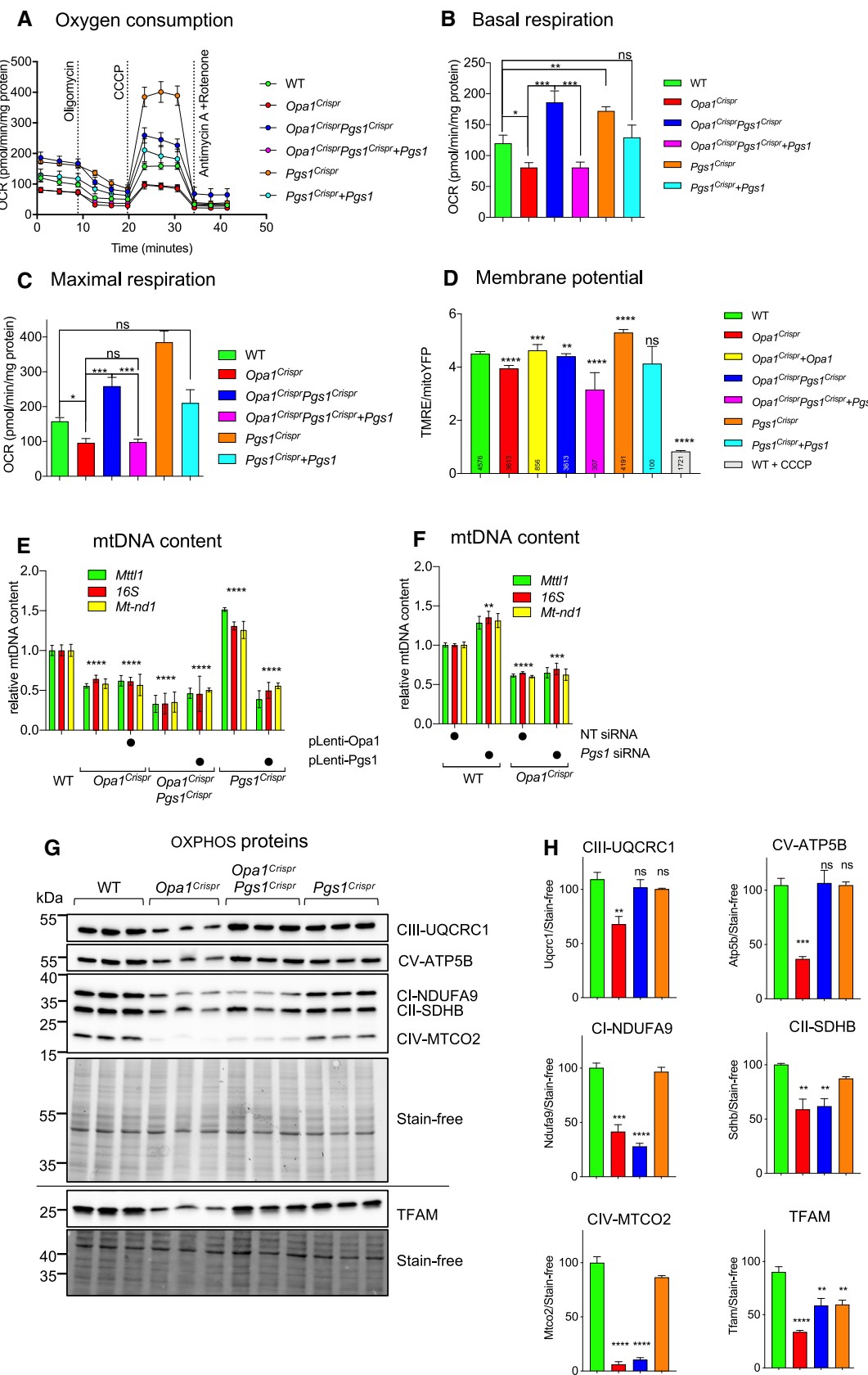

**Figure 7.**

level using confocal fluorescence microscopy (Fig 7D). We observed a reduction in membrane potential in $Opa1^{Crispr}$ MEFs that was rescued upon stable re-expression of untagged OPA1 (Fig 7D). $Opa1^{Crispr}Pgs1^{Crispr}$ MEFs exhibited a higher media membrane potential than $Opa1^{Crispr}$ MEFs but lower than that of WT cells measured by microscopy (Fig 7D). Thus, rescuing mitochondrial morphology of $Opa1^{Crispr}$ MEFs via PGS1 depletion improves mitochondrial respiration and membrane potential.

qPCR measurement of mitochondrial DNA (mtDNA) content using primer pairs targeting different regions of mtDNA revealed a depletion of mtDNA in $Opa1^{Crispr}$ MEFs, which was not rescued by PGS1 depletion by Crispr/Cas9-mediated ablation ($Opa1^{Crispr}Pgs1^{Crispr}$ MEFs) or siRNA depletion (Fig 7E and F). These data demonstrate that mitochondrial fragmentation and mtDNA maintenance defects in OPA1-deficient cells can be uncoupled.

To assess the impact of rebalancing mitochondrial dynamics on the oxidative phosphorylation (OXPHOS) complexes, we measured the levels of structural subunits by Western blot analyses (Fig 7G). $Opa1^{Crispr}$ MEFs showed reduced levels of NDUFA9 (Complex I), SDHA (Complex II), UQCRC2 (Complex III), MT-CO2 (Complex IV), and ATP5B (Complex V). Additional depletion of $Pgs1$ in $Opa1^{Crispr}Pgs1^{Crispr}$ MEFs could rescue the levels of SDHA, UQCRC2, and ATP5B, but not of NDUFA9 nor MT-CO2, which belong to the two respiratory complexes that derive the most structural subunits from mtDNA (Fig 7H). Consistent with elevated membrane potential measured (Fig EV4F) and mtDNA content (Fig 7E) in $Pgs1^{Crispr}$ cells, we observed an increase in oxygen consumption rates relative to WT MEFs, which could be lowered by functional complementation with re-expression of PGS1. Altogether, our data demonstrate functional amelioration of OXPHOS and bioenergetic defects in OPA1-deficient cells by depleting PGS1.

## Discussion

In this study, we present a new imaging approach using supervised machine learning to classify mitochondrial morphology of human and mouse fibroblasts according to pre-defined categories representing unopposed fusion and fission. This classification strategy is robust and can reproducibly recognize mitochondrial fragmentation induced either by accelerated fission or blocked fusion resulting from genetic or chemical manipulation without introducing user bias (Fig EV1, Appendix Fig S1A and B). Importantly, this workflow is highly scalable and robust at all levels, as evidenced by its application to automated, high-throughput phenotypic screens performed in human fibroblasts using the *Mitome* siRNA library (Fig 2A and D). This classification approach is relative and based on ground truths for mitochondrial morphologies generated by siRNA-mediated depletion of known fission or fusion genes, which are measured and applied at each image acquisition experiment. This compensates for the variability in mitochondrial morphologies generated due to experimental reasons (e.g., cell culture conditions, cell density, gas levels) and intrinsic population heterogeneity. We applied this unbiased strategy to classify mitochondrial morphology in an array of skin fibroblasts derived from patients suffering from Dominant Optic Atrophy plus (DOA+) and discovered that not all pathogenic variants in *OPA1* trigger mitochondrial fragmentation, despite patients belonging to the same clinical grouping (Table 1).

Neither the steady-state levels of OPA1 protein in fibroblasts (Appendix Fig S2D) nor the clinical manifestation could predict the p.Ser545Arg and p.Arg445His GTPase domain missense variants to be most phenotypically severe with respect to mitochondrial morphology (Fig 1B and C). While the molecular explanation for this discordance remains unexplained, we posit that additional factors may modulate mitochondrial morphology in patient fibroblasts. Yet to our knowledge no known genetic modifiers of disease genes involved in mitochondrial dynamics, including *OPA1*-related diseases have yet been identified. Within *OPA1*, genotype-phenotype relationships can be loosely established but it is not possible to fully predict clinical outcome for DOA and DOA+ patients solely based on the location and nature of pathogenic, mono-allelic variants in *OPA1*. Recently, intra-allelic variants of OPA1 have been documented to modify clinical and biochemical phenotypes in DOA+ (Bonifert *et al*, 2014), indicating that genetic modulation of OPA1 and its function(s) is formally possible. Moreover, chemically induced mitochondrial dysfunction in epithelial cells can be buffered by loss of function of other metabolic genes (To *et al*, 2019), arguing for the existence of genetic modifiers of MD beyond OPA1. To seek out genetic modifiers of mitochondrial morphology, we coupled our mitochondrial morphology imaging and quantification workflow to a bespoke siRNA library targeting the entire mitochondrial proteome (*Mitome*) and identified known and novel genes whose depletion promoted mitochondrial fragmentation (Fig 2B) or hypertubulation (Fig 2C) in control fibroblasts as well as 91 genes whose depletion could rescue mitochondrial fragmentation in *OPA1* mutant patient-derived fibroblasts. As such, these data provide a valuable resource for further investigation of the genetic regulation of mitochondrial dynamics. We discovered that several of the genes capable of reversing mitochondrial fragmentation in *OPA1* mutant cells with the most severely fragmented mitochondrial network (Fig 2D) also led to hypertubulation in control fibroblasts (Fig 2F), including known components of the mitochondrial fission apparatus (*DNM1L*, *MIEF1*, *MFF*, *SLC25A46*), which demonstrates in patient fibroblasts that imbalanced mitochondrial dynamics can be genetically re-equilibrated, as previously documented in animal models of MD (Chen *et al*, 2015; Wai *et al*, 2015; Yamada *et al*, 2018). *Mitome* screening also identified a cluster of ribosomal genes, whose individual depletion would be predicted to impair protein synthesis. Indeed, we can show that treating human *OPA1^{S545R}* or mouse *Opa1^{Crispr}* fibroblasts with CHX can promote stress-induced mitochondrial hyperfusion (SiMH). SiMH can be promoted by inhibition of transcription (Appendix Fig S6E), translation (Fig 4F), as well as the induction of ER stress (Tondera *et al*, 2009; Morita *et al*, 2017; Lebeau *et al*, 2018). We discovered *CALR*, which encodes a highly conserved chaperone protein that resides primarily in the ER involved in ER stress responses to be a suppressor of mitochondrial fragmentation (Dataset EV4, Fig 2 F). Finally, we also discovered an array of nuclear-encoded mitochondrial genes that rescued mitochondrial fragmentation but that had not previously associated with mitochondrial dynamics. These genes cover various classes of mitochondrial functions including mitochondrial gene expression, oxidative phosphorylation, and amino acid metabolism yet how these genes (Dataset EV4) or processes (Fig EV2H, Appendix Fig S3A) influence mitochondrial dynamics is unclear and warrants further investigation. Intriguingly, a substantial proportion of these genes are found to be mutated in

inborn errors of metabolism (Dataset EV4), yet the effects on mitochondrial morphology have not yet been explored. Altogether, our results demonstrate that it is possible to apply an unbiased, high-throughput imaging approach to identify candidate suppressors of mitochondrial dysfunction in MD patient fibroblasts.

We discovered PGS1 depletion could restore mitochondrial morphology to hypomorphic *OPA1* patient-derived fibroblasts and mouse embryonic fibroblasts (MEFs) mutated for *Opa1* as well as OPA1-deficient HeLa cells (Appendix Fig S3A–C). This occurred via a reduction in mitochondrial fission that was independent of both the recruitment of DRP1 to mitochondria (Fig 4A and B) and OMA1-dependent proteolytic processing of OPA1, which are key steps in pathological and physiological fission (Giacomello *et al*, 2020). Similar to DRP1 or OMA1 ablation, PGS1 depletion can protect from induction of stress-induced fission and fragmentation triggered by chemical agents known to promote OMA1-dependent OPA1 processing and/or DRP1 recruitment, thereby implicating PGS1 in mitochondrial fission.

The notion that impairing PGS1 and thus CL biogenesis might rescue mitochondrial morphology defects caused by OPA1 defects seem counterintuitive at first, given existing reports of a requirement of CL on opposing membranes for OPA1-mediated fusion (Ban *et al*, 2017). $Opa1^{Crispr}Pgs1^{Crispr}$ MEFs exhibit a similar block of IMM fusion as $Opa1^{Crispr}$ MEFs and it is therefore not possible to assess the contribution of CL for mitochondrial fusion. Nevertheless, several observations exclude CL depletion as a mechanism for restored mitochondrial morphology in hypomorphic OPA1-deficient cells. First, lipidomic analyses of $Opa1^{Crispr}$ MEFs revealed a depletion and alteration of CL acyl chain composition, neither of which was restored by additional depletion of either *Pgs1* or *Tamm41* (Fig 5E) even though mitochondrial morphology was restored in these cells. In fact, TAMM41 or PGS1-depleted $Opa1^{Crispr}$ MEFs exhibit even lower steady-state CL levels than $Opa1^{Crispr}$ MEFs (Fig 5E) and unaltered CL saturation states (Appendix Fig S7A). Second, WT MEFs depleted of either PGS1 or TAMM41 show reduced CL levels similar to $Opa1^{Crispr}$ MEFs but without an induction of mitochondrial fragmentation (Fig 5B and C). Third, inhibition of cardiolipin synthase (CLS1) does not cause mitochondrial fragmentation but rather promotes mitochondrial elongation (Fig 5 B) (Matsumura *et al*, 2018). Fourth, depletion of the CL remodeling enzyme Tafazzin (TAZ1), does not impair mitochondrial fusion nor trigger mitochondrial fragmentation (Ban *et al*, 2018). Finally, the inhibition of fission mediated by PGS1 depletion can be reversed by additional suppression of PRELID1, which is upstream in the CL biosynthesis pathway (Fig 5A) and is needed to deliver PA to the IMM. PRELID1 inhibition prevents the delivery of PA to the IMM (Connerth *et al*, 2012), which is normally converted to CDP-DAG and PGP by TAMM41 and PGS1, respectively (Tamura *et al*, 2020). In yeast, the deleterious effect of PA accumulation in the IMM is supported by observations that altered mitochondrial structure in pgs1 mutant cells, which accumulate PA in the IMM, can be rescued by additional deletion of the Prelid1 orthologue Ups1(Connerth *et al*, 2012). We therefore propose that the accumulation of PA rather the depletion of CL drives the suppression of mitochondrial morphology defects caused by OPA1 dysfunction.

From a functional perspective, restoration of mitochondrial morphology in OPA1-deficient MEFs rescued respiration but not cristae dysmorphology, apoptotic sensitivity, CL dysregulation nor mtDNA depletion observed in $Opa1^{Crispr}$ MEFs. The programmed cell death response manifested by $Opa1^{Crispr}Pgs1^{Crispr}$ MEFs was similar to that of $Opa1^{Crispr}$ MEFs when treated with apoptotic inducers, indicating the mitochondrial morphology alone does not dictate cell death sensitivity. Disruption of IMM architecture in OPA1 deficiency has recently been linked to the stability of the MICOS complex (Stephan *et al*, 2020) but whether CL deficiency contributes to cristae dysmorphology in these cells is unclear. Interestingly, the *Mitome* screen in $OPA1^{S545R}$ fibroblasts identified *DNAJC4*, *DNACJ11*, and *MTX1*; interactors of the MICOS complex, which is known to facilitate intramitochondrial lipid transport (Aaltonen *et al*, 2016, 35). In yeast, mitochondria lacking pgs1 exhibit altered cristae structure, characterized by extremely elongated cristae sheets resembling onion-like structures (Connerth *et al*, 2012). However, in MEFs, depletion of $Pgs1^{Crispr}$ reduces CL levels without an observable impact on cristae, arguing that CL depletion alone is not sufficient to cause cristae loss in mammalian cells.

The notion that balanced mitochondrial dynamics is critical for cellular health arises from the observations that dampening of mitochondrial fission confers physiological benefits in animal models of mitochondrial dysfunction characterized by mitochondrial fragmentation (Xiao *et al*, 2014; Wai *et al*, 2015; Xie *et al*, 2015; Acin-Perez *et al*, 2018; Yamada *et al*, 2018) but whether this paradigm is applicable to human disease has never been investigated. However, the relevance of existing genetic targets of mitochondrial fission is limited due to the essential nature of these genes. For DRP1, whole-body ablation of *Dnm1l* in mice causes embryonic lethality and tissue-specific deletion in organs most critically affected in MD is crippling. Moreover, loss-of-function variants in *DNM1L* or genes involved in DRP1-dependent fission including *MIEF1* (Charif *et al*, 2021), *SLC25A46* (Abrams *et al*, 2015, 46), *MFF* (Koch *et al*, 2016), *GDAP1* (Baxter *et al*, 2002), and *INF2* (Boyer *et al*, 2011) all-cause severe neurodegenerative diseases, making them unlikely therapeutic targets. The *Mitome* screen in $OPA1^{S545R}$ fibroblasts identified a cluster of cytosolic ribosomal genes, whose individual depletion would be predicted to impair protein synthesis. These findings are consistent with the observation that treatment of $OPA1^{S545R}$ patient fibroblasts with the cytosolic protein synthesis inhibitor CHX can suppress mitochondrial fragmentation in both $OPA1^{S545R}$ patient-derived fibroblasts (Fig 1F) and hypomorphic $Opa1^{Crispr}$ MEFs (Appendix Fig S4C and D), although pharmacological inhibition of global protein synthesis does not represent a viable therapeutic strategy to rescue defects in mitochondrial form and function. PGS1 depletion does not appear to incur cellular dysfunction *in vitro*, yet future studies are needed to determine the physiological relevance of this gene and whether it will serve as a useful modulator of mitochondrial function *in vivo*. In the meantime, we believe it worthwhile to consider PGS1 and other genetic modifiers that we have identified using our *Mitome* screening approach when evaluating the genetic and phenotypic landscape of OPA1-related diseases.

# Materials and Methods

### Human skin fibroblasts

Primary fibroblast cultures obtained from patients suffering from DOA+ carrying mono-allelic pathogenic variants in *OPA1* ($OPA1^{S545R}$, $OPA1^{R445H}$, $OPA1^{Q297X}$, $OPA1^{I432V}$, $OPA1^{c.2356-1G>T}$) and

healthy individuals with no signs of optic atrophy (CTL-1, CTL-2, CTL-3), which served as controls, were established as previously described (Amati-Bonneau et al, 2005; Kane et al, 2017) and described in Table 1. Written, informed consent was obtained from all human subjects. The experiments conformed to the principles set out in the WMA Declaration of Helsinki and the Department of Health and Human Services Belmont Report. Approval for research was granted by the Ethics Committee of the University Hospital of Angers (Comité de Protection des Personnes CPP Ouest II—Angers, France; Identification number CPP CB 2014/02; Declaration number DC-2011-1467 and Authorization number AC-2012-1507); and the Yorkshire and the Humber—Bradford Leeds Research Ethics Committee (REC reference: 13/YH/0310). OPA1 variants are described according to the OPA1 transcript variant 1 (RefSeq: NM_015560.2) and variant 8 (RefSeq: NM_130837.2) in Table 1. Compared with transcript variant 1 (RefSeq: NM_015560.2), the original transcript identified, transcript variant 8, based on an alternate splice pattern that contains two additional exons, 4b and 5b (Delettre et al, 2001). However, it maintains the same reading frame encoding an isoform (8) of 1,015 amino acids (aa). To maintain historical compatibility, variants are also described in the article according to transcript variant 1. The numbering of the nucleotides reflects that of the cDNA, with "+1" corresponding to the "A" of the ATG translation initiation codon in the reference sequence, according to which the initiation codon is codon 1, as recommended by the version 2.0 nomenclature of the Human Genome Variation Society (HGVS; http://varnomen.hgvs.org) (Dunnen et al, 2016). Following the HGVS guidelines, deduced changes are indicated between brackets.

### Mouse embryonic fibroblasts

Mouse embryonic fibroblasts (MEF) expressing mitochondrially targeted YFP (mitoYFP MEF) were isolated from $Gt(ROSA26)Sor^{mitoYFP/+}$ embryos on a C57Bl6/N genetic background at E13.5 and immortalized using a plasmid encoding SV40 large T antigen as previously described (Anand et al, 2014). Wild-type and $Yme1l^{-/-}$ (YKO) MEFs lacking mitoYFP (MEF1 WT) expression were generated from $Yme1l^{Loxp/LoxP}$ embryos on a C57Bl6/N genetic as previously described (Anand et al, 2014). MEFs lacking MiD49/MiD51/Mff generated as previously described (Osellame et al, 2016) were a gift from Dr. Michael Ryan.

### Plasmids

Complementary DNA (cDNA) encoding mouse Pgs1 with a C-terminal Myc tag and Opa1 with a C-terminal 9X Myc tag (pclbw-opa1(isoform 1)-myc; a gift from David Chan [Addgene plasmid # 62845]) (Mishra et al, 2014) were cloned into pDONR-221 and then pLenti6/Ubc (Invitrogen) using Gateway Technology (Invitrogen). Generation of an Opa1 isoform 1 construct lacking the C-terminal Myc tag was achieved by site-directed mutagenesis. MitoTAG constructs pMXs-3XHA-EGFP-OMP25 (a gift from David Sabatini [Addgene plasmid # 83356]) and pMXs-3XMyc-EGFP-OMP25 (a gift from David Sabatini [Addgene plasmid # 83355]) were used for mitochondrial immunocapture studies. pCMV-mRFP-PASS and pCMV-mRFP-PASS-4E plasmids (Zhang et al, 2014) used for PA localization studies were a kind gift from Dr. Guangwei Du.

### Cell culture conditions

Primary human fibroblasts were cultured in growth media: Dulbecco's modified Eagle's medium (DMEM) containing 4.5 g/l D-Glucose, GlutaMAX™ and pyruvate supplemented with 10% Fetal Bovine Serum (FBS) and 50 µg/ml penicillin/streptomycin (P/S) in a 5% $CO_2$ atmosphere at 37°C and used at passages between 10 and 15. MEFs were cultured in growth media: DMEM containing 4.5 g/l D-Glucose, GlutaMAX™ and pyruvate supplemented with 5% FBS and 50 µg/ml P/S in a 5% $CO_2$ atmosphere at 37°C. Cells were routinely tested for Mycoplasma by PCR.

### Generation of OPA1-, PGS1-, and DRP1-deficient and DRP1 KI cell lines

Genetic disruption of Opa1, Pgs1, and Dnm1l (Drp1) in MEFs was performed via CRISPR-Cas9 gene editing. The single-guide RNAs (sgRNAs) were designed using the CRISPR-Cas9 design tool from Benchling (https://www.benchling.com/crispr/) and for Exon 4 of Opa1 (sgRNA: forward: 5′-caccgTGCCAGTTTAGCTCCCGACC-3′ and reverse: 5′-aaacGGTCGGGAGCTAAACTGGCAc-3′) and Exon 2 of Pgs1 (sgRNA: forward: 5′-caccgTATGTCCCGAGGGTGTACAC-3′ and reverse 5′-aaacGTGTACACCCTCGGGACATAc-3′), and Exon 1 of Dnm1l (sgRNA: forward: 5′ caccgGCAGGACGTCTTCAACACAG-3′ and reverse 5′-aaacCTGTGTTGAAGACGTCCTGCc-3′). sgDNA oligonucleotides were annealed and cloned into the BbsI digested pSpCas9(BB)-2A-GFP vector (SpCas9(BB)-2A-GFP (PX458) was a gift from Feng Zhang (Addgene plasmid # 48138). MEFs were transfected with 5 µg of pSpCas9(BB)-2A-GFP plasmid containing the respective sgRNA using Lipofectamine 3000 (Life Technologies, L3000008). After 24 h incubation, GFP-positive cells were individually isolated by fluorescence-activated cell sorting. Clones were expanded and were validated by Western blotting, Sanger sequencing, Illumina MiSeq PE300 deep sequencing of PCR amplicons generated using primers in Dataset EV5. Crispr/Cas9-mediated knock-in of mTurquoise2 into the Dnm1l locus directly downstream of the start codon was performed using a plasmid encoding mTurquoise 2 flanked by 723 bp of upstream and 696 bp of downstream homologous DNA sequence cloned into pEX-A258 to create plasmid pTW306 (pEX-A258-Turq2_mDNM1L_mut). Two pairs of sgDNA targeting Dnm1l (sgDNA1: forward 5′-caccgCCGGGATCAGCGCCTCCATGACC-3′ and reverse 5′-GGTCATGGAGGCGCTGATCCCGGc-3′, sgDNA2: forward 5′-caccgAGCAGGCCACTGCAATGAATGGG and reverse 5′-CCCATTCATTGCAGTGGCCTGCTc-3′) were cloned into pSpCas9n(BB)-2A-Puro (PX462) V2.0 (a gift from Feng Zhang (Addgene plasmid # 62987) to generated plasmids pTW305 (pSpCas9n(BB)-DNM1L sgDNA 1_TW662_663_2A-Puro (PX462) V2.0) and pTW307 (pSpCas9n(BB)-DNM1L sgDNA 2_TW664_665_2A-Puro (PX462) V2.0), respectively. MitoYFP WT and $Opa1^{Crispr}$ MEFs were transfected with 4 µg of plasmids (1 µg each of pTW305 and pTW307 and 2 µg of pTW306 linearized with EcoRV) using Lipofectamine 3000 (Life Technologies, L3000008). After 24 h incubation, mTurquoise-positive, YFP-positive cells were individually isolated by fluorescence-activated cell sorting. Clones were expanded and were validated by PCR genotyping of genomic DNA using primers in Dataset EV5 and confocal imaging.

Genetic disruption of OPA1 in HeLa cells was performed via CRISPR-Cas9 gene editing targeting Exon 17 of OPA1 (sgRNA:

forward: 5′-caccgAAAAGCAGTCTGATACTGCA-3′ and reverse: 5′-aaacTGCAGTATCAGACTGCTTTTc-3′). sgDNA oligonucleotides were annealed and cloned into the BbsI digested pLentiCRISPRv2 (Addgene plasmid # 52961). HeLa cells were transfected with 5 µg of pLentiCRISPRv2 (Addgene plasmid # 52961) plasmid containing the respective sgRNA using Lipofectamine 3000 (Life Technologies, L3000008). After 24 h incubation in puromycin, cells were individually isolated by cytometry. Clones were expanded and were validated by western blotting, Sanger sequencing. $OPA1^{CRISPR}$ HeLa cells carry a heterozygous 19bp deletion causing a nonsense mutation (c.1763delTGCTTTTGGAAAATGGTAC, p.Glu637* mutation).

### Lentiviral and retroviral transductions

Lentiviral particles were generated from the following plasmids: pLenti6/Ubc-Opa1-9xMyc, pLenti6/Ubc-Opa1, pLenti6/Ubc-Pgs1, pMXs-3XHA-EGFP-OMP25, and pMXs-3XMyc-EGFP-OMP25. VSV-G-pseudotyped vectors were produced by transient transfection of 293T cells with a packaging construct, a plasmid producing the VSV-G envelope (pMD.G) and pBA-rev and pHDMH-gpM2 plasmids. Culture medium was collected at 24, 48, and 72 h, pooled, concentrated approximately 1,000-fold by ultracentrifugation, aliquoted, and stored at −80°C until used. Vector titers were determined by FACS cell Sorting on infected HCT116 cells infected with serial dilutions of vector stock. Transduction of MEFs was performed as previously described (Wai *et al*, 2016).

### Mitochondrial morphology imaging

Human Fibroblasts Cells were seeded on CellCarrier-384 or 96well Ultra microplate (PerkinElmer) and incubated for at least 24 h in growth media. Fibroblasts were fixed with 4% PFA-PBS (w/v) for 15 min, permeabilized in 0.1% (v/v) Triton X-100-PBS for 10 min, and blocked in 10% FBS-PBS overnight at 4°C. The following day, permeabilized cells were first stained with the primary antibody anti-TOM40 (diluted 1:1,000 in 5% FBS-PBS), washed three times with PBS, and then incubated with fluorescently coupled secondary antibody Alexa Fluor 488. Nuclei were finally marked with DAPI (1:10,000 in PBS). Images were acquired using the Operetta CLS High-Content Analysis System (PerkinElmer), with 40× Air/0.6 NA or 63× Water/1.15 NA. Alexa Fluor 488 and DAPI were excited with the 460–490 and 355–385 nm LEDs, respectively.

MEFs Cells were seeded on 96well CellCarrier Ultra imaging plates (PerkinElmer) 24 h before imaging. Nuclei were labeled with NucBlue™ Live ReadyProbes™ Reagent (Thermo Fisher Scientific). Fluorescent labeling of mitochondria was achieved using Tetramethylrhodamine Ethyl Ester Perchlorate (TMRE) and/or MitoTracker Deep Red at 100 nM for 30 min at 37°C, 5% $CO_2$ and/or with genetically encoded mitochondrially targeted YFP (mitoYFP). Spinning disk confocal images were acquired using the Operetta CLS or Opera Phenix High-Content Analysis systems (PerkinElmer), with 20× Water/1.0 NA, 40× Air/0.6 NA, 40× Water/1.1 NA or 63× Water/1.15 NA. YFP (460–490 nm), TMRE (530–560 nm), MitoTracker Deep Red (615–645 nm), mTurquoise2 (435–460 nm), and DAPI (355–385 nm) were excited the appropriate LEDs (Operetta CLS) or lasers (Operetta Phenix). FRAP experiments were performed with a Nikon Ti2E spinning disk microscope 60× Oil objective/NA1.4 equipped with a Photometrics Prime 95b cMOS camera (pixel 11um). Photobleaching was performed with a 405 laser at (35% power, 400 ms dwell time) and image collection proceeded immediately thereafter at 200 ms intervals. Quantification FRAP studies was performed using Fiji (ImageJ).

### SDS–PAGE immunoblot analysis

Cells were lysed in ice-cold RIPA buffer (50 mM Tris–HCl, pH 7.4, 150 mM NaCl, 1% (v/v) Triton X-100, 0.1% SDS, 0.05% sodium deoxycholate, 1 mM EDTA, and complete protease inhibitor cocktail mix (Roche)). After 30 min of incubation on ice, lysates were centrifuged at 16,000 g for 10 min at 4°C. Protein quantification of the cleared lysates was performed by Bradford colorimetric assay (Sigma) using a BSA standard curve. Absorption was measurement at 595 nm by the microplate reader Infinite M2000 (TECAN). 15 µg of each sample was reduced and negatively charged with 4X Laemmli Buffer (355 mM 2-mercaptoethanol, 62.5 mM Tris–HCl pH 6.8, 10% (v/v) glycerol, 1% (w/v) SDS, 0.005% (v/v) Bromophenol Blue). Samples were heated 5 min at 95°C and separated on 4–20% Mini-PROTEAN® TGX Stain-Free™ Precast gels (Bio-Rad) or on home-made 7% polyacrylamide gel for OPA1 immunodetection. Gels were then transferred to nitrocellulose membranes with Trans-Blot® Turbo™ Transfer system (Bio-Rad). Equal protein amount across membrane lanes was checked by Ponceau S staining or Stain-free detection. Membranes were blocked for 1 h with 5% (w/v) semi-skimmed dry milk dissolved in Tris-buffered saline Tween 0.1% (TBST), incubated overnight at 4°C with primary antibodies dissolved 1:1,000 in 2% (w/v) Bovine Serum Albumin (BSA), 0.1% TBST. The next day membranes were incubated at least 1 h in secondary antibodies conjugated to horseradish peroxidase (HRP) at room temperature (diluted 1:10,000 in 5% milk). Finally, membranes were incubated in Clarity™ Western ECL Substrate (Bio-Rad) for 2 min and luminescence was detected using the ChemiDoc® Gel Imaging System. Densitometric analysis of the immunoblots was performed using Image Lab Software (Bio-Rad).

### siRNA transfection

Silencing of the indicated genes was performed using forward transfection: 20 nM of the specific siRNA was mixed with Lipofectamine RNAiMax (Invitrogen), added on top of seeded cells, and left at 37°C in a $CO_2$ incubator for 72 h. Specific and non-targeting siRNAs were obtained from Dharmacon. For mouse siRNA: Negative control NT: D-001210-04-05; *Opa1* siRNA: L-054815-01-0005; *Drp1* siRNA: L-05 4815-01-0005; *Pgs1* siRNA: L-064480-01-0005, *Tamm41* siRNA: M-056928-01-0005, *Ptpmt1* siRNA: M-047887-01-0005, *Cls1* siRNA: M-055736-01-0005, *Prelid1* siRNA: M-065330-01-0005. For human siRNA: Negative control NT: D-001210-04-05, *PGS1* siRNA: D-009 483-02-0002 + D-009483-13-0002 + D-009483-01-0002 + D-009483-04-0002; *TAMM41* siRNA: L-016534-02-0005; *DNM1L* siRNA: M-012092-01- 0005; *OPA1* siRNA: M-005273-00- 0005; *DNAJC11 siRNA*: D-021205-03-0002; *DNAJC4 siRNA*: D-020055-17-0002.

### RT–qPCR

Total RNA was extracted using TRIzol™ Reagent and chloroform, purified, and subjected to DNA digestion using the NucleoSpin RNA kit (MACHEREY-NAGEL). RNA concentration was measured using

NanoQuant Plate™ (Infinite M200, TECAN), and 1 µg of total RNA was converted into cDNA using the iScript Reverse Transcription Supermix (Bio-Rad). RT–qPCR was performed using the CFX384 Touch Real-Time PCR Detection System (Bio-Rad) and SYBR® Green Master Mix (Bio-Rad) using the primers listed in Table 3. Actin or APP were amplified as internal standards. Data were analyzed according to the $2^{-\Delta\Delta CT}$ method (Livak & Schmittgen, 2001).

## Analysis of oxygen consumption rates

Oxygen consumption was measured with the XFe96 Analyzer (Seahorse Biosciences) and High-Resolution Respirometry (O2k-Fluorespirometer, Oroboros, AT). For Seahorse experiments, cells (30,000 MEFs or 20,000 human fibroblasts experimentally optimized) were seeded onto 96-well XFe96 cell culture plates. On the following day, cells were washed and incubated with Seahorse XF Base Medium completed on the day of the experiment with 1 mM Pyruvate, 2 mM Glutamine, and 10 mM Glucose. Cells were washed with the Seahorse XF Base Medium and incubated for 45 min in a 37°C non-$CO_2$ incubator before starting the assay. Following basal respiration, cells were treated sequentially with: oligomycin 1 µM, CCCP 2 µM and Antimycin A 1 µM + 1 µM Rotenone (Sigma). Measurements were taken over 2-min intervals, proceeded by a 1-min mixing and a 30 s incubation. Three measurements were taken for the resting OCR, three for the non-phosphorylating OCR, three for the maximal OCR, and three for the extramitochondrial OCR. After measurement, the XFe96 plate was washed with Phosphate-Buffered Saline (PBS) and protein was extracted with RIPA for 10 min at room temperature. Protein quantity in each well was then quantified by Bicinchoninic acid assay (BCA). Absorption was measurement at 562 nm by the microplate reader (Infinite M200, TECAN) and used to normalize OCR data.

For O2k respirometry, 2 million intact MEFs were transferred to the 37°C-heated oxygraph chambers containing MiRO5 buffer. Basal respiration was measured first. Then, 10 nM oligomycin, 2 µM CCCP, and 25 µM AntimycinA + 5 µM Rotenone were sequentially injected, and non-phosphorylating OCR, maximal OCR, and extramitochondrial OCR were measured, respectively. Finally, cells were recovered and washed once with PBS. Protein was extracted with RIPA Buffer for 30 min at 4°C and quantified using Bradford assay. Absorption was measurement at 595 nm by the microplate reader (Infinite M200, TECAN) and used to normalize $O_2$ flux.

## mtDNA content quantification

Genomic DNA was extracted using the NucleoSpin Tissue (MACHEREY-NAGEL) and quantified with NanoQuant Plate™ (Infinite M200, TECAN). RT–qPCR was performed using the CFX384 Touch Real-Time PCR Detection System (Bio-Rad), 25 ng of total DNA, and the SYBR® Green Master Mix (Bio-Rad). Actin or APP was amplified as internal standards. Primers sequence are listed in Table 3. Data were analyzed according to the method (Livak & Schmittgen, 2001).

## Mitochondrial morphology quantification

Harmony Analysis Software (PerkinElmer) was used for automated image analysis as described in detail in Table 3 PhenoLOGIC sequence. Z-projected images first undergo brightfield correction. Nuclei and cellular segmentation were defined using the "Find

Nuclei" building block with the HOECHST 33342 channel and the "Find Cytoplasm" building block with the Alexa 488 or TMRE (mitochondria) channel. Mitochondrial network was analyzed using SER Texture properties (Ridge, Bright, Edge, Dark, Valley, Spot), and the PhenoLOGIC supervised machine learning algorithm was used to identify the best properties able to segregate the three populations: "Normal", "Fragmented" and "Hypertubulated" network. ~ 200–400 cells of each control (Normal: WT + DMSO or WT + NT siRNA, Fragmented: WT + CCCP or WT + OPA1 or Opa1 siRNA, Hypertubulated: WT + CHX or WT + DNM1L or Dnm1l siRNA) were selected to feed the algorithm for training. Automatic single-cell classification of non-training samples (i.e., unknowns) was carried out by the supervised machine learning module.

## High-content screening

The siRNA library (Mitome; 1,531 siRNAs) consists of a Cherrypick SmartPool siRNA library targeting all known and predicted mitochondrial genes based on Mitominer V4 and Mitocarta. 500 nl of 2 µM siRNAs (20 nM final concentration) were distributed on 6 different 384-well imaging plates (CellCarrier Ultra, PerkinElmer), as described in Table 3, using Echo 550 (Labcyte Inc.) and were left to dry under a sterile hood at least for 24 h. For each well, 10 µl of PBS containing 0.1 µl of Lipofectamine RNAiMax was automatically added using the pipetting robot VIAFLO 384 (Integra). After 1 h incubation at room temperature (RT), 2,000 $OPA1^{S545R}$ patient fibroblasts (in 40 µl) were added to each well for reverse transfection using the VIAFLO 384 (Integra). Cells were incubated at 37°C, 5% $CO_2$ for 72 h and finally immunostained using the automatic pipette VIAFLO 384 (Integra) as described below.

| Step | Solution | Incubation time and temperature |
|---|---|---|
| 1-Wash 1× | PBS | No incubation/37°C |
| 2-Fixation | PFA 4% | 15 min/37°C |
| 3-Wash 3× | PBS | No incubation/RT |
| 4-Permeabilization | 0.2% Triton | 10 min/RT |
| 5-Wash 3× | PBS | No incubation/RT |
| 6-Saturation | 10% FBS | Overnight/4°C |
| 7-Primary antibody | anti-TOMM40 (rabbit) 1:1,000 in 5% FBS-PBS | Overnight/4°C |
| 8-Wash 3× | PBS | No incubation/RT |
| 9-Secondary antibody | anti-rabbit-Alexa488 1:1,000 in 5% FBS-PBS | 2 h/RT |
| 10-Wash 3× | PBS | No incubation/RT |
| 11-Nuclei staining | DAPI 1:10,000 in PBS | 30 min/RT |
| 12-Wash 3× | PBS | No incubation/RT |

Images were acquired using the Operetta CLS High-Content Analysis system (PerkinElmer), with 40× Air/0.6 NA. 9 fields of view with 2 slices (z =− 6.5 and −7.5) were captured per well. Alexa 488 and DAPI were excited with the 460–490 and 355–385 nm LED, respectively.

Mitochondrial morphology was automatically quantified using the Harmony Analysis Software (PerkinElmer) as described in details in Table 3 PhenoLOGIC sequence. A brightfield correction (fixed, string) was applied to all Z-projected images. Nuclei and cells were first segmented using the "Find Nuclei" building block on the DAPI channel and the "Find Cytoplasm" building block on the Alexa 488 channel. SER Texture analysis (Ridge, Bright, Edge, Dark, Valley, Spot) of the mitochondrial network was then calculated. The PhenoLOGIC supervised machine learning module of Harmony (available through the "Select Population-Linear Classifier" building block) was used to identify the most relevant SER textures able to segregate the three populations (*fragmented*: $OPA1^{S545R}$ + NT siRNA, *hyperfragmented*: $OPA1^{S545R}$ + OPA1 siRNA and rescued: $OPA1^{S545R}$ + DNM1L siRNA mitochondrial morphologies). For training, ~ 800 cells per training class (ground truth) of mitochondrial morphologies were manually selected in each control well of each plate. The supervised machine learning algorithm is then able to classify mitochondrial morphology of each well into those three categories. To evaluate the quality of the screening, we calculated the Z-score of each plate using the following formula: $Z - score = 1 - 3(\sigma_p + \sigma_n)/|\mu_p - \mu_n|$ where $\mu_p$ and $\sigma_p$ are the mean and standard deviation values of the positive control $p$ (rescued morphology: $OPA1^{S545R}$ patient fibroblasts + DNM1L siRNA) and $\mu_n$ and $\sigma_n$ those of the negative control $n$ (fragmented morphology: $OPA1^{S545R}$ patient fibroblasts + NT siRNA). The Z-score of all plates were above 0.5 reflecting the robustness of the screening (Plate1 = 0.82, Plate2 = 0.55, Plate3 = 0.80, Plate4 = 0.79, Plate5 = 0.80, Plate6 = 0.70).

In order to define a threshold of phenotypic rescue of mitochondrial morphology, we designed and deployed a univariate three-components statistical model using R (https://www.R-project.org) to define the siRNAs able to re-establish mitochondrial morphology to same extent as with DNM1L siRNA. We used two models, one designed to identify *hypertubular* hits data and another to identify *hyperfragmented* hits among the *Mitome* library siRNA pools. Each model has three components. For the rescued threshold in $OPA1^{S545R}$ siRNA Mitome screen, these are the $OPA1^{S545R}$ NT siRNA (negative control), the $OPA1^{S545R}$ DNM1L siRNA (positive control for rescued morphology), and the $OPA1^{S545R}$ cells transfected with the 1,531 siRNAs of the *Mitome* library. For the hyperfragmented threshold in $OPA1^{S545R}$ siRNA Mitome screen, these are the $OPA1^{S545R}$ NT siRNA (negative control), the $OPA1^{S545R}$ OPA1 siRNA (positive control for hyperfragmented morphology), and the $OPA1^{S545R}$ cells transfected with the 1,531 siRNAs of the Mitome library.

**Membrane potential measurement**

Membrane potential was determined by FACS and live confocal microscopy. For FACS analyses, $1 \times 10^6$ MEFs or human fibroblasts were plated in 10 cm$^2$ dishes and incubated 24 h with growth media. The next day, cells were treated with 100 nM TMRE for 20 min at 37°C, 5% CO$_2$ or with 20 μM Carbonyl Cyanide m-chlorophenyl hydrazine (CCCP) for 30 min followed by 30-min incubation with 100 nM TMRE + 20 μM CCCP for 20 min at 37°C, 5% CO$_2$. Cells were washed with PBS, dissociated from the dish with 0.05% Trypsin (Thermo Fisher Scientific), and centrifuged 5 min at 2,000 g. The cell pellet was then suspended in PBS containing SYTOX™ Blue Dead Cell Stain (diluted 1:5,000). The single-cell fluorescence was measured using the CytoFLEX flow cytometer

(Beckman Coulter). Dead cells (SYTOX™ Blue positive cells) were detected with the channel PB450 (450/45 BP) and discarded from analysis. TMRE-positive cells were detected with the PE channel (585/42 BP), and the median of TMRE intensity was used for analysis. For MEFs expressing mitoYFP, YFP signal was detected using the channel FITC (525/40 BP) and compensation between FITC and PE channels was manually calculated.

For confocal microscopy, the genetically encoded mitochondrially targeted YFP MEFs were seeded in 96well CellCarrier Ultra imaging plates (PerkinElmer) 1 day before the measurement. The next day, nuclei were labeled with NucBlue™ Live ReadyProbes™ Reagent (Thermo Fisher Scientific) and cells were treated with 100 nM TMRE for 20 min at 37°C, 5% CO$_2$ or with 20 μM Carbonyl Cyanide m-chlorophenyl hydrazine (CCCP) for 30 min followed by 30-min incubation with 100 nM TMRE + 20 μM CCCP for 20 min at 37°C, 5% CO$_2$. Spinning disk confocal images were acquired using the Operetta CLS High-Content microscope (PerkinElmer) with 40× Air/0.6 NA. YFP, TMRE, and NucBlue were excited with the 460–490, 530–560, and 355–385 nm LEDs, respectively. TMRE and YFP signal per cell were quantified using the Harmony Analysis Software (PerkinElmer).

**Cell death assay**

MEFs were plated in 96- or 384-well imaging plates (CellCarrier Ultra, PerkinElmer) and incubated at least 1 day at 37°C, 5% CO$_2$. The day of experiment, cells were incubated with NucBlue™ Live ReadyProbes™ Reagent (Thermo Fisher Scientific) and Propidium Iodide (PI, Sigma) and treated either with 4 μM Actinomycin D + 10 μM ABT-737 ± 20 μM qVD or 0.5 μM Staurosporine ± 20 μM or 16 μM etoposide ± 20 μM qVD for the indicated time. Total cells (stained by NucBlue) and dead cells (stained by PI$^+$) were imaged every hour for the indicated time with the Operetta CLS High-Content microscope (PerkinElmer) at 40× Air/0.6 NA. PI and NucBlue were excited with the 530–560 and 355–385 nm LEDs, respectively. PI$^+$/total cells over time were quantified using the Harmony Analysis Software (PerkinElmer).

**Stress-induced mitochondrial fission and fusion imaging**

2,000 MEFs expressing mitoYFP were plated in 384-well and incubated 24 h at 37°C, 5% CO$_2$. The day of experiment, nuclei were labeled with NucBlue™ Live ReadyProbes™ Reagent (Thermo Fisher Scientific) for 30 min at 37°C, 5% CO$_2$. For stress-induced fission imaging, cells were treated with 5 μM CCCP or 16 μM 4Br-A23187 for the indicated time. For stress-induced hyperfusion imaging, cells were treated with 10 μM CHX or 0.5 μM ActD. Nuclei (NucBlue) and mitochondria (YFP) were imaged every hour for the indicated time using the Operetta CLS High-Content microscope (PerkinElmer) at 40× Air/0.6 NA. YFP and NucBlue were excited with the 460–490 and 355–385 nm LEDs, respectively. Finally, mitochondrial morphology was quantified as described in the "Mitochondrial morphology quantification" section.

**DRP1 mitochondrial recruitment assay**

2,000 MEFs expressing mitoYFP were plated in 384 well and incubated 24 h at 37°C, 5% CO$_2$.

Cells were then fixed for 10 min with 37°C -prewarmed 4% paraformaldehyde in PHEM Buffer (60 mM PIPES, 25 mM HEPES, 10 mM EGTA, 2 mM $MgCl_2$, pH 7.3), permeabilized for 10 min with 0.1% Triton X-100 in PBS, and blocked overnight at 4°C with 10% FBS in PBS. Mitochondria were stained overnight at 4°C with anti-TOMM40 (diluted 1:1,000; ProteinTech #18409-1-AP) primary antibody and DRP1 with α-DLP1 primary antibody (diluted 1:1,000, BD # 611112). Cells were incubated with anti-rabbit Alexa 568 (1:1,000; goat anti-rabbit IgG Alexa Fluor 568; Invitrogen #A11011) and anti-mouse Alexa A647 (1:1,000; goat anti-mouse IgG Alexa Fluor 647; Invitrogen #A21236) for 2 h at RT. Finally, nuclei were stained for 30 min at RT with Hoechst 33342 diluted 1:10,000 in PBS. Images were acquired using the Operetta CLS High-Content Analysis system (PerkinElmer), with 63× Water/1.15 NA. Five fields of view with three slices ($z = 0$, 0.5 and 1) were captured per well. Alexa 647, Alexa 568, and Hoechst were excited with the 615–645, 530–560, and 355–385 LED, respectively. Colocalization of DRP1 and TOMM40 was evaluated using the Harmony Analysis Software (PerkinElmer) as described in detail in Dataset EV1.

For DRP1KI MEFs, 2,000 cells were plated in 384 well, incubated 24 h at 37°C, 5% $CO_2$ and imaged in live using the Opera Phenix High-Content Screening system (PerkinElmer), with 63× Water/1.15 NA. 42 fields of view with three slices ($z = -1$, 0.5 and 0) were captured per well. YFP (mitochondrial network) and mTurquoise2 (Drp1-mTurquoise2 tagged) were excited with the 490–515 and 435–460 lasers, respectively. Colocalization of Drp1-mTurquoise2 and YFP was evaluated using the Harmony Analysis Software (PerkinElmer) as described in detail in Dataset EV1.

## DRP1 cross-linking assay

Cells were harvested in PBS and mechanically disrupted in an ice-cold buffer medium (containing 220 mM Mannitol, 70 mM Sucrose, 10 mM Tris-KOH, and 1 mM EDTA, pH=7,4 supplemented with complete protease inhibitor cocktail mix [Roche]) by passing through a 25-gage syringe (20 strokes). Nuclei and unbroken cells were discarded by centrifugation at 700 $g$ for 10 min at 4°C. The post-nuclear supernatants were incubated with 10 mM 1,6-bismaleimideohexane (BMH) or DMSO for 2 h at 4°C, and the cross-linking reactions were stopped by the addition of 50 mM dithiothreitol (DTT). Proteins were quantified as described previously and 15 µg of each post-nuclear supernatants was reduced and negatively charged with 4X Laemmli Buffer (355 mM 2-mercaptoethanol, 62.5 mM Tris–HCl pH 6.8, 10% (v/v) glycerol, 1% (w/v) SDS, 0.005% (v/v) Bromophenol Blue), were heated 5 min at 95°C, and separated on 4–20% Mini-PROTEAN® TGX Stain-Free™ Precast gels (Bio-Rad). Gels were then transferred to nitrocellulose membranes with Trans-Blot® Turbo™ Transfer system (Bio-Rad). Equal protein amount across membrane lanes was checked by Stain-free detection. Membranes were blocked for 1 h with 5% (w/v) semi-skimmed dry milk dissolved in Tris-buffered saline Tween 0.1% (TBST), incubated overnight at 4°C with DRP1 antibody (26187-1-AP, ProteinTech) dissolved 1:1,000 in 2% (w/v) Bovine Serum Albumin (BSA), 0.1% TBST. The next day membranes were incubated at least 1 h in secondary antibodies conjugated to horseradish peroxidase (HRP) at room temperature (diluted 1:10,000 in 5% milk). Finally, membranes were incubated in Clarity™ Western ECL Substrate (Bio-Rad) for 2 min and luminescence was detected using the ChemiDoc® Gel Imaging System. Densitometric analysis of the immunoblots was performed using Image Lab Software (Bio-Rad).

## Mitochondrial isolation

Mitochondria were isolated as previously published (Chen et al, 2017). In brief, MEFs were infected with retroviral particles containing pMXs-3XHA-EGFP-OMP25, selected with 10 µg/ml Blasticidin and the expression of HA-tag was verified by SDS–PAGE. The day of experiment, ~ 30 million MEFs were collected, washed with KPBS buffer (136 mM KCl and 10 mM $KH_2PO_4$, pH 7.25), and homogenized with 25 stokes of the plunger at 1000 rpm at 4°C. Nuclei and debris were discarded by centrifugation at 1,000 $g$ for 2 min at 4°C. The supernatant was collected and subjected to immunocapture with prewashed anti-HA magnetic beads for 30 min on end-over-end rotator 4°C. The beads were then washed three times and resuspended in 500 µl KPBS. 30% of the suspension beads was set aside and used for immunoblotting. The remaining beads were store at −150°C for the indicated analysis.

## Transmission electron microscopy

Cells were grown on sapphire disks of 3 mm diameter (Engineering Office M. Wohlwend GmbH, Switzerland) previously coated with a carbon film (McDonald et al, 2010) and frozen with a Leica ICE high-pressure freezer machine (Leica microsystems, Austria) with fetal calf serum as cryoprotectant. The freeze-substitution was done in a Leica AFS2 machine in dry acetone containing 1% osmium tetroxide, 0.1% uranyl acetate, and 5% water as previously published (Walther & Ziegler, 2002). Samples were gradually infiltrated at RT with epoxy resin and after heat polymerization the sapphire discs were removed from the plastic block. Sections with a thickness of 70 nm were cut with a Leica UCT microtome and collected on carbon, formvar-coated copper grids. Sections were contrasted with 4% aqueous uranyl acetate and Reynolds lead citrate. Generation of ultra-large high-resolution electron microscopy maps was acquired using a TECNAI F20 transmission electron microscope (FEI) with a field emission gun (FEG) as an electron source, operated at 200 kV, and equipped with a GATAN Ultrascan US4000 CCD camera. The SerialEM software (Mastronarde, 2005; Schorb et al, 2019) was used for multi-scale mapping as follows: Initially, a full grid map was acquired at 190× magnification (pixel size = 551.75 nm). Middle magnification maps at 2,500× (pixel size = 35.98 nm) were acquired in areas with cells. Finally, high magnification maps (14,500×, pixel size = 6.194 nm) were collected at areas of interest, usually covering large part of the cellular cytoplasm (maps consisted of 100–300 micrographs/pieces) where many mitochondria were observed. Stacks of montages were displayed using the *3dmod* interface of IMOD (Kremer et al, 1996). The initial piece coordinates for each micrograph are either saved at the header of the mrc stack file by SerialEM, or in case of very large montages, at the additional metadata file mdoc. The "Align Serial Sections/ Blend Montages" interface of IMOD (Mastronarde & Held, 2017) was used for blending the stack of micrographs to a single large image by calling the blendmont function of IMOD. Quantification of cristae number and OMM/IMM perimeter was performed using ImageJ (Rueden et al, 2017).

## Quantitative mass spectrometry of lipids

Mass spectrometric analysis was performed essentially as described (Özbalci *et al*, 2013; Tatsuta, 2017). All internal standards were purchased from Avanti Polar lipid. Lipids were extracted from isolated pure mitochondria or whole cell pellet in the presence of internal standards of major phospholipids (PC 17:0-20:4, PE 17:0-20:4, PI 17:0-20:4, PS 17:0-20:4, PG 17:0-20:4, PA 15:0-18:1-d7 and CL mix I), cholesterol (cholesterol-d7), cholesteryl esters (19:0 cholesterol ester), and TAG (D5 TAG mix I). Extraction was performed using automated liquid handling robot (CyBio FeliX, Analytik Jena) according to Bligh and Dyer with modifications. Briefly, 7.5 µg mitochondria or $1 \times 10^5$ cells in 80 µl water and internal standards (22, 17, 8.8, 6.5, 2.5, 3.0, 8, 10, 8.5, and 4 pmole of PC 17:0-20:4, PE 17:0-20:4, PI 17:0-20:4, PS 17:0-20:4, PG 17:0-20:4, PA 15:0-18:1-d7, CLs, cholesterol-d7, 19:0 cholesterol ester, and TAGs, respectively) mixed with 0.3 ml of chloroform/methanol (1:2 (v/v)) for 10 min. After addition of 0.1 ml chloroform and of 0.1 ml $H_2O$, the sample was mixed again for 10 min, and phase separation was induced by centrifugation (800 *g*, 2 min). The lower chloroform phase was carefully transferred to a clean glass vial. 20 µl of the neutral lipid extract was taken to a glass vial, dried and incubated in acetyl chloride/chloroform (1:5) for 2 h at 25°C under hume hood for chemical derivatization. The upper water phase was mixed with 20 µl 165 mM HCl and 100 µl chloroform for 10 min. After phase separation, the lower chloroform phase was carefully transferred to the glass vial with the rest of chloroform phase from the first extraction. The solvent was evaporated by a gentle stream of argon at 37°C. Lipids were dissolved in 10 mM ammonium acetate in methanol, transferred to Twin.tec PCR plate sealed with Thermowell sealing tape and analyzed on a QTRAP 6500 triple quadrupole mass spectrometer (SCIEX) equipped with nano-infusion splay device (TriVersa NanoMate with ESI-Chip type A, Advion).

## Statistical analysis

Experiments were repeated at least three times except for the following, which were repeated two times: Fig 1C (195–2,496 cells per cell line were analyzed per experiment), Fig 1G (879–4,154 cells per cell line were analyzed per experiment), and Fig 4E (two independent experiments with 18 to 52 cells analyzed per genotype). Quantitative analyses were conducted blindly. Randomization of groups (e.g., different genotypes) was performed when simultaneous, parallel measurements were not performed (e.g., Oroboros, flow cytometry). For high-throughput measurements (e.g., mitochondrial morphology, cell death), all groups were measured in parallel to reduce experimental bias. Statistical analyses were performed using GraphPad Prism 9 software. Data are presented as mean ± SD or SEM where indicated. The statistical tests used, and value of experiment replicates are described in the figure legends. Tests were considered significant at *P*-value < 0.05 (\**P* < 0.05; \*\**P* < 0.01; \*\*\**P* < 0.0001; \*\*\*\**P* < 0.0001).

## Data availability

This study includes no data deposited in external repositories.

**The paper explained**

**Problem**

Genetic mutations in the gene Optic Atrophy 1 (*OPA1*) cause autosomal dominant optic atrophy (DOA)—one of the most common forms of mitochondrial disease. The majority of patients develop isolated optic atrophy, which is a deterioration of the optic nerve, yet about 20% of patients develop more severe neurological disease (DOA+) that cannot be fully explained by the location or nature of the disease-causing mutation in *OPA1*. It has not yet been established whether phenotypic severity can be modulated by genetic modifiers of *OPA1*.

**Results**

We developed a mitochondrial imaging and analysis pipeline that allowed us to perform high-throughput phenotypic screening of primary fibroblast from patients suffering from DOA+. We screened 1,531 nuclear-encoded mitochondrial genes with a bespoke siRNA library and identified 91 genes whose depletion could suppress mitochondrial fragmentation in OPA1 mutant fibroblasts, including *PGS1*.

**Impact**

Our study demonstrates that mitochondrial defects cause by OPA1 deficiency are variable and can be influenced by the action of other mitochondrial genes. The Mitome screening approach we developed may pave the way for the functional screening of genetic modifiers directly in the cells of patients that suffer from DOA, which could be coupled with diagnostic applications of *omics* technologies already in routine clinical use to gain insights into the variable penetrance and expressivity of this disorder and other types of mitochondrial disease.

**Expanded View** for this article is available online.

## Acknowledgements

We thank Kristin Tsuo and Vincent Guillemot for statistical assistance in R, Etienne Kornobis for Illumina sequencing, and Pierre-Henri Commere and Sandrine Schmutz for flow cytometry services at the Institut Pasteur. Imaging on the Opera Phenix, funded by the Région Ile-de-France program DIM1-Health, was facilitated by Nathalie Aulner. We thank Sylvie Fabrega of the Viral Vector for Gene Transfer core facility of Structure Fédérative de Recherche Necker, Université de Paris for lentiviral particle synthesis. We thank Michael Ryan for providing MEFs lacking *MiD49/MiD51/Mff*, Nils-Göran Larsson for providing mitoYFP mice, and Guangwei Du for plasmids. We thank Arnaud Echard for critical reading of the manuscript and Marie Lemesle for excellent administrative assistance. T.W. is supported by the European Research Council (ERC) Starting Grant No. 714472 (Acronym "*Mitomorphosis*") and ATIP-AVENIR (INSERM/CNRS). E.C. is supported by a PhD scholarship from the French Ministry of Higher Education, Research, and Innovation (Ministère français de l'Enseignement supérieur, de la Recherche et de l'Innovation). T.L. was supported by funds of the German Research Council (CRC1218, project number 269925409. P.YWM. is supported by a Clinician Scientist Fellowship Award (G1002570) from the Medical Research Council (UK) and also receives funding from Fight for Sight (UK), Moorfields Eye Charity, the Isaac Newton Trust (UK), the National Eye Research Centre (UK), the International Foundation for Optic Nerve Disease (IFOND), the UK National Institute of Health Research (NIHR) as part of the Rare Diseases Translational Research Collaboration, the NIHR Cambridge Biomedical Research Centre (BRC-1215-20014), and the NIHR Biomedical Research Centre based at Moorfields Eye Hospital NHS Foundation Trust and UCL Institute of Ophthalmology. The views expressed are those of the author(s) and not necessarily those of the NHS, the NIHR, or the Department of Health.

## Author contributions

TW conceived the study. PY-W-M and PR provided human fibroblasts. EC performed all experiments with the exception of lipidomic profiling, which was conducted by TT, electron microscopy, which was conducted by AG and MS, FRAP fusion assay, which was conducted by TW. PL and EC developed the methodology and performed the Mitome siRNA screening. EV performed cell culture, transfections, and qPCR experiments with EC. EC, TT and TW analysed the data. TW provided third party funding for the study. TW and EC wrote the paper, which was read by all authors and revised by TW, EC, PYWM, PR, TT, and TL.

## Conflict of interest

The authors declare that they have no conflicts of interest.

## For more information

Cure ADOA Foundation: https://adoa.eu/en

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
