## [Review Process File · EMBO Molecular Medicine]

High-throughput screening identifies suppressors of mitochondrial fragmentation in OPA1 fibroblasts

Emma Cretin, Priscilla Lopes, Elodie Vimont, Takashi Tatsuta, Thomas Langer, Anastasia Gazi, Martin Sachse, Patrick Yu-Wai-Man, Pascal Reynier, and Timothy Wai

DOI: [10.15252/emmm.202013579](https://doi.org/10.15252/emmm.202013579)

Corresponding authors: Timothy Wai (timothy.wai@pasteur.fr)

Review Timeline:

Submission Data:	1st Dec 20
Editorial Decision:	23rd Dec 20
Revision Received:	9th Mar 21
Editorial Decision:	25th Mar 21
Revision Received:	29th Mar 21
Accepted:	1st Apr 21

Editor: Zeljko Durdevic

Transaction Report:

23rd Dec 2020

Dear Dr. Wai,

Thank you for the submission of your manuscript to EMBO Molecular Medicine. We have now received feedback from the three reviewers who agreed to evaluate your manuscript. As you will see from the reports below, the referees acknowledge the interest and novelty of the study but also raise serious and partially overlapping concerns that should be addressed in a major revision. During the cross-commenting session it became clear that the phospholipidomics experiments suggested by the referee #1 (point #4) are not required and should rather be addressed in writing.

Addressing the reviewers' concerns in full, in writing or experimentally, will be necessary for further considering the manuscript in our journal, and acceptance of the manuscript will entail a second round of review. EMBO Molecular Medicine encourages a single round of revision only and therefore, acceptance or rejection of the manuscript will depend on the completeness of your responses included in the next, final version of the manuscript. For this reason, and to save you from any frustrations in the end, I would strongly advise against returning an incomplete revision.

We realize that the current situation is exceptional on the account of the COVID-19/SARS-CoV-2 pandemic. Therefore, please let us know if you need more than six months to revise the manuscript.

I look forward to receiving your revised manuscript.

Yours sincerely,

Zeljko Durdevic

***** Reviewer's comments *****

Referee #1 (Remarks for Author):

Mutations in the mitochondrial fusion protein OPA1, which result in mitochondrial fragmentation, cause autosomal optic atrophy (DOA). About 20% of OPA1 mutation carriers develop a more severe phenotype named DOA+. In this manuscript, Cretin et al established a high-throughput imaging pipeline to perform quantitative mitochondria morphology analysis coupled with a siRNA library targeting the entire known mitochondrial proteome in fibroblasts. In DOA+ patient fibroblasts, the authors identified 91 candidate genes whose depletion prevents mitochondrial fragmentation, and focused on phosphatidyl glycerophosphate synthase (PGS1), which belongs to the cardiolipin

synthesis pathway. Cretin et al demonstrated that PGS1 depletion rescues mitochondrial fragmentation in hypomorphic OPA1 mutations by inhibiting mitochondrial fission, which improves defective respiration, but does not rescue mtDNA depletion, cristae dysmorphology or apoptotic sensitivity. The approach and results are novel and interesting however several important points need to be further addressed to strengthen the manuscript.

Specific comments:

1) The authors use three types of cells in which the levels of OPA1 were reduced: OPA S545R, OPA Crisper, and OPA siRNA. What are the differences in OPA1 expression in all three lines? This is an important issue since the authors claim that PGS1 depletion could only rescue cells in which OPA1 was not totally depleted. If this is indeed the case, why does this occur?

2) Fig 3B: knockdown of PGS1 in OPA S545R increases mitochondrial hypertubulation, decreases normal morphology, and increases fragmentation. On the other hand, in Fig 3D, knockdown of PGS1 in OPA1 Crisper cells decreases fragmentation and increases normal morphology. What is the reason for these differences and do they mean that PGS1 depletion has additional effects on mitochondrial morphology besides suppressing fragmentation?

3) If indeed the major effect is on mitochondrial fragmentation, how does PGS1 depletion rescue mitochondrial fragmentation in hypomorphic OPA1 mutations? The authors show that PGS1 depletion rescues mitochondrial fragmentation by inhibiting mitochondrial fission, and not by restoring basal fusion. The authors examined the fusion and fission machinery proteins and found elevated total levels of DRP1 in Opa1Crisper MEFs, which returned to WT levels in Opa1CrisperPgs1Crisper MEFs (Fig 4A). This is a striking difference, however, there did not seem to be a difference in the mitochondrial levels of DRP1 between the two genotypes (Fig 4A,B). The authors should reassess (perhaps using additional methods) whether PGS1 depletion reduces DRP1 recruitment to the mitochondria. Is there a change in the levels of the active/inactive forms of DRP1 (pS616/pS637) between the genotypes? Is there a difference in DRP1 oligomerization?

4) It has been reported that changes in the lipid composition of the mitochondrial membranes regulated by the cardiolipin synthesis pathway regulate the recruitment of DRP1 to mitochondria: phosphatidic acid (PA) inhibits fission by reducing DRP1 recruitment to mitochondria, and PA can be converted to diacylglycerol (DAG) to promote DRP1 recruitment and mitochondrial fragmentation. Based on the results presented in Fig 5 the authors conclude that accumulation of PA in the inner membrane of the mitochondria (which can occur also because of PGS1 depletion) is responsible for the inhibition of mitochondrial fragmentation in a DRP1-dependent manner. The authors should perform phospholipidomics of the outer mitochondria membrane (OMM) to determine whether PGS1 depletion increases the levels of PA in the OMM, which would possibly reduce DRP1 recruitment to the mitochondria (or perhaps its activity), resulting in inhibition of fragmentation.

Referee #2 (Comments on Novelty/Model System for Author):

The model system is largely human fibroblast cell lines harboring pathogenic OPA1 variants. Given the phenotypic screen that has been developed relies on the integrity and morphology of these

cells, it would be helpful and informative to have details of the cell passage used in these experiments to know that these are consistent across the board. It's an assumption that primary and not immortalised cells are being utilised although this should be specifically mentioned.

Referee #2 (Remarks for Author):

Cretin, Wai and colleagues present an impressive set of data based on a high-throughput, confocal screening of mitochondrial morphology in which a mitochondrial proteome siRNA library has been used to identify proteins which can lengthen or shorten the dynamic mitochondrial network; through uncovering genetic modifiers of mitochondrial fragmentation the authors demonstrate a role for PGS1, a protein which plays a crucial role in mitochondrial cardiolipin biosynthesis, in being able to restore mitochondrial dynamics in primary human cells carrying pathogenic OPA1 gene variants.

The manuscript is well-written, the approach of wider interest given the integration of machine learning technology to the image capture and analysis, and the data beautifully presented.

I have few comments for the authors although the manuscript is lengthy and could benefit from shortening to focus on a clearer narrative for both the expert and non-expert reviewer.

Minor comments include the description of the OPA1 gene variants that present within the cellular models - please follow HGVS guidelines to refer to these as pathogenic variants not mutations, include a RefSeq when describing the variants and ensure that the correct nomenclature for the variants (c. and p.) are used; three-letter, not single-letter, abbreviations should be given in the text and tables.

It would be beneficial to have some further information about the source and phenotype of the cell lines used in the study (Some details given in the supplement). Please can you explicitly state whether these are primary or immortalised lines at an early point in the text. Given the fact that primary human fibroblast senesce following multiple passage, changing their morphology, some comment on the passage of the cells used in the experiments would be incredibly informative to ensure that subject and control cells were interrogated identically.

Lines 168-169, Figure EV1D: western blotting is semi-quantitative so it's a little tricky to place numbers and significance on the numerical data revealed a loss of steady state OPA1 protein (L- and S- isoforms); the data for the p.Ser545Arg cell line is not significant. Please rephrase these sentences.

Lines 269-272 - the implication that inhibition of cytosolic protein synthesis can affect mitochondrial fragmentation is interesting; why were these genes previously linked to mitochondria? some further explanation or discussion would be helpful. Moreover, the mitome screen described identifies a "wide array" of mitochondrial proteins (Figure 2E, Table S4) - what is the implication for this - do you think these are all important in being able to rescue the morphology phenotype in the OPA1-deficient patient cells?

The work investigating the link between OPA1 function and cardiolipin levels, with a focus on mitochondrial morphology, is very interesting, particularly the identification of PGS1 being able to redress the balance in OPA1 deficient cells, which is not necessarily linked to mtDNA copy number.

In conclusion, this is an impressive manuscript that makes a significant contribution to our understanding of this area of mitochondrial biology. Some focus on making a clearer message

through the Results and Discussion sections will help the general readership to fully appreciate what has been achieved.

Referee #3 (Remarks for Author):

In this study, Cretin et. al, developed an elegant high-throughput, supervised machine learning-based screen to discover - (1) novel regulators of mitochondrial morphology in wild-type cells; and (2) novel proteins that prevent mitochondrial fragmentation in OPA1 mutant fibroblasts. Through this screen targeting all known mitochondrial proteins, they identified numerous novel regulators of mitochondrial morphology, one of which is PGS1, an enzyme involved in the synthesis of cardiolipin. Interestingly, the authors found that co-knockdown of PGS1 in MEFs rescued the fragmentation phenotype observed following loss of OPA1, along with partially rescued mitochondrial function but not mtDNA depletion or apoptotic induction. While cardiolipin was reduced in OPA1 KD cells, co-knockdown of OPA1/PGS1 did not rescue cardiolipin levels; rather, it was genetic knockdown of other proteins of the cardiolipin synthesis pathway that revealed phosphatidic acid accumulation at the IMM to potentially have a role in the rescue of the mitochondrial defects observed in OPA1 KD cells.

The study is clear, well executed and well written. The imaging and its corresponding quantification is especially well presented. The work helps clarify the role and importance of existing modulators of mitochondrial morphology and also helps tease out the contributions played by Opa1 in the maintenance of mitochondrial morphology and function. I have only minor points that the authors could address:

1. In control human fibroblasts, you identify that PGS1 siRNA KD leads to a hypertubulated mitochondrial network. However, in MEFs PGS1 siRNA KD has no effect on mitochondrial fusion. Please explain how this difference can be interpreted.
2. Figure 6C/D (TEM images) - "Inner membrane structure of Opa1CRISPR Pgs1CRISPR was not restored to WT morphology." Despite this statement and others in the text, the representative image and the quantification of figure 6D does show at least a partial rescue. Therefore, this should be reflected in the text.
3. It would be useful to include a statistics section in the methods and outlining whether the data presented represents an average of all the cells or an average of each experiment (where appropriate).

Spelling errors/mistakes -

- Missing "C" label in Figure 2
- Spelling error, "deficienct", line 481
- Spelling error, "MFES" line 499
- Bars for Fis1 bands in Figure EV4B are missing

Cretin et al.

High-throughput screening identifies suppressors of mitochondrial fragmentation in OPA1 fibroblasts

EMM-2020-13579

EDITOR

"... Thank you for the submission of your manuscript to EMBO Molecular Medicine. We have now received feedback from the three reviewers who agreed to evaluate your manuscript. As you will see from the reports below, the referees acknowledge the interest and novelty of the study but also raise serious and partially overlapping concerns that should be addressed in a major revision. During the cross-commenting session it became clear that the phospholipidomics experiments suggested by the referee #1 (point #4) are not required and should rather be addressed in writing.

E-mail correspondence Dec 23rd 2020

We thank the three reviewers for their critical appraisal of our manuscript and for their constructive criticism and comments. We have addressed each comment and question raised by the reviewers below in point-by-point responses below.

REFEREE #1

In this manuscript, Cretin et al established a high-throughput imaging pipeline to perform quantitative mitochondria morphology analysis coupled with a siRNA library targeting the entire known mitochondrial proteome in fibroblasts. In DOA+ patient fibroblasts, the authors identified 91 candidate genes whose depletion prevents mitochondrial fragmentation, and focused on phosphatidyl glycerophosphate synthase (PGS1), which belongs to the cardiolipin synthesis pathway. Cretin et al demonstrated that PGS1 depletion rescues mitochondrial fragmentation in hypomorphic OPA1 mutations by inhibiting mitochondrial fission, which improves defective respiration, but does not rescue mtDNA depletion, cristae dysmorphology or apoptotic sensitivity. The approach and results are novel and interesting however several important points need to be further addressed to strengthen the manuscript.

Specific comments:

- 1) The authors use three types of cells in which the levels of OPA1 were reduced: OPA S545R, OPA Crisper, and OPA siRNA. What are the differences in OPA1 expression in all three lines?*

We thank the reviewer for their comment and fully agree with the importance of assessing OPA1 expression, which we address with existing and newly-added data. For steady-state protein levels, we previously performed western blots to measure the levels of OPA1 in primary human fibroblasts including those carrying the p.Ser545Arg pathogenic variant, which revealed a reduction of OPA1

protein of $21.5\% \pm 3.2$ in *OPA1*^{S545R} lysates (figure EV1D) that was not statistically significantly lower when compared to control fibroblasts CLT-1, CTL-2, and CTL-3 using a non-parametric ANOVA test (Kruskal-Wallis). This point was raised by reviewer 3 (see below). In fact, OPA1 protein levels were unchanged in all patient-derived fibroblasts except for those carrying the non-sense p.Gln297*(Q297X) pathogenic variant, which were lowered to $58.2\% \pm 9.2$ in *OPA1*^{Q297X} lysates (figure EV1D), which is expected given that the non-sense mutation would lead either to the degradation of a truncated protein or non-sense mediated decay of RNA.

For immortalized mouse embryonic fibroblasts carrying hypomorphic variants of *Opa1* (*Opa1*^{Crispr}) we observed an even more substantial reduction in Opa1 protein levels, which we had previously indicated in the text with the sentence beginning on Line 325 of the revised manuscript: “missense mutations causing a ~80% reduction in steady-state protein levels in *Opa1*^{Crispr} MEFs (Figure 3G, H) and a ~50% reduction in *Opa1* mRNA levels (figure EV3C)”. Finally, we have now added new data as requested demonstrating that siRNA-mediated depletion of Opa1 causes significant depletion of *Opa1* mRNA and protein levels (qRT-PCR data and immunoblot data demonstrating now shown in figure EV3D) and have therefore amended the sentence beginning on line 327 to reflect this :” ... which could also be effectively achieved by siRNA-mediated downregulation (figure EV3D)”. These new data are show below.

figure EV3D

(d) (left) quantitative RT-PCR (qRT-PCR) measurement of normalized *Opa1* expression (relative to *Gapdh*) WT MEFs treated with *Opa1* siRNA for 72 hours relative to NT (non-targeting) control. Data represent mean \pm SD of 4 independent experiments, unpaired t-test. (right) Equal amounts of protein extracted from WT MEFs treated with indicated siRNAs for 72 hours were separated by SDS-PAGE, immunoblotted for Opa1.

We observed mitochondrial fragmentation in both human and mouse fibroblasts mutated for OPA1 (Figures 1B,C) and Opa1 (Figure 3C, D), which could be further exacerbated by siRNA-mediated depletion of OPA1 (Figure 3A) in human fibroblasts and *Opa1*^{Crispr} MEFs (now figure EV3i, j).

Together, these data demonstrate that despite differential reduction of OPA1 and Opa1 protein caused by the initial human and mouse genetic perturbations, both *OPA1*^{S545R} patient fibroblasts and

Opa1^{Crispr} MEFs exhibit hypomorphic effects on mitochondrial dynamics. This statement is reflected in the sentence beginning on line 334 of the revised text, which reads “Similarly to hypomorphic *OPA1*^{S545R} patient-derived fibroblasts, *Opa1*^{Crispr} MEFs exhibited hypomorphy, as evidenced by the ability of *Opa1* siRNA treatment to further increase mitochondrial fragmentation (figure EV3i, j) to levels observed in *Opa1*^{KO} MEFs (figure EV3m, n) and the ability of *Opa1*^{Crispr} MEFs to undergo SiMH (figure EV3k, l)”.

2) This is an important issue since the authors claim that PGS1 depletion could only rescue cells in which OPA1 was not totally depleted. If this is indeed the case, why does this occur?

We fully agree. Indeed, siRNA-mediated depletion of OPA1 in *OPA1*^{S545R} fibroblasts or *Opa1* in *Opa1*^{Crispr} MEFs, abolishes the restorative effects of PGS1/Pgs1 depletion on mitochondrial morphology, as evidenced by the data initially presented in human (Figures 3A,B) and mouse (figure EV3i, j) fibroblasts. We argue this is because of the severity of the ablation of *Opa1*. We have further extended these findings by analyzing a new, hypomorphic *OPA1* HeLa cell line (*OPA1*^{CRISPR}) whose *OPA1* steady-state protein expression is reduced by 62% according to immunoblot analysis. As we had observed in human and mouse fibroblasts deficient for *OPA1/Opa1*, our newly added data showed that PGS1 depletion in *OPA1*^{CRISPR} HeLa cells was able to suppress mitochondrial fragmentation (see figure EV2l-n, below).

figure EV2l-n cont:

(l) Equal amounts of protein extracted from WT and *OPA1*^{CRISPR} HeLa were separated by SDS-PAGE, immunoblotted with OPA1 antibody and quantified by densitometry relative to tubulin

(m) Representative confocal images of wild type (WT) and *OPA1*^{CRISPR} HeLa cells treated with indicated siRNAs for 72 hours. Live imaging of mitochondria labelled with TMRE (Mitochondria, green), and NucBlue (Nuclei, blue). Scale bar=50um.

(n) Supervised ML mitochondrial morphology quantification of (m) using WT HeLa cells treated with *OPA1* siRNA (fragmented), NT siRNA (normal), or *DNM1L* siRNA (hypertubulated) training sets. Data represent mean ± SD of 8 replicates, One-way ANOVA.

Since Pgs1 depletion did not restore mitochondrial morphology in *Opa1*^{KO} MEFs according to data originally presented in figure EV5, we hypothesized that the severity of the mitochondrial fragmentation defect could define the ability for Pgs1 silencing to rescue network morphology. To further test this, **we performed new experiments** in which we downregulated Pgs1 in MEFs with a fragmented mitochondrial network due to a deletion of the *i*-AAA protease *Yme1l* (new figures EV5c-e). *Yme1l* deletion in MEFs leads to the hyperactivation of its proteolytic substrate Oma1, resulting in the accelerated cleavage of L-Opa1 that promotes increased mitochondrial fission in the face of normal inner membrane fusion¹. We performed live-cell imaging of WT and *Yme1l*^{-/-} MEFs labeled with membrane potential-dependent (TMRE, orange), and -independent (Mitotracker DeepRed) fluorescent probes. *Yme1l*^{-/-} MEFs exhibited a degree of mitochondrial fragmentation comparable to *Opa1*^{Crispr} MEFs, which could be suppressed by depletion of *Dnm1l* but not of *Pgs1*. These new data are shown below.

figure EV 5 cont:

(c) Representative confocal images of wild type (WT) and *Yme1l*^{-/-} MEFs treated with indicated siRNAs for 72 hours. Live imaging of mitochondria labelled with Mitotracker DeepRed (MTDR, green), TMRE (pink) and NucBlue (Nuclei, blue). Scale bar=10μm.

(d) Supervised ML mitochondrial morphology quantification of (c) using WT MEFs treated with *Opa1* siRNA (fragmented), NT siRNA (normal), or *Dnm1l* siRNA (hypertubular) training sets. (

(e) Quantification of mitochondrial membrane potential (TMRE/(MTDR*Cell area) of cells imaged in (c). Number of analyzed cells inset. Data represent mean ± SD, One-way ANOVA.

These data are consistent with our observations that mitochondrial fragmentation in *OPA1*^{S545R} fibroblasts treated with *OPA1* siRNA can be rescued by DNM1L depletion but not PGS1 depletion (Figure 3A, B). Taken together, our existing and newly-added data argue strongly that hypomorphic effects on mitochondrial morphology caused by deleterious or pathogenic genetic lesions in *OPA1* can be rescued by Pgs1 depletion.

3) *Fig 3B: knockdown of PGS1 in OPA S545R increases mitochondrial hypertubulation, decreases normal morphology, and increases fragmentation. On the other hand, in Fig 3D, knockdown of PGS1 in OPA1 Crispr cells decreases fragmentation and increases normal morphology. What is the reason for these differences and do they mean that PGS1 depletion has additional effects on mitochondrial morphology besides suppressing fragmentation?*

We thank the reviewer for raising this point, which was also raised by reviewer 3 (see below). While the depletion of PGS1/Pgs1 generally promotes mitochondrial elongation in *OPA1*/*Opa1* deficient cells, we wondered whether the modest divergence pointed out by the reviewer between human *OPA1*^{S545R} fibroblasts and mouse *Opa1*^{Crispr} fibroblasts was due to species differences or the fact that *OPA1*^{S545R} fibroblasts are primary cells and *Opa1*^{Crispr} fibroblasts are immortalized (by SV40 large T antigen). To explore this possibility, we generated new, hypomorphic *OPA1* HeLa cells by Crispr/Cas9 targeting of Exon 18 (which is the same domain mutated in *OPA1*^{S545R} patient-derived fibroblasts) that created a heterozygous truncating mutation (c.1763delTGCTTTTGAAAATGGTAC, p.Glu637*) leading to a 62% reduction of *OPA1* protein, as mentioned above (new figure EV2I). Depletion of *PGS1* by siRNA in *OPA1*^{CRISPR} HeLa cells rescued mitochondrial morphology without increasing mitochondrial fragmentation (figure EV2m, n). Moreover, depletion of PGS1 in wild type HeLa cells did not significantly hypertubulate the mitochondrial network. Therefore, like MEFs, HeLa cells respond similarly to *OPA1* and/or PGS1 depletion, leading us to posit that the reason for the aforementioned differences could be due to differences between slow-growing, primary cells (i.e. human fibroblasts) versus highly proliferative, immortalized cells (MEFs, HeLa) and not due to species differences as we had initially speculated.

As the reviewer insinuated, depletion of Pgs1 clearly has a number of effects on mitochondria that go beyond the modulation of mitochondrial morphology. For example, Pgs1-deficient MEFs (*Pgs1*^{Crispr}) exhibit increased mtDNA content, increased respiration, increased membrane potential, and increased apoptotic resistance triggered with either ABT-737+Actinomycin D (Figure 6), Etoposide, or Staurosporine (figure EV6). Of course, Pgs1 deficiency also reduces cardiolipin (CL) levels because the generation of PGP by Pgs1 is required for CL synthesis. Thus, beyond the novel discovery of the role of Pgs1 in mitochondrial morphology regulation our work in *Pgs1*^{Crispr} MEFs, has

also uncovered the protective effects of Pgs1 depletion for mitochondrial and cellular stress responses.

4) *If indeed the major effect is on mitochondrial fragmentation, how does PGS1 depletion rescue mitochondrial fragmentation in hypomorphic OPA1 mutations?*

As the reviewer rightly pointed out in their review “**The authors show that PGS1 depletion rescues mitochondrial fragmentation by inhibiting mitochondrial fission, and not by restoring basal fusion**”. We argue that accumulation of PA is responsible for the anti-fission effect, and this is further discussed below in response to the reviewer’s critique.

5) *The authors examined the fusion and fission machinery proteins and found elevated total levels of DRP1 in Opa1Crisper MEFs, which returned to WT levels in Opa1CrisperPgs1Crisper MEFs (Fig 4A). This is a striking difference, however, there did not seem to be a difference in the mitochondrial levels of DRP1 between the two genotypes (Fig 4A,B). The authors should reassess (perhaps using additional methods) whether PGS1 depletion reduces DRP1 recruitment to the mitochondria. Is there a change in the levels of the active/inactive forms of DRP1 (pS616/pS637) between the genotypes? Is there a difference in DRP1 oligomerization?*

We thank the reviewer for their suggestions and have added **four new data sets** obtained using a variety of methods to further clarify the impact of Opa1 depletion on Drp1 in MEFs.

First, we have performed new western blots to further substantiate this striking increase in steady-state levels of Drp1 in *Opa1^{Crispr}* MEFs. Indeed, we observed an ~1.8 fold increase in Drp1 protein levels in whole cell lysates of *Opa1^{Crispr}* MEFs. These data are now represented in figure EV4g (see below).

Second, to further substantiate the absence of altered subcellular distribution of Drp1 in *Opa1^{Crispr}* MEFs that we originally reported by indirect immunofluorescence in Figure 4B, we used Crispr/Cas9 genome editing to knock in mTurquoise2 upstream of the Dnm1l locus to generate an endogenously, fluorescently tagged Drp1 in both WT and *Opa1^{Crispr}* MEFs. The knockin strategy was modeled on the YFP-Drp1 construct devised by the Youle group, which included a linker region separating mTurquoise (instead of GFP since the MEFs we use stably express mitoYFP) from the start methionine and is now described in the Materials and Methods section of the revised manuscript. We then performed live confocal microscopic assessment of mTurquoise2-Drp1 and mitoYFP and observed no significant differences in subcellular Drp1 distribution between WT and *Opa1^{Crispr}* MEFs. **These new data are represented now in figures EV4d, e** and are shown below. Importantly, additional depletion of *Pgs1* by siRNA in both WT and *Opa1^{Crispr}* MEFs also did not alter the cellular

distribution of mTurquoise2-Drp1. While we did observe less (non-mitochondrial) mTurquoise2-Drp1 in the cytosol than using classical indirect immunocytochemistry approaches to monitor Drp1 as we initially reported in Figure 4B (which we may attribute to the tagging of Drp1), these data nevertheless support the the absence of altered Drp1 recruitment in *Opa1*^{Crispr} MEFs.

figure EV 4 cont:

(d) Representative confocal images of MEFs knocked in for mTurquoise2-Dnm1l by Crispr/Cas9 genome editing in WT (*WT*^{Drp1KI}) and *Opa1*^{Crispr} (*Opa1*^{Crispr-Drp1KI}) MEFs treated with non-targeting (NT) or *Pgs1* siRNA for 72 hours. Drp1 (mTurquoise2, purple), mitochondria (mitoYFP, green). Scale bar=20um.

(e) Bar graph representation of Drp1 localized to mitochondria (green) vs cytosol (blue). Data represent mean ± SD of 5 replicates, (193-1062 cells per cell line), One-way ANOVA.

Third, to assess the oligomerization capacity of endogenous Drp1 in WT, *Opa1*^{Crispr}, *Opa1*^{Crispr} *Pgs1*^{Crispr}, and *Pgs1*^{Crispr} MEFs, we performed crosslinking studies using 10mM 1,6-bismaleimideohexane (BMH) and immunoblot analysis as had been previously described in mouse and human cells²⁻⁴ to assess the oligomerization of Drp1. Consistent with our existing and newly added data (figure EV4d-g), we did not observe any impairment in Drp1 oligomerization in WT and *Opa1*^{Crispr} MEFs. These new data are now represented in figure EV4h.

figure EV 4 cont:

(h) Equal amount of post-nuclear supernatant +/- cross-linked with 10mM BMH from MEFs of the indicated genotypes were separated by SDS-PAGE and immunoblotted with Drp1 antibody. * and ** indicate Drp1 monomers and Drp1 complexes, respectively. The ratio of Drp1 complexes over Drp1 monomers was quantified by densitometry relative to Stain-Free.

Forth, we performed immunoblot analysis of MEFs to assess the phosphorylation status of Drp1 at the most widely studied amino acid residues: S579 (which corresponds to human S616) and S600 (which corresponds to human S637) using commercially available phosphorylation-specific antibodies and compared these levels to total Drp1 levels in the cell. To validate the specificity of these antibodies, we pre-treated WT cells with either forskolin (PKA activator) or CalyculinA (Serine/Threonine phosphatase inhibitor) and performed immunoblot analysis (figure EV4f), which lead to the characteristic, previously described decrease and increase in phosphorylation of Drp1 respectively^{5,6}. The specificity of these signals was further confirmed by the absence of immunoreactivity in *Drp1^{Crispr}* MEFs lacking Drp1 (figure EV4f). We observed an increased level of S579 but not S600 phosphorylation in *Opa1^{Crispr}* MEFs, which was restored back to WT levels in *Opa1^{Crispr} Pgs1^{Crispr}* MEFs. Given the specific increase in S579 phosphorylation of Drp1 and the controversial role of S600 (human S637 phosphorylation) for Drp1 recruitment⁵⁻⁸, we decided to pursue more quantitative immunoblotting analyses of S579 in WT, *Opa1^{Crispr}*, *Opa1^{Crispr} Pgs1^{Crispr}*, and *Pgs1^{Crispr}* MEFs (figure EV4g). Indeed, we could demonstrate an increase in S579 levels in *Opa1^{Crispr}* MEFs which was subsequently rescued in *Opa1^{Crispr} Pgs1^{Crispr}* MEFs, but when normalizing to Drp1 levels, we observed

no significant differences in pDrp1^{S579} to total Drp1 ratios. Together, these four new sets of data further confirm the findings that were initially presented (Figure 4A, B) and further support our conclusion that despite increased steady state levels of Drp1 and pro-fission phosphorylation of Drp1 at S579 observed in *Opa1*^{Crispr} MEFs, this does not lead to an increased mitochondrial recruitment of Drp1.

figure EV 4 cont:
(f-g) Equal amounts of protein extracted from MEFs were separated by SDS-PAGE, immunoblotted with the indicated antibody and quantified by densitometry relative to Stain-Free or Ponceau. Data represent mean ± SD of three independent experiments, One-way ANOVA.

6) *It has been reported that changes in the lipid composition of the mitochondrial membranes regulated by the cardiolipin synthesis pathway regulate the recruitment of DRP1 to mitochondria: phosphatidic acid (PA) inhibits fission by reducing DRP1 recruitment to mitochondria, and PA can be converted to diacylglycerol (DAG) to promote DRP1 recruitment and mitochondrial fragmentation. Base on the results presented in Fig 5 the authors conclude that accumulation of PA in the inner membrane of the mitochondria (which can occur also because of PGS1 depletion) is responsible for the inhibition of mitochondrial fragmentation in a DRP1-dependent manner.*

We fully agree with the reviewer's proposition that PA accumulation in the IMM caused by the depletion of PGS1 could potentially lead to an accumulation of PA at the outer mitochondrial membrane (OMM). Based on our current understanding of PA accumulation at the OMM, largely derived from seminal work from the Frohman and Sesaki groups⁹⁻¹¹, we would therefore expect Drp1 recruitment to the OMM to be increased due to the lipophilic interactions. Yet, our existing and new data (see previous response #5 above) do not support altered Drp1 recruitment in Pgs1-deficient cells. This of course could mean that PA accumulation is intramitochondrial (e.g. at the IMM) and not at the OMM.

7) *The authors should perform phospholipidomics of the outer mitochondria membrane (OMM) to determine whether PGS1 depletion increases the levels of PA in the OMM, which would possibly reduce DRP1 recruitment to the mitochondria (or perhaps its activity), resulting in inhibition of fragmentation.*

Phospholipidomic profiling of isolated OMMs requires highly pure mitochondria that devoid of any membranes of other organelle, especially of ER. However, we have never successfully obtained adequately pure mitochondria from MEFs, which is presumably due to the low mitochondrial mass and stronger mito-ER contacts in these cells. Nevertheless, PA is rapidly converted to other lipid species in and at mitochondria, which is why its levels are so characteristically low in intact, purified mitochondria¹². Indeed, we have never observed the accumulation of PA after knockdown of the PA transporter (PRELID1-TRIAP1) complex within the IMS¹², indicating that PA can be distributed between the OMM and other membranes and be metabolized into other lipid species outside of mitochondria. Therefore, we cannot expect a bulk accumulation of PA at the OMM which can be detected by lipidomic analysis.

Despite these existing technical limitations, we decided to assess OMM accumulation of PA using a fluorescent PA reporter¹³, which is an approach that had previously been used to demonstrate an increase in PA levels at the OMM in cells over-expressing the PA producing enzyme mitoPLD¹⁰. We transfected the PA-binding reporter construct pCMV-mRFP-PASS (or the mutant pCMV-mRFP-PASS-4E version lacking PA binding capacity) into WT and *Pgs1*^{Crispr} MEFs and assessed the colocalization of mRFP (PA sensor) to mitoYFP (mitochondria). We observed a clear cytosolic distribution of mRFP in both cell lines (figure EV5h), demonstrating that Pgs1 depletion does not cause an appreciable increase in PA levels at the OMM. Taken together, these data indicate that Pgs1 depletion does not cause a measurable increase in OMM PA levels and argues for the intramitochondrial accumulation of PA.

figure EV 5 cont:

(h) Representative confocal images of wild type (WT) and *Pgs1*^{Crispr} MEFs transfected with wild type (RFP-PASS) or mutant (RFP-PASS-4E) PA sensors 48 hours. Live imaging of mitochondria labelled with mitoYFP (green), RFP (orange) and NucBlue (Nuclei, blue) revealed no recruitment of PA sensors to mitochondria. Scale bar=50um.

REFEREE #2

1) *The model system is largely human fibroblast cell lines harboring pathogenic OPA1 variants. Given the phenotypic screen that has been developed relies on the integrity and morphology of these cells, it would be helpful and informative to have details of the cell passage used in these experiments to know that these are consistent across the board. It's an assumption that primary and not immortalised cells are being utilised although this should be specifically mentioned.*

We thank the reviewer for their appraisal and their note. Indeed, the human fibroblasts used in this study are primary skin fibroblasts, derived from patients either in France or the UK as indicated in Table 1. We have added additional information regarding the passage number of the human fibroblasts used in this study in the materials and methods (Line 775-778) and in the text (Line 1167-1168, 1173, 1178, 1192, 1209, 1223-1224).

2) *The manuscript is well-written, the approach of wider interest given the integration of machine learning technology to the image capture and analysis, and the data beautifully presented. I have few comments for the authors although the manuscript is lengthy and could benefit from shortening to focus on a clearer narrative for both the expert and non-expert reviewer.*

We have shortened the manuscript to achieve a clearer narrative.

3) Minor comments include the description of the OPA1 gene variants that present within the cellular models - please follow HGVS guidelines to refer to these as pathogenic variants not mutations, include a RefSeq when describing the variants and ensure that the correct nomenclature for the variants (c. and p.) are used; three-letter, not single-letter, abbreviations should be given in the text and tables.

We thank the review for pointing this out. For the human genetic variants in *OPA1*, we have made the necessary changes according to HGVS guidelines and have made the requested modifications in the text and tables. Specifically, we have added a paragraph in the Methods section of the revised manuscript to clearly define the nomenclature, and we have specified the reference sequence in Table 1 and in its legend. Furthermore, to maintain historical compatibility and facilitate understanding with recent publications, variants are described according to the two transcript variants frequently used. Finally, the name of all the variations were double-checked with the Mutalyzer Name Checker (<https://mutalyzer.nl>).

Moreover, we have corrected the term “mutation” to “pathogenic variant” throughout the text where appropriate.

4) It would be beneficial to have some further information about the source and phenotype of the cell lines used in the study (Some details given in the supplement). Please can you explicitly state whether these are primary or immortalised lines at an early point in the text. Given the fact that primary human fibroblast senesce following multiple passage, changing their morphology, some comment on the passage of the cells used in the experiments would be incredibly informative to ensure that subject and control cells were interrogated identically.

We thank the reviewer for their appraisal and their note. Indeed, the human fibroblasts used in this study are primary skin fibroblasts, derived from patients either in France or the UK. This information was originally presented in Table 1, which included clinical description of the patients and studies in which these mutations and cell lines were initially reported. Now, we have added additional information regarding the passage number of the human fibroblasts used in this study in the materials and methods (Line 775-778) and in the text (Line 1167-1168, 1173, 1178, 1192, 1209, 1223-1224). and have now explicitly stated, where appropriate, that these are indeed non-senescent primary human fibroblasts. With the exception of the Mitome siRNA screens (Figure 2), all patient and control human fibroblast experimentation was performed in parallel using 3 different control fibroblasts (2 from the UK and 1 from France).

Nevertheless, given the challenges of working with primary human fibroblasts listed described in the text, we decided to perform mechanistic interrogations in immortalized mouse embryonic fibroblasts, which do not senesce nor change their mitochondrial morphology during successive passages.

5) Lines 168-169, Figure EV1D: western blotting is semi-quantitative so it's a little tricky to place numbers and significance on the numerical data revealed a loss of steady state OPA1 protein (L- and S- isoforms); the data for the p.Ser545Arg cell line is not significant. Please rephrase these sentences.

We apologize for this oversight. We have corrected this error in the text and the sentence now reads : *“Western blot analyses revealed a reduction of OPA1 protein of 58.2% ± 9.2 in OPA1^{Q297X} lysates (figure EV1d) relative to control fibroblasts and no significant differences in other patient-derived fibroblasts”*.

6) Lines 269-272 - the implication that inhibition of cytosolic protein synthesis can affect mitochondrial fragmentation is interesting; why were these genes previously linked to mitochondria? some further explanation or discussion would be helpful.

Indeed, the treatment of human and mouse fibroblasts with the cytosolic protein synthesis inhibitor cycloheximide clearly triggers (stress-induced) mitochondrial hyperfusion, which is consistent with previous studies¹⁴⁻¹⁶ cited within the text.

The genes *RPL15*, *RPS15A*, *RPLP2*, *RPL36AL*, *RPL5*, and *RPS18* were previously linked to mitochondria according to the Mitominer 4.0 database¹⁷, which uses a variety of empirical and in silico resources. The table below illustrates some of the predictive outputs for these genes.

Gene Symbol	Mito Evidence				Mito Targeting Seq			
	Mass-Spec Studies	IMPI	MitoCarta	IMPI score	iPSORT	MitoProt	TargetP	Mito Fates
RPL15	2	Predicted NOT mitochondrial	FALSE	0.794098449	1	0.412	0.509	0.443
RPL36AL	0	Predicted NOT mitochondrial	FALSE	0.726994518	1	0.4243	0.295	0.013
RPL5	4	Predicted mitochondrial	FALSE	0.886808154	0	0.9413	0.285	0.012
RPLP2	4	Predicted mitochondrial	FALSE	0.971741642	0	0.371	0.217	0
RPS15A	3	Predicted	TRUE	0.980925335	1	0.1009	0.355	0.426

		mitochondrial						
RPS18	3	Predicted mitochondrial	TRUE	0.999995664	1	0.8858	0.249	0.209

To more precisely describe the manner in which these genes have been associated to mitochondria, we have modified the sentence that initially read “*our data revealed a cluster of ribosomal genes previously linked to mitochondria ...*” to now read “*Like in control fibroblasts, our data revealed a cluster of ribosomal genes **bioinformatically predicted to be targeted to mitochondria** according to the Integrated Mitochondrial Protein Index (IMPI) score of the Mitominer 4.0 database*” beginning on line 276 of the revised text.

7) *Moreover, the mitome screen described identifies a "wide array" of mitochondrial proteins (Figure 2E, Table S4) - what is the implication for this - do you think these are all important in being able to rescue the morphology phenotype in the OPA1-deficient patient cells?*

Indeed, as we initially wrote in the discussion “These genes cover various classes of mitochondrial functions including mitochondrial gene expression, oxidative phosphorylation, and amino acid metabolism yet how these genes (Table S4) or processes (figure EV2h, i) influence mitochondrial dynamics is unclear and warrants further investigation.” The implication is that depletion of these genes in *OPA1^{S545R}* human fibroblasts rescues aberrant mitochondrial morphology either by indirectly reducing fission and/or increasing fusion acting either at or inside mitochondria or perhaps by affecting upstream processes like ER stress, cytosolic translation, or membrane contacts. Beyond PGS1, which is the subject of investigation of the current study, systematic and in-depth experimentation of the other 90 candidates will be important to determine how they are able to suppress mitochondrial fragmentation caused by OPA1 deficiency. Nevertheless, to satisfy the reviewer’s curiosity, as well as our own, we performed new experiments on two candidate genes identified in the Mitome screen: *DNAJC4* and *DNAJC11*, which are proteins that have been associated with the MICOS complex. As predicted from the results of the primary Mitome screen, we observed that siRNA-mediated depletion of either *DNAJC4* or *DNAJC11* in *OPA1^{S545R}* patient-derived fibroblasts were able to individually rescue mitochondrial fragmentation. These new data are displayed below and have been included as supplemental data in EV2h, i.

figure EV2 cont

(h) Representative confocal images and **(i)** mitochondrial morphology quantification of control (CTL-1) fibroblasts and *OPA1*^{S545R} patient fibroblasts treated with indicated siRNAs for 72 hours. Supervised ML training performed on cells with fragmented (*OPA1* siRNA), normal (non-targeting NT siRNA), and hypertubulated (*DNM1L* siRNA) mitochondria. Data represent mean ± SD of one independent experiments, (419 to 1783 cells).

8) *In conclusion, this is an impressive manuscript that makes a significant contribution to our understanding of this area of mitochondrial biology. Some focus on making a clearer message through the Results and Discussion sections will help the general readership to fully appreciate what has been achieved.*

We thank the reviewer for their appraisal and general suggestion, and have made a number of edits to the text to allow for a more concise and focused delivery of the message.

REFEREE #3

The study is clear, well executed and well written. The imaging and its corresponding quantification is especially well presented. The work helps clarify the role and importance of existing modulators of mitochondrial morphology and also helps tease out the contributions played by Opa1 in the maintenance of mitochondrial morphology and function. I have only minor points that the authors could address:

- In control human fibroblasts, you identify that PGS1 siRNA KD leads to a hypertubulated mitochondrial network. However, in MEFs PGS1 siRNA KD has no effect on mitochondrial fusion. Please explain how this difference can be interpreted.*

We thank the reviewer 3 for raising a very similar point raised by reviewer 1 regarding the hypertubulation that can be triggered by depletion of PGS1 in control primary human fibroblasts but

not in wild type MEFs. As mentioned above, we have now added new data demonstrating that depletion of *PGS1* by siRNA in wild type HeLa cells did not significantly hypertubulate the mitochondrial network (figure EV2h, i). Thus, HeLa cells respond similarly to *PGS1* depletion as compared to MEFs, leading us to posit that the reason for the aforementioned differences are due to differences between slow-growing, primary cells (human fibroblasts) versus highly proliferative, immortalized cells (MEFs, HeLa) and not species differences. The current working model, supported by our data and explicitly by comments made by reviewer 1 above, is that PA accumulation caused by the depletion of *PGS1* is responsible for the anti-fission effect on mitochondrial morphology, which we posit is specific to the IMM. Of course, we also show that *PGS1* depletion causes a reduction in CL levels in mitochondria (Figure 5) so we might speculate that CL depletion caused by *PGS1* depletion may also negatively impact heterotypic IMM fusion, as previously suggested by Ban et al.¹⁸, perhaps to differing degrees depending on the endogenous production of CL and possible cell-type specific expression of CL producing enzymes, which could counteract the anti-fission effect of PA accumulation leading to a seemingly normal mitochondrial morphology in wild type cell lines like HeLa and MEFs.

2. *Figure 6C/D (TEM images) - "Inner membrane structure of Opa1CRISPR Pgs1CRISPR was not restored to WT morphology." Despite this statement and others in the text, the representative image and the quantification of figure 6D does show at least a partial rescue. Therefore, this should be reflected in the text.*

We have modified the text accordingly and the sentence now reads “not restored to WT morphology, **despite a modest increase in cristae number and OMM/IMM ratios**, indicating that mitochondrial morphology and cristae organization are largely uncoupled in these cells” (on line 538 of the revised text).

3. *It would be useful to include a statistics section in the methods and outlining whether the data presented represents an average of all the cells or an average of each experiment (where appropriate).*

We have now added a “Statistical analysis” section in the material and methods (see Line 1157-1161). Data represents an average of each experiment (described as “mean ± SD or SEM” in the figure legends) except for figure EV4f where the specification Line 172 “Data represent mean of all the cells ± SEM of four independent experiments, One-way ANOVA” was added.

4. Spelling errors/mistakes -

- Missing "C" label in Figure 2

This error has been corrected

- Spelling error, "deficienct", line 481

This error has been corrected

- Spelling error, "MFEs" line 499

This error has been corrected

- Bars for Fis1 bands in Figure EV4B are missing

This error has been corrected

References

1. Anand, R. *et al.* The i-AAA protease YME1L and OMA1 cleave OPA1 to balance mitochondrial fusion and fission. *J. Cell Biol.* **204**, 919–929 (2014).
2. Prudent, J. *et al.* MAPL SUMOylation of Drp1 Stabilizes an ER/ Mitochondrial Platform Required for Cell Death. *Mol. Cell* **59**, 941–955 (2015).
3. Karbowski, M., Neutzner, A. & Youle, R. J. The mitochondrial E3 ubiquitin ligase MARCH5 is required for Drp1 dependent mitochondrial division. *J. Cell Biol.* **178**, 71–84 (2007).
4. Otera, H., Miyata, N., Kuge, O. & Mihara, K. Drp1-dependent mitochondrial fission via MiD49/51 is essential for apoptotic cristae remodeling. *J. Cell Biol.* **212**, 531–544 (2016).
5. Cribbs, J. T. & Strack, S. Reversible phosphorylation of Drp1 by cyclic AMP-dependent protein kinase and calcineurin regulates mitochondrial fission and cell death. *EMBO reports* **8**, 939–944 (2007).
6. Yu, R. *et al.* The phosphorylation status of Ser-637 in dynamin-related protein 1 (Drp1) does not determine Drp1 recruitment to mitochondria. *Journal of Biological Chemistry* **294**, 17262–17277 (2019).
7. Chang, C.-R. & Blackstone, C. Cyclic AMP-dependent protein kinase phosphorylation of Drp1 regulates its GTPase activity and mitochondrial morphology. *J Biol Chem* **282**, 21583–21587 (2007).
8. Cereghetti, G. M. *et al.* Dephosphorylation by calcineurin regulates translocation of Drp1 to mitochondria. *PNAS* **105**, 15803–15808 (2008).
9. Adachi, Y. *et al.* Coincident Phosphatidic Acid Interaction Restrains Drp1 in Mitochondrial Division. *Mol. Cell* **63**, 1034–1043 (2016).
10. Baba, T. *et al.* Phosphatidic Acid (PA)-preferring Phospholipase A1 Regulates Mitochondrial Dynamics. *Journal of Biological Chemistry* **289**, 11497–11511 (2014).
11. Huang, H. *et al.* piRNA-Associated Germline Nuage Formation and Spermatogenesis Require MitoPLD Profusogenic Mitochondrial-Surface Lipid Signaling. *Developmental Cell* **20**, 376–387 (2011).
12. Potting, C. *et al.* TRIAP1/PRELI Complexes Prevent Apoptosis by Mediating Intramitochondrial Transport of Phosphatidic Acid. *Cell Metabolism* **18**, 287–295 (2013).
13. Zhang, F. *et al.* Temporal Production of the Signaling Lipid Phosphatidic Acid by Phospholipase D2 Determines the Output of Extracellular Signal-Regulated Kinase Signaling in Cancer Cells. *Molecular and Cellular Biology* **34**, 84–95 (2014).
14. Tondera, D. *et al.* SLP-2 is required for stress-induced mitochondrial hyperfusion. *The EMBO Journal* **28**, 1589–1600 (2009).
15. Wai, T. *et al.* The membrane scaffold SLP2 anchors a proteolytic hub in mitochondria containing PARL and the i-AAA protease YME1L. *EMBO Rep.* **17**, 1844–1856 (2016).

16. Lebeau, J. *et al.* The PERK Arm of the Unfolded Protein Response Regulates Mitochondrial Morphology during Acute Endoplasmic Reticulum Stress. *Cell Rep* **22**, 2827–2836 (2018).
17. Smith, A. C. & Robinson, A. J. MitoMiner v4.0: an updated database of mitochondrial localization evidence, phenotypes and diseases. *Nucleic Acids Res.* **47**, D1225–D1228 (2019).
18. Ban, T. *et al.* Molecular basis of selective mitochondrial fusion by heterotypic action between OPA1 and cardiolipin. *Nat. Cell Biol.* **19**, 856–863 (2017).

25th Mar 2021

Dear Dr. Wai,

Thank you for the submission of your revised manuscript to EMBO Molecular Medicine. I am pleased to inform you that we will be able to accept your manuscript pending the following final amendments:

1) Figures: Please upload individual, high-resolution file for each main and EV figure. All panels of a figure should fit on one file page. Place EV figure legends in the main manuscript file. For more information on figure presentation please check "Author Guidelines".

<https://www.embopress.org/page/journal/17574684/authorguide#datapresentationformat>

2) Movies: Rename movie files to "Movie EV1" etc. (also in the text) and zipp their legends as a .doc file with respective movie file.

3) Tables: Please move Table 1 to the main manuscript file. Tables S1-5 and computer script file should be submitted as Dataset files. Files should be renamed to "Dataset EV1" etc. Each file should have a title and a short description within the file, for .xls files in a separate tab. Please also change callouts for datasets in the manuscript text.

4) In the main manuscript file, please do the following:

- Add up to 5 keywords.
- Remove text colour.
- Make sure that all special characters display well.
- Add callout for figure 5F.
- In M&M, statistical paragraph should reflect all information that you have filled in the Authors Checklist, especially regarding randomization, blinding, replication.
- In M&M, include a statement that informed consent was obtained from all human subjects and that the experiments conformed to the principles set out in the WMA Declaration of Helsinki and the Department of Health and Human Services Belmont Report.
- Place conflict of interest statement in a separate paragraph and name it "Conflict of interest".
- Add author contributions. The nature of every author's contribution must be specified both in the manuscript submission system (using the CRediT contributor role taxonomy) and in the manuscript under the heading "Author Contributions". Please check "Author Guidelines" for more information.

<https://www.embopress.org/page/journal/17574684/authorguide#availabilityofpublishedmaterial>

- Correct the reference citation in the text and reference list. In the text of the manuscript, a reference should be cited by author and year of publication. Include a space between a word and the opening parenthesis of the reference that follows. In the reference list, citations should be listed in alphabetical order. Where there are more than 10 authors on a paper, 10 will be listed, followed by "et al.". Please check "Author Guidelines" for more information.

<https://www.embopress.org/page/journal/17574684/authorguide#referencesformat>

5) The Paper Explained: Please provide "The Paper Explained" and add it to the main manuscript text. Please check "Author Guidelines" for more information.

<https://www.embopress.org/page/journal/17574684/authorguide#researcharticleguide>

6) Synopsis:

- Synopsis image: Please resize the visual abstract and submit it as a high-resolution jpeg file 550

px-wide x (250-400)-px high.

- Synopsis text: Please provide a short stand first (maximum of 300 characters, including space) as well as 2-5 one sentence bullet points that summarise the paper as a .doc file. Please write the bullet points to summarise the key NEW findings. They should be designed to be complementary to the abstract - i.e. not repeat the same text. We encourage inclusion of key acronyms and quantitative information (maximum of 30 words / bullet point). Please use the passive voice.

7) For more information: There is space at the end of each article to list relevant web links for further consultation by our readers. Could you identify some relevant ones and provide such information as well? Some examples are patient associations, relevant databases, OMIM/proteins/genes links, author's websites, etc...

8) As part of the EMBO Publications transparent editorial process initiative (see our Editorial at <http://embomolmed.embopress.org/content/2/9/329>), EMBO Molecular Medicine will publish online a Review Process File (RPF) to accompany accepted manuscripts. This file will be published in conjunction with your paper and will include the anonymous referee reports, your point-by-point response and all pertinent correspondence relating to the manuscript. Let us know whether you agree with the publication of the RPF and as here, if you want to remove or not any figures from it prior to publication. Please note that the Authors checklist will be published at the end of the RPF.

9) Please provide a point-by-point letter INCLUDING my comments as well as the reviewer's reports and your detailed responses (as Word file).

I look forward to reading a new revised version of your manuscript as soon as possible.

Yours sincerely,

Zeljko Durdevic

***** Reviewer's comments *****

Referee #1 (Remarks for Author):

Great Effort! Congratulations!

Referee #2 (Comments on Novelty/Model System for Author):

nothing else to add at this stage

Referee #2 (Remarks for Author):

I find this to be an impressive manuscript that is now much clearer to both readers with expertise and the non-expert reader; the authors have responded in detail to the comments I made in review to provide a more concise and clearer narrative.

The authors performed the requested changes.

We are pleased to inform you that your manuscript is accepted for publication and is now being sent to our publisher to be included in the next available issue of EMBO Molecular Medicine.

Corresponding Author Name: Timothy WAI
Journal Submitted to: EMBO MOLECULAR MEDICINE
Manuscript Number: EMM-2020-13579